# Geographical migration and fitness dynamics of *Streptococcus pneumoniae*

Sophie Belman[1,2,3 ✉], Noémie Lefrancq[2], Susan Nzenze[4], Sarah Downs[5], Mignon du Plessis[6,7], Stephanie W. Lo[1,8], The Global Pneumococcal Sequencing Consortium*, Lesley McGee[9], Shabir A. Madhi[5,10], Anne von Gottberg[6,7,11], Stephen D. Bentley[1,67] & Henrik Salje[2,67]

*Streptococcus pneumoniae* is a leading cause of pneumonia and meningitis worldwide. Many different serotypes co-circulate endemically in any one location[1,2]. The extent and mechanisms of spread and vaccine-driven changes in fitness and antimicrobial resistance remain largely unquantified. Here using geolocated genome sequences from South Africa (*n* = 6,910, collected from 2000 to 2014), we developed models to reconstruct spread, pairing detailed human mobility data and genomic data. Separately, we estimated the population-level changes in fitness of strains that are included (vaccine type (VT)) and not included (non-vaccine type (NVT)) in pneumococcal conjugate vaccines, first implemented in South Africa in 2009. Differences in strain fitness between those that are and are not resistant to penicillin were also evaluated. We found that pneumococci only become homogenously mixed across South Africa after 50 years of transmission, with the slow spread driven by the focal nature of human mobility. Furthermore, in the years following vaccine implementation, the relative fitness of NVT compared with VT strains increased (relative risk of 1.68; 95% confidence interval of 1.59–1.77), with an increasing proportion of these NVT strains becoming resistant to penicillin. Our findings point to highly entrenched, slow transmission and indicate that initial vaccine-linked decreases in antimicrobial resistance may be transient.

The greatest public health burden from infectious diseases remains stubbornly endemic pathogens. Once established, pathogens such as *Mycobacterium tuberculosis*, HIV and, now, SARS-CoV-2 are difficult to control, even when vaccines are available[1]. Their persistence in the population can be partially explained by the co-circulation of multiple strains of the same pathogen. Endemic pathogens are complicated to study as we rarely understand the mechanisms that drive spread, including the role of human behaviour and why some lineages increase in prevalence over time whereas others disappear. Underlying genetic diversity is particularly extreme in the case of the bacterium *S. pneumoniae* (the pneumococcus), which is the leading cause of morbidity and mortality worldwide because of lower respiratory infections[2–4]. The pneumococcus comprises >100 known antigenically distinct serotypes and >900 classified lineages (also known as global pneumococcal sequence clusters (GPSCs))[5–7]. Moreover, it is not uncommon for more than 30 antigenically distinct serotypes to co-circulate within a country or region or for a human host to concurrently carry multiple serotypes[8]. We refer to lineage synonymously with GPSC, whereas a strain

references any particular circulating phenotype (including specific serotypes and antimicrobial resistance (AMR)). Here we develop mathematical models using thousands of geolocated genome sequences from South Africa collected over a 15-year period to clarify several key uncertainties in pneumococcal migration. We model the rate and breadth of mobility geographically and how fitness changes linked to vaccine implementation and AMR may affect its spread.

The pneumococcus resides in the human upper respiratory tract. Carriage is a prerequisite for disease, and rates of carriage in children under 5 years old range from 20 to 90%[9]. Occasionally, asymptomatic carriage goes on to cause local infections such as otitis media or, more severely, invasive pneumonia and meningitis. More than 500,000 deaths per year linked to pneumococcus are estimated to occur globally[3,10]. Penicillin was first used to treat pneumococcal disease in the 1930s, and it successfully reduced pneumococcal disease until the late 1960s when penicillin non-susceptibility was first noted. Multidrug-resistant strains were described soon after penicillin resistance[11,12]. By 2019, 19% of deaths associated with AMR had pneumococcal aetiology[13]. In this

[1]Parasites and Microbes, Wellcome Sanger Institute, Hinxton, UK. [2]Department of Genetics, University of Cambridge, Cambridge, UK. [3]Global Health Resilience, Earth Sciences Department, Barcelona Supercomputing Center, Barcelona, Spain. [4]Division of Public Health Surveillance and Response, National Institute for Communicable Diseases of the National Health Laboratory Service, Johannesburg, South Africa. [5]South African Medical Research Council Vaccines and Infectious Diseases Analytics Research Unit, School of Pathology, Faculty of Health Sciences, University of the Witwatersrand, Johannesburg, South Africa. [6]Department of Clinical Microbiology and Infectious Diseases, School of Pathology, Faculty of Health Sciences, University of the Witwatersrand, Johannesburg, South Africa. [7]Centre for Respiratory Diseases and Meningitis, National Institute for Communicable Diseases of the National Health Laboratory Service, Johannesburg, South Africa. [8]Milner Centre for Evolution, Department of Life Sciences, University of Bath, Bath, UK. [9]National Center for Immunization and Respiratory Diseases, Centers for Disease Control and Prevention, Atlanta, GA, USA. [10]Department of Science and Technology/National Research Foundation, South African Research Chair Initiative in Vaccine Preventable Diseases, Faculty of Health Sciences, University of the Witwatersrand, Johannesburg, South Africa. [11]Division of Medical Microbiology, Department of Pathology, Faculty of Health Sciences, University of Cape Town, Cape Town, South Africa. [67]These authors jointly supervised this work: Stephen D. Bentley, Henrik Salje. *A list of authors and their affiliations appears at the end of the paper. ✉e-mail: sophie.belman@sanger.ac.uk

context, vaccines are pivotal to disease control. A pneumococcal polysaccharide vaccine that included 23 serotypes was licensed in the USA in 1983. However, the absence of mucosal immunity has seen it replaced by pneumococcal conjugate vaccines (PCVs) except for in older adults and immunocompromised individuals[14]. PCVs (conjugated with toxin to stimulate mucosal immunity) target a small subset of the polysaccharide capsular serotypes, with the most common formulations including PCV7 and PCV13 (Pfizer)[15] and PCV10 (GlaxoSmithKline)[16]. These target 7, 13 and 10 serotypes, respectively (with all the serotypes included within PCV7 and PCV10 also included within PCV13). In 2021, additional PCV formulations, PCV15 (Merck) and PCV20 (Pfizer), were licensed for use in the USA and Europe[17]. Pneumococcal vaccination is dynamic, and new vaccine compositions are frequently tested. Vaccine serotypes are often selected because of their high prevalence and AMR among disease isolates from infants and children. PCVs are now included in 76% of national immunization schedules, with different formulations in different countries. For example, in South Africa, PCV7 was implemented in 2009 and replaced by PCV13 in 2011, excluding PCV10 (ref. 18). Despite their success at reducing disease, their use has been linked to serotype replacement by NVTs in both invasive pneumococcal disease (IPD) and carriage[8,19–21]. In South Africa, this has been characterized by increases in NVT serotypes 8 and 15B among IPD, and increases in NVT serotypes 16F, 24, 35B and 11A among carriage isolates (8, 15B and 11A are now included in PCV20)[20,22,23]. Although there has been success in predicting the fitness of individual isolates based on the overall gene distribution in a population, quantitative measures of fitness linked to the serotype of individual isolates are lacking[24,25]. This includes quantifying the time it takes after implementation for vaccines to affect the serotype composition in the country. Moreover, quantifying the serotype growth before and after vaccine implementation at different time points is needed. These are crucial knowledge gaps, as serotype distributions ultimately drive vaccine development and deployment strategies. In addition, vaccine implementation has resulted in reductions in AMR among both IPD and carriage isolates[20,22,26]. However, it remains unclear whether these reductions will persist over time at the population level or whether AMR may rebound.

Mathematical models applied to geolocated pathogen genome sequence data are useful to disentangle the changing prevalence of different lineages. However, most phylogeographical models focus on the rate of pathogen flow between locations, which represents the overall effect of multiple transmission chains. Such models therefore consider a different ecological scale to the specific behaviours of infected people and the surrounding population at each transmission generation[27]. The relationship between behaviours of individuals at each transmission step and the overall patterns of pathogen flow between locations are complex and nonlinear. Most existing phylogeographical approaches also struggle to account for changing levels of surveillance in both space and time. Here we develop mechanistic models that use the generation time distribution to estimate the number of transmission events that separate the most recent common ancestors (MRCAs) from each pair of tips in a time-resolved phylogeny. Taken together with measures of human mobility probabilities and human population distribution, we can infer mechanisms of pneumococcal migration at each transmission generation. We implemented this model with a focus on South Africa, where approximately 65% of children ≤5 years of age (35% across all age groups) carry the pneumococcus[28,29]. We incorporate both uncertainty in the phylogenetic reconstructions and sampling uncertainty through a bootstrapping approach[30]. We explore the robustness of our approach to highly biased observation processes using simulated data with known parameter values. Finally, we quantify the changing fitness of strains in response to vaccine introduction, including those containing AMR.

## Quantifying spatial structure

In partnership with the South African National Institute for Communicable Disease and the Wits Vaccines and Infectious Diseases Analytics

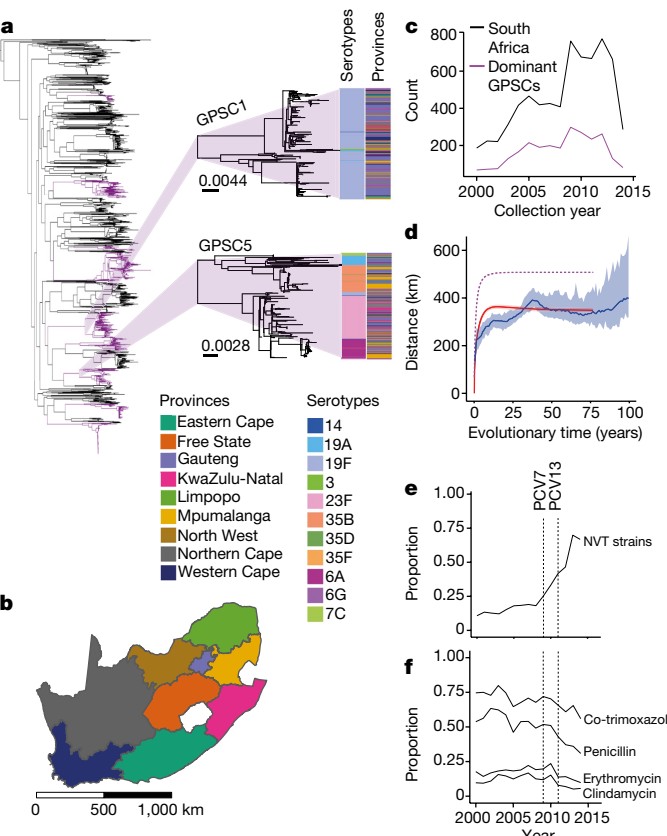

**Fig. 1 | Descriptive summary of *S. pneumoniae* isolates. a**, Phylogenetic tree of 6,910 South African isolates included in this study. Dominant GPSCs (*n* > 50) are in purple. GPSC1 (top) and GPSC5 (bottom) are highlighted. The columns describe the serotypes and provincial region for each isolate. The branch length legends refer to single nucleotide polymorphisms (SNPs) per site and trees are midpoint rooted. **b**, Map of the nine provinces of South Africa coloured by province. Scale bar is included in kilometres (km). **c**, Count of isolates (*n* = 6,910) per collection year from 2000 to 2014 used in the lineage-level analysis (black) and the 9 dominant GPSCs used in the divergence time analysis (maroon). **d**, The mean geographical distance for sequence pairs as a function of cumulative evolutionary distance across all GPSCs with 95% CI (blue). The model fit is shown in red. The implied true pattern of spread is shown in purple, after accounting for a biased observation process. **e**, The proportion of NVT serotypes across the study period. **f**, The proportion of in silico predicted AMR isolates for four drugs across the study period. The vertical lines denote the introduction of PCV7 in 2009 and PCV13 in 2011. An interactive phylogeny and metadata are available at Microreact (https://microreact.org/project/7wqgd2gbBBEeBLLPKonbaT-belman2024southafricapneumococcus).

Research Unit, we sequenced the whole genomes of isolates from each of South Africa's nine provinces between 2000 and 2014 (*n* = 6,910, 5,060 from individuals with invasive pneumococcal disease and 1,850 from carriage studies) (Fig. 1a–c). Despite the large number of sequences, this dataset only represents a very small proportion (≪0.1%) of the circulating pneumococcus over this time period. We identified 184 GPSCs with 69 different serotypes (31.9% NVT) (Fig. 1a,e and Supplementary Table 1). This diversity persisted across provinces, and the distribution of serotypes within GPSCs did not follow a distinct geographical structure (Fig. 1a). In silico predicted AMR was common (penicillin, 48.2%; erythromycin, 17.3%, clindamycin, 11.2%; and co-trimoxazole: 68.4%), with similar distributions for isolates from both carriage and disease (Fig. 1f, Extended Data Fig. 10, Supplementary Fig. 13 and Supplementary Table 3).

Taking the 9 most dominant GPSCs in turn (each comprising more than 50 sequences, 2,575 sequences in total), we built recombination-free, time-resolved phylogenetic trees to determine the divergence

times between sequence pairs (Extended Data Fig. 1a–i and Supplementary Table 4). We included 1,157 genomes from 14 other countries in Africa and 2,944 from 31 countries outside Africa in our phylogenies. We compared the geographical distance spread per divergence time between pairs in South Africa and found a clear geographical structure. The geographical distance between pairs increased from a mean distance of 142 km (95% confidence interval (CI) = 54–207 km) for those separated by less than 2 years of evolutionary time to 297 km (95% CI = 274–323 km) for those separated by 10–20 years (Fig. 1d). We obtained consistent results when using only sequences that came from individuals with disease (Supplementary Fig. 1A and Supplementary Table 5) and across the different GPSCs (Supplementary Fig. 2). This result is consistent with largely common patterns of spread irrespective of which particular GPSC an individual is infected with. Despite high heterogeneity in GPSC composition within any province, overall, pairs of isolates that are from the same province had 1.27 (95% CI = 1.23–1.31) times the relative risk (RR) of being the same GPSC compared with pairs of isolates from distant provinces (>1,000 km). This RR fell to 1.09 (95% CI = 1.03–1.12) for pairs separated by 500–1,000 km (Fig. 2a and Supplementary Table 5). We obtained consistent results when we limited our analysis to only disease isolates and when we subsampled to have even numbers of sequences by province to mitigate sampling bias (Extended Data Fig. 2a,b and Supplementary Table 5). As there can be hundreds of years of diversity within a single GPSC, we refined the analysis by using different evolutionary windows of separation between pairs of isolates as determined from the phylogenetic trees. Pairs from the same province had 3.87 (95% CI = 3.01–5.1) times the RR of having a recent common ancestor (within 5 years) than distal pairs (>1,000 km apart) (Fig. 2b and Supplementary Table 5). However, as the evolutionary time between isolates increases the relative probability of being from the same province decreases, it is only after around 50 years that the pneumococcus seemed to be well mixed throughout the country (Fig. 2c–f and Supplementary Table 6). Furthermore, comparisons of the spatial location of closely related pairs showed that pneumococcus flow was dominated by within-country movement compared with either movement between South Africa and other African countries or non-African countries (Fig. 2c–e, Supplementary Fig. 3A and Supplementary Table 5). Recognizing that our samples span age groups, we stratified the analysis by the age difference between pairs. Genome pairs from individuals who were greater than 5 years apart in age took slightly longer to become mixed across South Africa (Supplementary Fig. 3B and Supplementary Table 7). These findings are consistent with a highly entrenched pathogen that moves slowly within a country and with slow cross-border transmission.

## Inferring migration using human mobility

To understand whether human mobility can explain the slow spread of the pneumococcus, we built a mechanistic model of geographical spread fit to the nine dominant GPSCs and the observed province in which our genome sequences were isolated. We used the generation time distribution, time from one person being infected to infecting the next person (estimated mean of 35 days, standard deviation of 35 days (gamma distribution)) to translate branch lengths to the number of generations between pairs of sequences[31,32] (Supplementary Fig. 4). Each transmission generation is an opportunity for pneumococcal mobility. We used directional human mobility probabilities between each of the 234 South African municipalities from Meta Data for Good[30] to infer the probable location of a single transmission event, allowing for mobility of both the infected individual and the surrounding susceptible population. As Meta data captures mobility over a single day, we adjusted the duration in any cell to consider total mobility over the infectious period. Furthermore, as Meta users may act differently to those involved in pneumococcal transmission, we incorporated a parameter that allows individuals to have a different probability of staying within their home

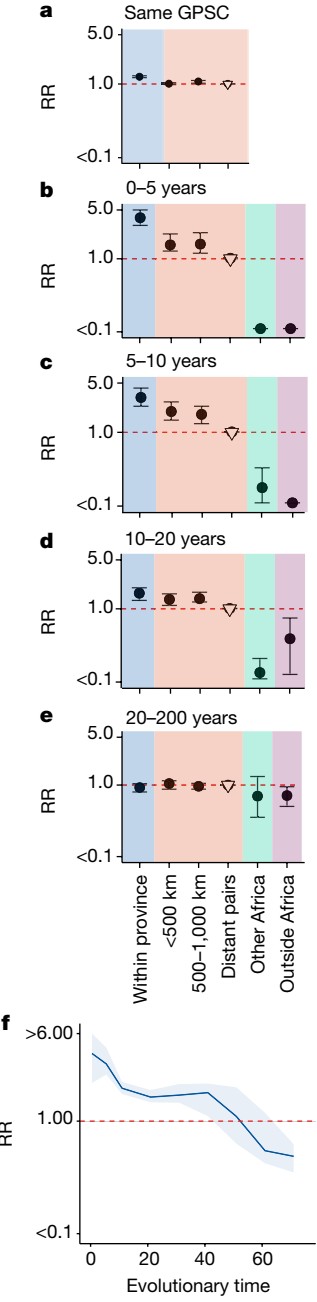

**Fig. 2 | RR framework to determine geographical structure. a**, RR of being the same GPSC within a province (blue), between different provinces over increasing distance (red) and compared with geographically distant pairs (>1,000 km) (reference). (South Africa; *n* = 6,910). **b–e**, RR of having a time to most recent common ancestor (tMRCA) 0–5 years (**b**), 5–10 years (**c**), 10–20 years (**d**) and 20–200 years (**e**) ago within South African provinces (blue), across larger distances within South Africa (red), from South Africa to other countries in Africa (*n* = 1,157) (green), and from South Africa to countries outside of Africa (*n* = 2,944) (purple). All plots use a reference of pairs that are from distant provinces in South Africa (open triangle). **f**, RR of similarity over rolling 20-year windows of divergence times for pairs isolated within the same South African province compared with pairs from distant provinces in South Africa (>1,000 km apart). For **a–f**, plots are centred at the median and error bars represent 2.5 and 97.5 percentiles across posterior phylogenies.

municipality than Meta users (Fig. 3b). We then calculated the probability of pneumococcal movement between each pair of locations for each transmission generation. This approach integrates over all possible pathways linking two locations. We incorporated the probability of an

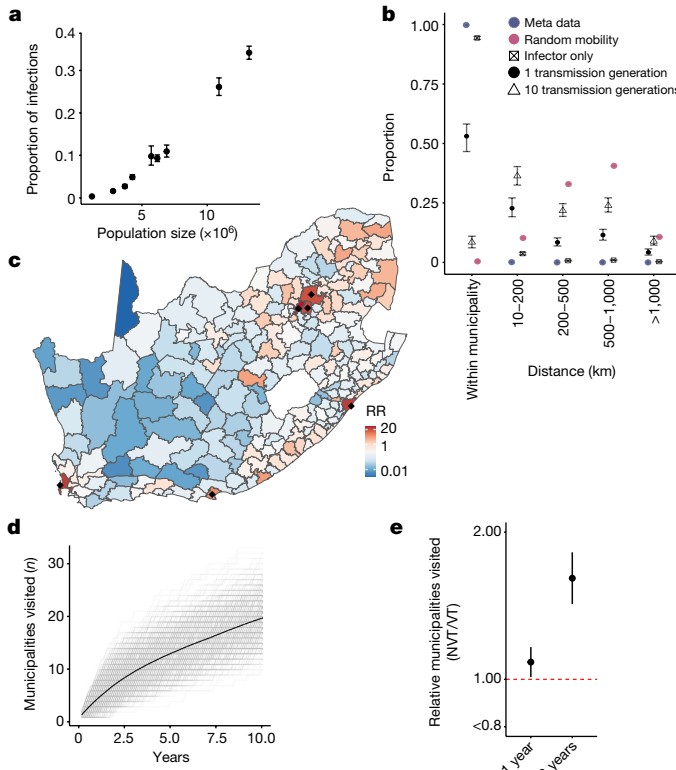

**Fig. 3 | Mechanisms of geographical migration. a**, The estimated probability of the location (province) of each MRCA (*y* axis) compared with the population size (*x* axis) in that province. Points are centred at the median and error bars represent 95% credible intervals. **b**, The proportion of individual or pathogen mobility as a function of distance from the origin location. We compared the mean distance travelled when we consider the infector only (crossed square), when we consider the mobility of both the infector and the infectee (black filled circle) after a single transmission generation, as well as the overall mobility of the pathogen after ten generations (triangle). As a comparison, we present the expected pathogen spread after a single generation if transmission was completely spatially random (maroon). We also present the difference between Meta users (blue circle) and the movement of those involved in transmission. Points are centred at the median and error bars represent 95% credible intervals. **c**, The RR of being in each of the 234 municipalities of South Africa after 1 year (10 transmission generations) of sequential person-to-person transmission compared with being in a randomly selected municipality. Black dots denote municipalities with populations of >3 million people. **d**, The number of unique municipalities visited for 500 unique sequential simulations (grey) and the mean (black-dashed) across years of transmission following an introduction in a randomly selected municipality given the modelled migration probabilities at each transmission generation. **e**, The relative number of unique municipalities visited by NVT serotypes compared with VT serotypes after 1 and 2 years of transmission. Points are centred at the median and error bars represent 95% credible intervals.

isolate being sequenced at each geographical location and within each collection year to account for the number of isolates being sequenced differing by year and location. We fit the model in a Bayesian framework using Markov chain Monte Carlo (MCMC). We compared the performance of our models using the Meta data to a gravity model in which the probability of mobility is a function of the distance to a location and its population size (Extended Data Fig. 4). We also separately modelled mobility that depended on distance only.

Both the gravity model and the model that relied on Meta data were able to recover the observed spatial spread in pneumococcus (Fig. 1d, red line). However, model fit was better for the Meta data model (difference in deviance information criterion (DIC) of 3.4). Both these models outperformed the distance-only model (Supplementary Table 8). The Meta

data model showed that the relative proportion of the population carrying the pneumococcus per province at any time is strongly correlated with the population size in that province ($R^2 = 0.97$, $P \leq 0.01$; Fig. 3a). This model enabled us to infer the true underlying spread of pneumococcus (that is, in which we accounted for the biased observation process and mobility in both the infected individual and the surrounding susceptible population) (Fig. 1d, dashed line). We estimated that among individuals involved in pneumococcal transmission, the daily probability of staying in their home municipality was 94.3% (95% CI = 93.8–95.0%) (Fig. 3b, black squares) compared with 99.8% (ranging from 86.6% in Mogale City, Gauteng, to 99.9% in Ba-Phalaborwa, Limpopo) for Meta users (Fig. 3b, blue points). When we incorporated the mobility of both infector and infectee and accounted for the mobility across the infectious period, we estimated that after a single transmission generation (35 days), 53.1% (95% CI = 46.6–58.2) of strains remained in their starting municipality, 22.8% (95% CI = 19.2–27.1%) were in a neighbouring municipality and a small minority were more than 500 km away (Fig. 3b, black points). As the number of transmission generations increased, the probability of reaching distal municipalities also increased (Fig. 3b). The size of the community seemed to be key to determining where lineages travel. After 1 year of sequential transmission, the probability of being in a municipality with a population size of >3 million people was 26.7 (95% CI = 19.8–40.10) times that of being in a randomly selected municipality. This result is consistent with most pathogen movement passing through urban centres (Fig. 3c and Supplementary Table 9).

The municipality in which a strain emerges also seemed important. After 1 year of sequential transmission, a new strain that first occurred in a rural municipality (population density of <50 people per km²) has travelled a median distance of 468.7 km (95% CI = 71.3–1,204.4 km), whereas in the same time window, a variant first occurring in an urban municipality (>500 people per km²) has travelled only 285.6 km (95% CI = 36.3–967.0 km). Furthermore, the variant that emerged in a rural municipality would have travelled to 1.53 times as many municipalities as the urban variant (Extended Data Fig. 3a–c). This result is corroborated by previous research that demonstrated high levels of in and out migration among individuals in rural settings owing to travel for work or education[33,34]. On average, 1 year after emerging, transmission chains visited 4 (95% CI = 1–8) municipalities, and after 10 years they visited 20 (95% CI = 13–27) municipalities (Fig. 3d). Overall, these results show that the breadth of geographical spread is driven by a small number of long-range transmission events, with most transmissions remaining local. Incorporating our model into a branching epidemic, we found that after 10 years, transmission events are an average 465 km (95% CI = 456–472 km) from where they began (Supplementary Fig. 5A,B).

To test the performance of our model, we simulated transmission within and between districts using the Meta matrix adjusted by a known parameter that determines the probability of staying within one's home district. We then fit our model using a subset of infections to re-estimate this parameter. Even when only a small minority of infections were sequenced, we recovered the true probability of staying within a district. Our estimates were even robust to an extremely biased observation scenario whereby data from only two locations were available. Using this framework, we also explored the effect of misspecification of the generation time distribution. Using a 50% shorter generation time led to a small overestimate in the probability of staying in one's home location each day (96.7% versus 95.3%), whereas using a 50% longer generation time had the opposite effect (Extended Data Fig. 5).

## Vaccine-induced fitness changes

Implementation of PCV7 in 2009 and PCV13 in 2011 was associated with a substantial disruption in the patterns of circulating serotypes, results that are consistent with what has been previously observed[8,20,35,36]. However, the introduction of vaccines was also associated with a marked

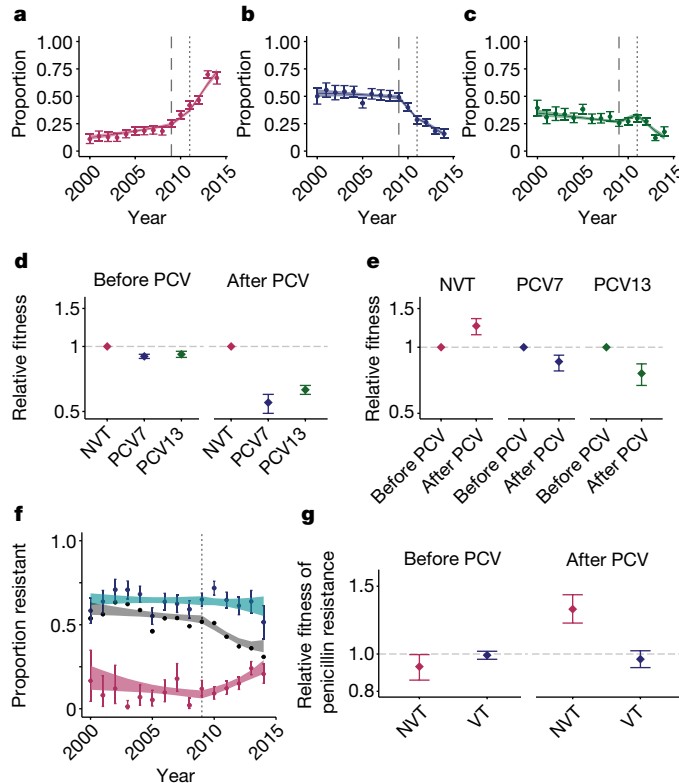

**Fig. 4 | Vaccine-induced fitness dynamics. a–c**, Data (points) and model fit (lines) for the proportion of serotypes from NVTs (**a**), PCV7 types (**b**) and additional PCV13 types not included in PCV7 (**c**) from the years 2000 to 2014 in this study. The long dashed line indicates the time of PCV7 implementation (2009) and the short, dashed line indicates the time of PCV13 implementation (2011). **d**, Relative fitness for the three groups of serotypes compared with the NVT fitness estimates before and after PCVs were introduced. **e**, Relative fitness estimates for all three groups of serotypes comparing the before and after PCV eras. For **a–e**, before PCV refers to before 2009 for NVT serotypes, before 2009 for PCV7 type serotypes and before 2011 for PCV13 type serotypes. **f**, Proportion of penicillin resistance overall (black line), within NVT strains (maroon points) and within VT strains (turquoise points) with model fits. The dashed line indicates the time of PCV implementation (2009). **g**, Relative fitness of penicillin resistance among NVTs (pink) and VTs (blue) in before (left) and after (right) PCVs. Data in **d**, **e** and **g** are on a log scale. For **a–f**, plots are centred at the median and include error bars representing 95% credible intervals around the posterior parameter distributions (*n* = 6,798).

change in fitness. By 2014, serotypes included in PCV13 represented 33.2% of all isolates in our dataset, a reduction from 85.0% in the pre-vaccine era (Fig. 4a). These patterns were consistent across the nine provinces in South Africa (Supplementary Fig. 6). To quantify changes in fitness linked to the vaccines, we fitted models to the annual distribution of serotypes across 184 GPSCs from the full dataset, allowing for differential fitness in serotypes included in PCV7 (serotypes 4, 6B, 9V, 14, 19F, 18C and 23F), PCV13 (which includes additional serotypes 1, 3, 5, 6A, 7F and 19A), and those not included in the vaccine (NVT). This method tracks the proportion of all serotypes at the population level over time and quantifies the relative advantage of each of them following the implementation of vaccines. This simple formulation was able to recover the observed distribution of serotype proportions in each group, by year, across provinces (Fig. 4a–c and Supplementary Fig. 6). We note that the number of NVT and VT isolates we used in our model do not represent the underlying incidence of NVT and VT, as only a small proportion of all infections were detected and sequenced. However, as we focused on the relative abundance of NVT and VT strains per year, our approach is robust to changes in the absolute numbers of isolates sequenced.

Before the implementation of vaccines (2000–2008), NVTs had a relative fitness of 1.08 (95% CI = 1.06–1.09) compared with serotypes included in the vaccine (combining serotypes in PCV7 and PCV13 into VT). Following the implementation of PCV7 and PCV13, the fitness of the serotypes they target declined respectively compared with NVT serotypes (Fig. 4d and Supplementary Tables 10 and 11). When comparing serotype fitness before and after the introduction of PCVs, VTs had a relative fitness of 0.86 (95% CI = 0.78–0.92) and 0.76 (95% CI = 0.67–0.84) for PCV7 and PCV13 serotypes, respectively (Supplementary Table 11). Meanwhile, for the NVTs, vaccines were associated with a 1.25 (95% CI = 1.14–1.35) times increase in relative fitness from 2009 to 2014 compared with before the implementation of vaccines (Fig. 4e and Supplementary Tables 10 and 11). When we directly compared the fitness advantage of NVTs compared with VTs in the PCV era, NVTs had a relative fitness advantage of 1.68 (95% CI = 1.59–1.77), which is equivalent to a 1.05 (95% CI = 1.05–1.06) growth advantage (relative to VTs) at each transmission generation. In a sensitivity analysis, the results remained consistent for carriage and disease isolates, respectively, despite them having been sampled from different cohorts (Extended Data Fig. 6 and Supplementary Fig. 7). We also found consistent results across provinces (Supplementary Fig. 6). We additionally assessed whether there was a delay between vaccine implementation and resulting changes in strain fitness (Extended Data Fig. 7). The best fitting model assumed the change in fitness occurred in the same year as vaccine implementation.

Refining this model to look at the fitness of individual serotypes showed a wide range of fitness across serotypes. Our findings highlight that all strains are fundamentally different in underlying fitness (Extended Data Fig. 8), which means that NVTs will differ in their ability to alter their ecological niche following changes in vaccine formulation. This result needs to be taken into consideration in the development of new vaccine formulations. Among NVTs, serotypes 15A, 35B and 8 had the greatest fitness advantage after PCVs were used (Supplementary Figs. 8 and 9), a result concordant with what has been previously observed[20,22,37]. Shifts in lineage fitness also resulted in changing patterns of spread. By incorporating our fitness estimates for NVT and VT strains after vaccination into our mobility model, we estimated that the number of affected municipalities from a strain of a NVT was 2.02 (95% CI = 1.81–2.25) times the number of affected municipalities from types included in the vaccine (Fig. 3e).

We next explored whether using the proportion of isolates that were VT versus NVT within each GPSC at the start of the study period and our fitness estimates could explain the subsequent dynamics of individual GPSCs. Simply using VT and NVT fitness estimates could explain 60% of the variance in individual GPSC prevalence at any time. Allowing for serotype-specific differences produced a small improvement, explaining 65% of the variance (Supplementary Fig. 10 and Supplementary Table 13). The unexplained variance reflects GPSC-specific fitness and is probably driven by negative-frequency dependent selection (NFDS)[24]. This result highlights the predictive nature of the serotype composition of GPSCs in determining PCV-driven GPSC dynamics (Supplementary Figs. 11 and 12).

The serotypes included in the vaccines were prevalent in childhood disease and had high levels of AMR in the USA, where the vaccines were developed[38]. The high levels of AMR in the VT strains was also present globally[26]. In South Africa, similar to other countries, reductions in AMR have been noted since vaccine implementation; however, it remains unclear whether this trend will persist or whether AMR eventually rebounds[26]. In South Africa, before vaccines, 63.6% of VT and 8.8% of NVT strains were resistant to penicillin. We found that there was a clear reduction in overall penicillin resistance following vaccine implementation, which was driven by reductions in the proportions of strains that are VT (Fig. 4f and Supplementary Table 12). The trends, although still present, were less clear in the other investigated antimicrobials (Extended Data Fig. 9). Owing to the relevance of penicillin as a first-line antimicrobial for pneumococcal disease and the high proportion of

resistance in this population, we used our same modelling framework and were able to recover the observed proportions of strains that were resistant to penicillin over time. Before vaccines, among both NVT and VT strains, there was limited difference in the fitness between those that were penicillin-resistant and penicillin-susceptible. However, following implementation of vaccines, NVT-resistant strains were 1.30 (95% CI = 1.19–1.43) times as fit as penicillin-susceptible NVT strains (Fig. 4f,g and Supplementary Fig. 19). Conversely, resistance did not seem to have changed among VT penicillin-resistant strains (relative fitness of 0.97 (95% CI = 0.91–1.03)) (Fig. 4f,g, Supplementary Fig. 13A–C and Supplementary Table 12). Expansion of NVTs within typically VT-associated lineages is the most common mechanism for serotype replacement[36]. As a result, the penicillin resistance associated with these newly expanded NVT lineages is able to persist in the population[19,36]. Together with our quantification of growing penicillin resistance among NVTs following the use of PCVs, this result suggests that the overall reduction in penicillin resistance seen following vaccine implementation may not persist. Our data also highlight the nuanced effect that vaccines can have on patterns of AMR (Supplementary Fig. 13D–F). It is probable that next-generation higher valency PCVs may also lead to increased AMR prevalence in those serotypes not included in the vaccine. Changing patterns of antimicrobial use in this population may also result in shifts in resistance patterns. The widespread presence of penicillin resistance at the beginning of our data collection period would implicate expansion of resistance among existing lineages; however, we cannot exclude the emergence of some new resistance forms.

## Limitations

We did not have complete carriage and invasive disease data across the entire time period. However, we performed sensitivity analyses to determine whether our results are robust to including carriage and invasive disease together. The human mobility data are Meta baseline data that were released by Meta owing to the SARS-CoV-2 pandemic in 2020. We used aggregated data across 17 months (January 2020 to June 2021) within a single mobility pattern matrix. As mobility was altered during this period and to address the possibility that Meta mobility data are different to the movement of individuals involved in pneumococcal transmission, we included an additional parameter that adjusts the human mobility data to account for more or less time being spent in home municipalities. As we were able to obtain good fits to the observed spread of pneumococcus, and these models outperformed standard gravity models, our findings highlight how imperfect mobility data can nevertheless be useful. Vaccination levels for PCV7 were reported to be 89% by 2022 (ref. 39). We did not have the data to consider changes in coverage in time or across provinces. Other settings with different levels of coverage may observe different fitness effects from the implementation of the vaccine. Within the fitness model, as we were looking at relative proportions of strains with increasing proportions of resistance, there may still be decreased total burden of resistant disease if those strains carrying it remain low in prevalence.

## Conclusion

Here we quantified and explained the movement of a persistent human pathogen for which the geographical course has been largely hidden by its diversity and endemicity. The pneumococcus has an affinity for urban centres through which it channels its wider geographical spread. Although it is characterized by slow transmission overall, the use of vaccines can substantially and rapidly change pneumococcal lineage ecology. Although vaccine-associated fitness dynamics have been previously described in the pneumococcus[8,36], they have not been directly quantified. Increasing proportions of NVTs in the disease isolates from the PCV era can be largely attributed to the decrease in number of VTs rather than the increasing prevalence of NVTs[22]. Vaccination has had a secondary

effect on penicillin resistance, with a decrease in recent years in South Africa. Given the estimated growth advantage of penicillin-resistant NVT strains, we may see a reversal of this benefit; however, estimating the carrying capacity of this growth is beyond the scope of this model. Furthermore, we quantified pneumococcal geographical spread and the spatial impact of NVT expansion after vaccination. Our findings highlight how directly observed characterizations of human mobility using mobile phone data or Meta data can be used to obtain a mechanistic understanding of how pathogens spread within phylogeographic frameworks. This includes considering mobility of both infected individuals and the susceptible population and how we can adjust these datasets to account for systematic differences in behaviour between mobile phone data and Meta data and those involved in transmission. We note that basic gravity models also performed well, providing a useful alternative in settings in which mobile phone data options are not available. Our description of pneumococcal geographical spread provides new insight into the movement of emergent strains. In South Africa, the population density in the emergence location of a NVT strain may affect its speed of spread across the country and have implications for public health responses to emergent strains. Emergence in a highly mobile, peri-urban area may enable both rapid proximal geographical spread and less frequent distal seeding events, and thus more complete distribution across the country. The fitness model and human mobility model together provide frameworks to quantify and better understand the migratory and fitness dynamics of this globally endemic pathogen. The magnitude of South Africa and its provinces demonstrates that these frameworks may be applied to other large regions.

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

**The Global Pneumococcal Sequencing Consortium**

Alejandra Corso[12], Paula Gagetti[12], Abdullah W. Brooks[13], Md Hasanuzzaman[14], Samir K. Saha[14], Senjuti Saha[14], Alexander Davydov[15], Leonid Titov[15], Samanta Cristine Grassi Almeida[16], Paul Turner[17], Chunjiang Zhao[18], Hui Wang[18], Margaret Ip[19], Pak Leung Ho[20], Pierra Law[20], Jeremy D. Keenan[21], Robert Cohen[22,23,24], Emmanuelle Varon[25], Eric Sampane-Donkor[26], Balaji Veeraraghavan[27], Geetha Nagaraj[28], K. L. Ravikumar[28], J. Yuvaraj[28], Varun Shamanna Noga[28], Rachel Benisty[29], Ron Dagan[29], Godfrey Bigogo[30], Jennifer Verani[31], Anmol Kiran[32], Dean B. Everett[33,34], Jennifer Cornick[32], Maaike Alaerts[32], Shamala Devi Sekaran[35], Stuart C. Clarke[36], Benild Moiane[37], Betuel Sigauque[37], Helio Mucavele[37], Andrew J. Pollard[38], Rama Kandasamy[39], Philip E. Carter[40], Stephen K. Obaro[41,42], Deborah Lehmann[43], Rebecca Ford[44], Theresa J. Ochoa[45], Anna Skoczynska[46], Ewa Sadowy[46], Waleria Hryniewicz[46], Weronika Puzia[46], Sanjay Doiphode[47], Ekaterina Egorova[48], Elena Voropaeva[48], Yulia Urban[48], Tamara Kastrin[49], Kedibone Ndlangisa[7], Linda De Gouveia[7], Mushal Ali[7], Nicole Wolter[7], Cebile Lekhuleni[7], Carmen Muñoz Almagro[50], Alba Redin Alonso[50], Desiree Henares[50], Somporn Srifuengfung[51], Brenda Kwambana-Adams[32], Ebenezer Foster-Nyarko[32], Ebrima Bojang[32], Martin Antonio[32,52,53,54], Peggy-Estelle Tientcheu[32], Jennifer Moïsi[55], Michele Nurse-Lucas[56], Patrick E. Akpaka[56], Özgen Köseoglu Eser[57], Anthony Scott[58], David Aanensen[59], Nicholas Croucher[60], John A. Lees[61], Rebecca A. Gladstone[62], Gerry Tonkin-Hill[62], Chrispin Chaguza[63], David Cleary[64], Kate Mellor[1], Bernard Beall[9], Keith P. Klugman[65], Gail Rodgers[65], Paulina A. Hawkins[9], Anne J. Blaschke[66] & Nicole L. Pershing[66]

[12]Servicio Antimicrobianos, National Reference Laboratory (NRL), Instituto Nacional de Enfermedades Infecciosas (INEI)-ANLIS 'Dr Carlos G. Malbrán', Buenos Aires, Argentina. [13]International Centre for Diarrhoeal Diseases Research, Dhaka, Bangladesh. [14]Child Health Research Foundation, Dhaka, Bangladesh. [15]Department for Microbiology, Virology and Immunology, Belarusian State Medical University, Minsk, Belarus. [16]Center of Bacteriology, Institute Adolfo Lutz, São Paulo, Brazil. [17]Cambodia–Oxford Medical Research Unit, Angkor Hospital for Children, Siem Reap, Cambodia. [18]Peking University People's Hospital, Beijing, China. [19]Department of Microbiology, Faculty of Medicine, The Chinese University of Hong Kong–Prince of Wales Hospital, Hong Kong, China. [20]Department of Microbiology and Carol Yu Centre for Infection, The University of Hong Kong–Queen Mary Hospital, Hong Kong, China. [21]Francis I. Proctor Foundation and Department of Ophthalmology, University of California San Francisco, San Francisco, CA, USA. [22]Clinical Research Center, Centre Hospitalier Intercommunal de Créteil, Créteil, France. [23]Université Paris Est, IMRB-GRC GEMINI, Créteil, France. [24]Groupe de Pathologie Infectieuse Pédiatrique de la Société Française de Pédiatrie, Nice, France. [25]National Reference Center for Pneumococci, Data and Research Department – DRIM, Centre Hospitalier Intercommunal de Créteil, Créteil, France. [26]Department of Medical Microbiology, University of Ghana Medical School, Accra, Ghana. [27]Department of Clinical Microbiology, Christian Medical College, Vellore, India. [28]Central Research Laboratory, Kempegowda Institute of Medical Sciences, Bangalore, India. [29]The Shraga Segal Department of Microbiology, Immunology and Genetics, Faculty of Health Sciences, Ben-Gurion University of the Negev, Beer-Sheva, Israel. [30]Center for Global Health Research, Kenya Medical Research Institute (KEMRI), Kisumu, Kenya. [31]Division of Bacterial Diseases, National Center for Immunization and Respiratory Diseases, Centers for Disease Control and Prevention, Atlanta, GA, USA. [32]Malawi–Liverpool–Wellcome Trust Clinical Research Programme, University of Malawi College of Medicine, Blantyre, Malawi. [33]Department of Public Health and Epidemiology, College of Medicine and Health Sciences, Khalifa University, Abu Dhabi, UAE. [34]Infection Research Unit, Khalifa University, Abu Dhabi, UAE. [35]Faculty of Applied Sciences, UCSI University, Kuala Lumpur, Malaysia. [36]Faculty of Medicine and Life Sciences, University of Southampton, Southampton, UK. [37]Centro de Investigação em Saúde da Manhiça, Maputo, Moçambique. [38]Oxford Vaccine Group, Department of Paediatrics, University of Oxford, NIHR Oxford Biomedical Research Centre, Oxford, UK. [39]Faculty of Medicine and Health, University of Sydney, Sydney, New South Wales, Australia. [40]Institute of Environmental Science and Research Limited, Kenepuru Science Centre, Porirua, New Zealand. [41]Division of Pediatric Infectious Diseases, University of Alabama at Birmingham (UAB), Birmingham, AL, USA. [42]International Foundation against Infectious Diseases in Nigeria (IFAIN), Abuja, Nigeria. [43]Wesfarmers Centre of Vaccines and Infectious Diseases, Telethon Kids Institute, The University of Western Australia, Nedlands, Western Australia, Australia. [44]Infection and Immunity Unit, Papua New Guinea Institute of Medical Research, Goroka, Papua New Guinea. [45]Instituto de Medicina Tropical Alexander von Humboldt, Universidad Peruana Cayetano Heredia, Lima, Peru. [46]National Medicines Institute, Warsaw, Poland. [47]Hamad Medical Corporation, Doha, Qatar. [48]G. N. Gabrichevsky Research Institute for Epidemiology and Microbiology, Moscow, Russia. [49]Department for Public Health Microbiology, National Laboratory of Health, Environment and Food, Ljubljana, Slovenia. [50]Department of RDI Microbiology, Institut de Recerca Sant Joan de Deu, Hospital Sant Joan de Deu, Universitat Internacional de Catalunya and CIBER Epidemiology and Public Health (CIBERESP), Barcelona, Spain. [51]Faculty of Pharmacy, Siam University, Bangkok, Thailand. [52]Centre for Epidemic Preparedness and Response, London School of Hygiene and Tropical Medicine, London, UK. [53]Department of Infection Biology, Faculty of Infectious and Tropical Diseases, London School of Hygiene and Tropical Medicine, London, UK. [54]Medical Research Council Unit The Gambia at London School of Hygiene and Tropical Medicine, Fajara, Banjul, The Gambia. [55]Medical Affairs, Vaccines and Antivirals Pfizer, Paris, France. [56]Department of Para-Clinical Sciences, The University of the West Indies St Augustine Campus, St Augustine, Trinidad and Tobago. [57]Hacettepe University Faculty of Medicine, Department of Medical Microbiology, Ankara, Türkiye. [58]Department of Infectious Disease Epidemiology, Faculty of Epidemiology and Population Health, London School of Hygiene and Tropical Medicine, London, UK. [59]Centre for Genomic Pathogen Surveillance, Big Data Institute, Nuffield Department of Medicine, University of Oxford, Oxford, UK. [60]School of Public Health, Imperial College London, London, UK. [61]European Molecular Biology Laboratory, European Bioinformatics Institute, Wellcome Genome Campus, Hinxton, UK. [62]Department of Biostatistics, Faculty of Medicine, University of Oslo, Oslo, Norway. [63]Department of Epidemiology of Microbial Diseases, Yale School of Public Health, New Haven, CT, USA. [64]Institute of Microbiology and Infection, College of Medical and Dental Sciences, University of Birmingham, Birmingham, UK. [65]Pneumonia and Pandemic Preparedness, Global Health Division, Bill & Melinda Gates Foundation, Seattle, WA, USA. [66]Department of Pediatrics, Division of Pediatric Infectious Diseases, School of Medicine, University of Utah, Salt Lake City, UT, USA.

## Methods

### Data sources and processing

**Pneumococcal sequence data and metadata.** The genomes included in this study were collected as part of the Global Pneumococcal Sequencing project (GPS), which is a global genomic survey of *S. pneumoniae*[40]. The invasive disease isolates included here were collected by the National Institute for Communicable Disease in South Africa from 2000 to 2014 (ref. 22). In the initial phase of genome sequencing, approximately 300 invasive-disease isolates from each year (2005–2014) were selected with a specific target age breakdown (50% from <3 year olds, 25% from 3–5 year olds, 25% from >5 year olds). In the second phase of sequencing, approximately 200 disease isolates from 2000 to 2004 and approximately 100 invasive disease isolates from 2005 to 2010 from children <5 years old were randomly selected for sequencing[22] (Supplementary Table 2). The carriage isolates were collected in Soweto, Gauteng ($n = 736$; collection years 2010, 2012 and 2013) and Agincourt, Mpumalanga ($n = 1,114$; collection years 2009, 2011 and 2013) by the Wits Vaccines and Infectious Diseases Analytics Research Unit. A random sample of 400 carriage isolates from each year were chosen for genome sequencing (Supplementary Table 2). We included both carriage and invasive-disease isolates and conducted sensitivity analyses throughout to confirm the methods were robust to both carriage and invasive disease individually. Invasive disease is defined as the bacterium being isolated from a typically sterile site. The majority of total isolates were from children aged <5 years (75.0%); 7.6% were from individuals aged 5–20 years and only 17.5% were from adults (>20 years) (Supplementary Table 7). These were distributed across the before and after PCV periods. We additionally included similar GPSCs from the GPS database (for context within the global population) for the RR analysis. We utilized metadata that included collection year and month, residence province of the patient, age of the patient, sampling site and clinical manifestation. The range of sampling sites included nasopharyngeal swabs (for carriage), blood, pleural fluid, cerebrospinal fluid, peritoneal fluid, pus and other joint fluid (for invasive disease).

**Population data.** We estimated the population for each municipality ($n = 234$) across South Africa using the population-size estimates from LandScan 2017 (refs. 41,42) (Extended Data Fig. 11).

**Mobility data.** The mobility data used in this study were collected using Meta Data for Good Disaster maps from South Africa. These are initiated at the onset of a disaster—in this case, the SARS-CoV-2 pandemic—and track the geographical movement of Meta users[43]. We used the baseline (adjusting to 2 weeks before) human mobility for each month from January 2020 to July 2021 (refs. 43,44) to attain a mobility probability from and to each municipality. For each origin (home) municipality ($n = 234$), we determined the mean monthly number of Meta users that were in each destination municipality in South Africa. Location pairs with a value of zero were given a value of ten. We divided each cell by the total number of users across all destinations for that origin municipality. Each cell in the resultant original–destination matrix therefore represented the probability of being in each destination municipality given your home municipality. This is the probability of mobility from each municipality to each other municipality after 2019. Because we do not have mobility data from the exact years the genomes were sampled, we adjusted the diagonal of the mobility matrix using an estimated parameter. This allows people to stay more or less at home and mitigates the effect of non-year matched mobility data. We define the radius of gyration ($R_i$) for each municipality as follows:

$$R_i = \sqrt{\sum_i m_i d_i^2} \qquad (1)$$

Where $d_i$ is distance to region $i$, and $m_i$ is the probability of mobility to region $i$. The sum is across all municipalities ($n = 234$)[45] (Extended Data Fig. 11).

**Generation time distribution.** We used a simulation framework to estimate the overall generation time using the separate contributions of the carriage durations and the incubation period. This approach has previously been used for other pathogens[46].

We sampled 1,000 carriage durations from an exponential distribution with means that are inverse to the clearance rates estimated across serotypes in ref. 31 (clearance rate = 0.026 (95% CI = 0.025–0.028) episode per day) and in ref. 32 (clearance rate = 0.032 (95% CI = 0.030–0.034) episodes per day)[32].

To sample the day of transmission, we randomly sampled a time point between zero and the time of clearance for each individual. We then separately sampled an incubation period using a uniform distribution of between 1 and 5 days. To account for longer carriages resulting in more opportunities for transmission, we sampled from the distribution of generation, with the probability of sampling each individual weighted by the total carriage duration. The total generation time is then the sum of the duration to transmission and the incubation period.

We repeated these steps 10,000 times and estimated the mean and standard deviation of this final distribution assuming that the generation time follows a gamma distribution with a mean of 35 and a standard deviation of 35 (exponential distribution with a rate of 1/0.096). By comparing the histogram of the distribution with the gamma distribution, this seems a reasonable assumption (Supplementary Fig. 4).

**Sample culture and genome sequencing.** The pneumococcal isolates were selectively cultured on BD Trypticase soy agar II with 5% sheep blood (Beckton Dickinson) and incubated overnight at 37 °C in 5% $CO_2$. Genomic DNA was then manually extracted using a modified QIAamp DNA Mini kit (Qiagen) protocol. As part of GPS, pneumococcal isolates were whole-genome sequenced on an Illumina HiSeq platform to produce paired-end reads with an average of 100–125 bp in length, and data were deposited into the European Nucleotide Database. Whole-genome sequence data were processed as previously described[36,47].

**AMR.** We performed predictive antimicrobial susceptibility profiling using the CDC-AMR pipeline for three classes of antimicrobials: β-lactams (penicillin; encoded by the genes *pbp1A*, *pbp2B* and *pbp2X*)[48,49]; sulfonamides (co-trimoxazole; *folA* and *folP*); and macrolides (erythromycin and clindamycin; *ermB* and *mefA*)[50,51]. This was done for 6,798 randomly selected isolates[40].

**Constructing time-resolved phylogenetic trees.** We selected the GPSCs for which we had genomes from each of South Africa's nine provinces and for which we had a minimum of 50 sequences in total to build phylogenies, henceforth referred to as 'dominant GPSCs'. There were nine dominant GPSCs: GPSC1, GPSC2, GPSC5, GPSC10, GPSC13, GPSC14, GPSC17, GPSC68 and GPSC79 ($n = 2.575$). Assembly was performed using Wellcome Sanger Institute pathogen informatics automated pipelines and is freely available for download from GitHub under an open-source licence, GNU GPL 3 (ref. 52). For each sample, sequence reads were used to create multiple assemblies using VelvetOptimiser (v.2.2.5) and Velvet (v.1.2.10)[53]. An assembly improvement step was applied to the assembly with the best N50 and contigs scaffolded using SSPACE (v.2.0)[54], and sequence gaps were filled using GapFiller (v.1.11)[55]. Assembly quality control parameters included a minimum average sequencing depth of 20× and an assembly length of 1.9–2.3 Mb. Sequences with more than 15% heterozygous SNP sites were excluded.

We created reference genomes for each GPSC using ABACAS (v.1.3.1) to order the contigs from a representative of each GPSC mapped to *S. pneumoniae* (strain ATCC 700669/Spain 23F-1) (EMBL accession: FM211187)[56]. Any contigs that did not align were concatenated to the end. We multiply mapped all genomes from each dominant GPSC against these references, respectively, using a custom mapping, variant calling and local realignment around indels pipeline

(multiple_mappings_to_bam.py)[57] using bwa-MEM (v.0.7.17)[58] and samtools mpileup (v.1.6)[59]. The minimum base quality for a base to be considered was 50. The minimum mapping quality for a SNP to be called was 20, with a minimum of 8 reads matching the SNP. We built trees masking recombination regions using Gubbins (v.2.4.1)[60] with the hybrid model that uses FastTree for the first iteration and RAxML subsequently[61] and a GTR model. We converted branch length to time using BactDating (v.1.0) with a mixed gamma, relaxed clock model[62]. We compared concordance between BEAST (v.1.10.4)[63] with both strict and relaxed clocks, and a Bayesian skyline prior. As the results were concordant, we used BactDating owing to its shorter runtime (Supplementary Fig. 14).

## RR framework

We used a RR framework to investigate the risk of genetic similarity across geographical distance[64]. We compared the location (loc) and label ($G$) (that is, GPSC or genetic similarity) of pairs of sequences that were collected around the same time ($t$). This approach has been shown to be robust to substantial biases in timing and location of isolate collection[64]. We first constructed pair-wise matrices comparing every isolate to every other isolate ($n$ pairs = 6,910). In the numerator was the ratio of pairs that were the same GPSC, collected within a year of each other, from the same province, over the total number of pairs collected within a year of each other from the same province. The denominator was the ratio of pairs that were the same GPSC, collected within a year of each other, from distant provinces (>1,000 km apart) ($L_{ref}$), over the total number of pairs collected within a year of each other from distant provinces. Geographical distances were calculated based on the centroid coordinates of each province. To demonstrate the suitability of using centroid distances, we simulated a spatial transmission process for 1,000 separate chains in which at each generation, a daughter point is placed at a randomly located location 350 m in each of the $x$ and $y$ direction. This was repeated over 20 generations. We then identified the centroid of each case based on the closest coordinate rounded to the nearest kilometre. We then calculated the total distance covered for both the true distance and the centroid distances and found that the resulting distances were similar (Supplementary Fig. 15).

To quantify uncertainty, we used a bootstrapping approach whereby in each bootstrap iteration, we randomly sampled with replacement the isolates before recalculating the statistic. We report the 2.5 and 97.5 percentiles from the resulting distribution.

We also repeated the same analysis but used the time-resolved phylogenetic trees to interrogate pairs across increasing divergence times (breaking the GPSCs into higher resolution). For this, rather than matrices designating whether pairs were the same GPSC or different GPSCs, the divergence time between each pair was included. We only included the divergence times between like GPSCs (Fig. 2c–e).

$$\text{RR}_{\text{loc}}(g1, g2)$$
$$= \frac{\sum_{i=1}^{n} \sum_{j\neq i}^{n} I(\text{loc}_i = \text{loc}_j \cap t_{ij} \leq 1\,\text{year} \cap (G_{ij} > g_1, G_{ij} < g_2)) \Big/ \sum_{i=1}^{n} \sum_{j\neq i}^{n} I(\text{loc}_i = \text{loc}_j \cap t_{ij} \leq 1\,\text{year})}{\sum_{i=1}^{n} \sum_{j\neq i}^{n} I(L_{\text{ref}\,i} = L_{\text{ref}\,j} \cap t_{ij} \leq 1\,\text{year} \cap (G_{ij} > g_1, G_{ij} < g_2)) \Big/ \sum_{i=1}^{n} \sum_{j\neq i}^{n} I(L_{\text{ref}\,i} = L_{\text{ref}\,j} \cap t_{ij} \leq 1\,\text{year})} \quad (2)$$

We utilized the framework to compare a range of geographical distances, keeping the reference distance to pairs that were >1,000 km apart (Fig. 2).

To identify the divergence time at which pairs had an equal risk of being in the same province as distant provinces (time to homogenization across South Africa), we investigated the divergence time at which there was no increased risk of similarity within a province compared against distant provinces (RR = 1). We stratified distances in South Africa into groups of distances, including the 9 within provinces, 14 pairs that were <500 km apart, 16 pairs 500–1,000 km apart and 6 pairs of provinces >1,000 km apart. We repeated the framework across rolling 20-year time windows at 10-year intervals from 0 to 100 years ($g1,g2$)

(Fig. 2f). We repeated this for pairs for which one was from South Africa and the other was from another country in Africa (Supplementary Fig. 3A). We sampled 300 sequences from each province with replacement to compensate for biased sampling (Extended Data Fig. 2a). Furthermore, to incorporate phylogenetic uncertainty into the statistical framework, we sampled 100 individual phylogenies from the BactDating posterior. We report the 2.5 and 97.5 percentiles from the resulting distribution. We also repeated the analyses only including pairs isolated from patients with pneumococcal disease (Extended Data Fig. 2b).

## Probabilistic mobility model

**Overall strategy.** We extended a previously published mechanistic phylogeographic model[65] to estimate the mobility of the pneumococcus between pairs of municipalities at each transmission step. To infer the probable path of transmission between sequence pairs, we used the divergence time and the generation time distribution to estimate the number of transmission generations between pairs of sequences. Each generation is a possible transmission event and provides an opportunity for a mobility event.

The approach ultimately aims to estimate an origin destination matrix for a transmission step, whereby each cell represents the probability that the pneumococcus is now in location $j$ after one transmission step given it was previously in location $i$. As the phylogenetic trees combined with the generation time provide an estimate of how many transmission steps separate pairs of samples, we can use repeated matrix multiplication to integrate over all possible pathways linking two locations (see below for more details). We incorporated the probability of sampling at each geographical location, within each collection year, by GPSC to account for our observation process.

**Notation.** We follow notation per a previous study[65]. A pair of isolates, $C_a$ and $C_b$, with sequences, $\text{Seq}_a$ and $\text{Seq}_b$ included in a phylogeny are found in locations, $L_a$ and $L_b$, and the samples were taken in the years, $T_a$ and $T_b$. The inferred MRCA between $C_a$ and $C_b$ is time $T_m$ and located in $L_m$. The number of transmission generation from the MRCA to $C_a$ is $G_a$ and to $C_b$ is $G_b$.

## Model fitting

**Single transmission generation.** Considering a single transmission generation, the probability that persons $i$ and $j$ come into contact with each other given $i$ lives in location $a$ and $j$ lives in location $b$ can be written as follows:

$$P(\text{person } i \text{ and } j \text{ come into contact}|L_i = a, L_j = b)$$
$$= \sum_{k}^{N} P(V_i = k|L_i = a) \cdot P(V_j = k|L_j = b) \cdot B_k \quad (3)$$

Where $P(V_i = k|L_i = a)$ is the probability that individual $i$, whose home location is in $a$, visits location $k$ and $P(V_j = k|L_j = b)$ is the probability that individual $j$ whose home location is in $b$ visits location $k$, and $B_k$ is the location-specific probability of transmission for $k$.

At time $\tau$, one infector in $i$ in location $a$ is expected to transmit to this number of persons in location $b$:

$$E(\text{number of persons from } b \text{ infected at time } \tau|L_i = a)$$
$$= \sum_{k}^{N} P(V_i = k|L_i = a) \cdot P(V_j = k|L_j = b) \cdot B_k \cdot S_{b,\tau,\text{gpsc}} \quad (4)$$

Where $S_{b,\tau,\text{gpsc}}$ is the number of susceptible people in location $b$ at time $\tau$ with some lineage = gpsc.

The total number of people infected by the infector is:

$$E(\text{number of persons from all locations infected at time } \tau|L_i = a)$$
$$= \sum_{m}^{N} \sum_{k}^{N} P(V_i = k|L_i = a) \cdot P(V_j = k|L_j = m) \cdot B_k \cdot S_{b,\tau,\text{gpsc}} \quad (5)$$

Conditional on transmission occurring, the probability that the infectee's isolate is taken in location $b$ is:

$$\delta_{a,b,\tau,\mathrm{gpsc}} = P(L_j = b | L_i = a)$$

$$= \frac{\sum_k^N P(V_i = k | L_i = a) \cdot P(V_j = k | L_j = b) \cdot B_k \cdot S_{b,\tau,\mathrm{gpsc}}}{\sum_m^N \sum_k^N P(V_i = k | L_i = a) \cdot P(V_j = k | L_j = m) \cdot B_k \cdot S_{b,\tau,\mathrm{gpsc}}} \quad (6)$$

We then created an NXN transmission matrix, $\Delta_{\tau,\mathrm{gpsc},\mathrm{gen}=1}$, with $N$ being the total number of locations, containing the transmission probabilities, asymmetrically, between all pairs of locations at each point in time; $\delta_{a,b,\tau,\mathrm{gpsc}}$ is element $[a,b]$ of the matrix.

**Human mobility characterization.** We use Meta mobility data (Met-Mob), as described in the mobility data section above, to characterize mobility between the 234 municipalities of South Africa. We aggregated these to the province level ($n = 9$) to fit the model. Initially we extracted a 234 × 234 matrix that sets out the probability that an individual from municipality $a$ visits location $k$; again, where $k$ = any municipality.

MetMob comes from individuals using Meta; however, this may not be representative of the amount of time spent at home by those involved in pneumococcus transmission. The mean of the diagonal of the Meta Mobility matrix (234 × 234) is 0.989, implying, on average, 98.9% of Meta Users stay in their home municipality, $H_i$. To allow for individuals to spend more or less time at home than represented in the MetMob data, we incorporated a parameter to adjust the probability of being at home ($\theta$). We adjusted the probability of staying home using a standard logistic function and restricted it with bounds of −0.04 and 0.6 to facilitate exploration of a sensible space. The adjustment allowed by the bounds limits the range of movement $H_i$−0.6 and $H_i$+0.04.

The probability of a person remaining in location $a$ therefore becomes:

$$P(V_i = a | H_i = a)$$

$$= \mathrm{MetMob}[a, a] - \left( -0.04 + \left( \frac{(\exp(\theta))}{(1 + \exp(\theta))} \right) \cdot (0.6 + 0.04) \right) \quad (7)$$

Where a value of greater than 1.0 is obtained for a specific municipality, this is replaced by a value of 0.999.

The estimates thus far are mobility per day, but we were interested in mobility across the infectious period. Therefore, we adjusted the diagonal to account for mobility each day within the infectious period:

$$\mathrm{MetMob}[a, a] = 1 - \mathrm{MetMob}[a, a]^G \quad (8)$$

We rescaled the probabilities so that the sum of all mobility is equal to 1:

$$P(V_i = a | H_i = a, k \neq a) = \mathrm{MetMob}[a, k] / (1 - P(V_i = a | H_i = a)) \quad (9)$$

Where $\mathrm{MetMob}[a,k]$ considers all mobility probabilities from the Meta Mobility data.

The sum of the movements to South African municipalities is equal to 1, which assumes that the analysis contains all possible movements of both the pneumococcus and people and implying no external introductions. The outcome of this is that some mobility may be missed, especially around the country borders.

**Probability of the pneumococcus being in each location after $G$ transmission generations.** To determine the probability that location $k$ contains the home location after $G$ transmission generations, we used matrix multiplication, which integrates across all possible pathways connecting two locations.

$$\Delta_{\tau = T_G, \mathrm{gpsc}, \mathrm{gen}=G} = \prod_{r=2}^{G} \Delta_{\tau = t_{r-1}, \mathrm{gpsc}, \mathrm{gen}=r-1} \cdot \Delta_{\tau = t_r, \mathrm{gpsc}, \mathrm{gen}=1} \quad (10)$$

Where $t_r$ is the time of generation $G_r$.

**Probability of observing a pair of cases in two specific locations.** The probability that $C_A$ has home location $L_A$ and $C_B$ has home location $L_B$ is conditional on the sequences being observed in locations $L_A$ and $L_B$ at times $T_A$ and $T_B$. We assumed that the location of two cases, $L_i$ and $L_j$, is dependent on the location of their MRCA, $L_m$, and the number of transmission generations separating them from their MRCA, $G_A$ and $G_B$.

The observations processes across locations are independent of each other, and each transmission event is independent of other transmission events. The probability of observing (Obs) a case at $L_i$ at time $T_A$ is not dependent on the number of generations to, or location of, the MRCA. We considered discretized space of the nine provinces of South Africa, resulting in the following equation:

$$P(L_A, L_B | \mathrm{Obs}_{L_B, T_B}, \mathrm{Obs}_{L_B, T_B}, \mathrm{Seq}_A, \mathrm{Seq}_B, T_A, T_B)$$

$$= \frac{\sum_{L_m} \sum_{G_A} \sum_{G_B} P(\mathrm{Obs}_{L_A, T_A}) P(\mathrm{Obs}_{L_B T_B}) P(L_A | L_m, G_A) P(L_B | L_m, G_B)}{\sum_{L_i} \sum_{L_j} \sum_{L_m} \sum_{G_A} \sum_{G_B} P(\mathrm{Obs}_{L_i, T_A}) P(\mathrm{Obs}_{L_i T_B}) P(L_i | L_m, G_A)}$$

$$\frac{P(L_m) P(G_A, G_B | \mathrm{Seq}_A, \mathrm{Seq}_B, T_A, T_B)}{P(L_j | L_m, G_B) P(L_m) P(G_A, G_B | \mathrm{Seq}_A, \mathrm{Seq}_B, T_A, T_B)} \quad (11)$$

**Probability of G generations between MRCA and a sequenced isolate.** Under the previous equation, $P(G_A, G_B | \mathrm{Seq}_A, \mathrm{Seq}_B, T_A, T_B)$ represents the generation time distribution.

We can extract the joint probability of $C_A$ and $C_B$ being separated from the MRCA by $G_A$ and $G_B$ transmission generations, respectively, using the above-derived generation time distribution and the time-resolved phylogenetic trees.

Assuming the generation time is gamma distributed and all transmission events are independent, the sum of the gamma distribution is also gamma distributed. Additionally, we can extract the evolutionary times $E_A$ and $E_B$, separating $C_A$ and $C_B$ from the MRCA. As previously described[65], using equation (19), we can estimate the probability of $g$ transmission events over many trees, allowing us to incorporate phylogenetic including evolutionary parameters from the tree structure.

We determined the probability for the number of generations from MRCA for each isolate for 1–1,000 generations, using the generation time derived above.

**Location of the MRCA ($P(L_m)$).** We estimated the probability that, on average, an MRCA is in each of the nine provinces in South Africa. We estimated parameters for each of the eight provinces, setting Western Cape aside as a reference, and dividing by the total across all nine to ensure that the sum of the probabilities is 1. This again assumes no external introductions.

**Calculation of likelihood.** We calculated the likelihood using all pairs of available sequenced *S. pneumoniae* as previously described[65]. We accounted for the observation process by incorporating the probability of sampling in each location for isolates belonging to each GPSC annually.

**Likelihood equation.** We calculated the likelihood using all pairs of sequenced pneumococci as follows:

$$L \propto \prod_{\mathrm{province}=1}^{9} \prod_{i=1}^{n_{\mathrm{gpsc}}} \prod_{j \neq i} P(L_i, L_j | \mathrm{Obs}_{L_i, T_i}, \mathrm{Obs}_{L_j, T_j}, \mathrm{Seq}_i, \mathrm{Seq}_j, T_i, T_j) \quad (12)$$

Where $n_{\mathrm{gpsc}}$ are the number of sequences available from GPSC gpsc.

**Hamiltonian MCMC.** We used an MCMC approach to estimate our parameters using the package fmcmc (v.0.5-1) implemented in R[66]. We estimated nine parameters: a parameter that adjusts the probability of staying in the home municipality compared with Meta mobility data; and eight parameters capturing the relative probability that the MRCA of a pair of individuals was in each of the other eight provinces compared with the province of Western Cape (the reference).

We only used pairs of sequences that were separated by less than 10 years of evolutionary time between them. After 10 years, there are limited spatial signals remaining, as the bacterium has had many opportunities to move. This approach was used to make the model computationally tractable. To incorporate phylogenetic uncertainty, we repeatedly refit the model using 50 randomly selected phylogenies from the BactDating posterior. We report the 2.5 and 97.5 percentiles from the resulting distribution. In a sensitivity analysis, we showed that increasing the model to 15 years resulted in similar estimates (Extended Data Fig. 12a).

## Model performance

To assess the performance of our model, we used a simulation framework. We simulated 50,000 pairs of events, locations and the location of the MRCA between pairs, whereby the probability of mobility between each pair of locations was determined by the Meta mobility matrix adjusted by a parameter of known value of −2. We tested our ability to recapture this input parameter. To incorporate the biased observation process, we downsampled the simulated data based on the by-province sampling probabilities from our true data.

We used the downsampled data to fit our model with 20,000 steps of a MCMC with a jump step of 0.08 in 3 chains. We were able to recapture the downsampled data utilizing the human mobility framework and the estimated parameter for the probability of staying at home, accounting for the sampling probability per province. We then utilized the same human mobility framework and the estimated parameter, excluding the sampling probability, and were able to recapture the complete simulated data, including the input parameter (Extended Data Fig. 5a,b).

We repeated these simulations but downsampled with various biases. We determined how well the model performed when we only sampled two provinces. We also tested how far off our estimates would be if our generation time estimate was 50% higher or 50% lower than we had estimated given the Kenyan and Gambian data[31,32] (Extended Data Fig. 5c,d).

**Model sensitivity analyses.** We estimated our parameters only to include isolates from patients with invasive pneumococcal disease (Supplementary Fig. 1B).

To test the impact of a range of generation times on the parameter estimates, we also re-ran our MCMC with generation times of 15, 35 (as included in the model) and 55 days. To quantify uncertainty, we sampled posterior parameters, reporting the 2.5 and 97% percentiles (Extended Data Fig. 12b).

We confirmed that the chains converged for each of the nine parameters estimated (Supplementary Fig. 16).

## Probability guided transmission simulations

**Person-to-person transmission chains.** We simulated person-to-person transmission chains seeded in a starting municipality weighted by the population size. We determined the RR of being in a specific municipality after 10 transmission generations (approximately 1 year) across 500,000 simulations.

We fixed the starting municipality to be rural (population density <50 km$^{-2}$) or urban (population density >500 km$^{-2}$) and repeated the above simulation. For 10,000 sequential simulations, we counted the number of unique municipalities affected and distance travelled at each transmission interval weighting by population size. We then determined the number of municipalities travelled to across all transmission chains (Extended Data Fig. 3).

**Branching epidemic.** We simulated a branching epidemic in which we drew the number of transmission events seeded by each event from a Poisson distribution around an effective reproductive number ($R_{eff}$) of 1 and amplitude of 0.15 over 100 iterations. We determined the mean distance from the starting municipality after 60 generations (5.8 years with a 35-day generation time) and the number of municipalities visited over that time. We calculated the uncertainty at the 95% CI of a normal distribution.

**Gravity model.** We compared the performance of the model that used Meta data with the performance of a simple gravity model whereby the probability of mobility is determined by the distance and human population size of the municipalities. We calculated the probability of mobility between locations $i$ and $j$ (GravMob$_{i,j}$) to be the log of the destination population size (popsize$_j$) raised to parameter $\beta$ divided by the distance between locations, loc$_i$ and loc$_j$ (dist$_{i,j}$) raised to parameter $\gamma$.

$$\text{GravMob}_{i,j} = \frac{\log(\text{popsize}_j^{\beta})}{\text{dist}_{i,j}^{\gamma}} \tag{13}$$

We also tested a model including only distance and estimating an exponent adjustment parameter $\gamma$.

We calculated the DIC comparing the three models and found that the Meta mobility model (DIC = 12,290.97) was the best-performing model[67] when compared with the gravity model (DIC = 12,294.34). Both of these models performed better than distance alone (DIC = 12,424.32) (Extended Data Fig. 4).

All statistical analysis for the RR framework and the mobility model was performed in R (v.3.6.2)[68].

## Population-level fitness model

**Overall strategy.** We developed logistic growth models to fit the changing prevalence of different serotypes and assess the impact of vaccine implementation, utilizing a method that has previously been implemented for the endemic bacterium *Bordetella pertussis*[69].

**Vaccine-type model.** We first binned serotypes into three groups: those not included in the vaccine (NVTs), serotypes included in PCV7 (4, 6B, 9V, 14, 18C, 19F and 23F), and additional serotypes included in PCV13 (1, 3, 5, 6A, 7F and 19A). We used the full data for this analysis. We computed the relative abundance $f_{i,\text{ref}}$ of each group of serotypes, $i$, compared with a reference type, ref.

$$f_{i,\text{ref}}(t) = \frac{N_i(t)}{N_i(t) + N_{\text{ref}}(t)} \tag{14}$$

We chose to use NVTs as the reference. Varying the reference did not affect the model, as long as the reference samples span all years. We then used a simple logistic model, assuming a constant growth rate to capture the evolution of this abundance, at each time $t$.

$$f_{i,\text{ref}}(t) = \frac{1}{1 + \left(\frac{1 - f_{i,\text{ref},0}}{f_{i,\text{ref},0}}\right)\exp(-r_{i,\text{ref}} \cdot t)} \tag{15}$$

where $r_{i,\text{ref}}$ is the growth rate of that abundance shared across all provinces, and $f_{i,\text{ref},0}$ is the initial relative abundance of the serotype group $i$ with respect to a chosen ref.

To control for the varying presence of all circulating serotypes through time, we present fitness as the average relative growth rate, $\bar{r}_i$, for each group with respect to a randomly selected group in the population:

$$\bar{r}_i = \sum_{j \neq i}^{n} \overline{f}_j \, (r_{i,\mathrm{ref}} - r_{j,\mathrm{ref}}) \qquad (16)$$

where $n$ is the number of groups, and $\overline{f}_j$ is the average absolute frequency of the group in the period of time considered.

This average relative growth rate, $\bar{r}_i$, can be identified as the selection rate coefficient of the group in the population considered[69]. The selection rate coefficient is a direct measure of the fitness advantage of emerging variants and is one of the best indicators as to whether a strain will increase in frequency during an outbreak[70,71].

We can further multiply the selection rate by the mean generation time (35 days) to obtain the selection coefficient per generation, which is the relative fitness advantage per transmission generation.

In the Article, we also present estimates of the relative fitness advantage of groups in particular time frames. We used the same computation as for the average relative growth rate, but tailored it to specific references. For example, to compute the relative fitness advantage of NVTs compared with VTs in the PCV era $\Delta\overline{r}_{\mathrm{NVT,VT,after\,PCV}}$, we computed:

$$\Delta\overline{r}_{\mathrm{NVT,VT,after\,PCV}} = \frac{1}{\overline{f}_{\mathrm{PCV7}} + \overline{f}_{\mathrm{PCV13}}} \sum_{i \in (\mathrm{PCV7,PCV13})} \overline{f}_i \, (\overline{r}_{\mathrm{NVT,after\,PCV}} - \overline{r}_{i,\mathrm{after\,PCV}}) \qquad (17)$$

To fit the model, we used Hamiltonian Monte Carlo as implemented in R-Stan[72], with stan (v.2.26.1). We used a Poisson likelihood to fit the observed proportion of sequences that were of each category and the total number of isolates in that year as an offset. The model estimated the proportion of isolates that were of each type at the start of the dataset and the fitness parameters.

**Exploration of vaccine effect.** To investigate whether the serotypes fitness changed across after implementation of PCV7 and PCV13, we tested a range of models. We considered a model without any shift of fitness, a model with a single shift in 2009 and a model with two growth rates represented by a shift in fitness in both 2009 and 2011. The 2009 shift pertains only to those serotypes included in PCV7 and the 2011 shift pertains to those additional 6 serotypes in PCV13. Model comparison was done using the Watanabe–Akaike information criterion (WAIC) implemented in the loo package[73]. Furthermore, we tested for a potential delay between implementation of PCVs and the change in fitness. Model comparison is presented in Extended Data Fig. 7.

The model performed best with an initial fitness switch in 2009 (the initial year of vaccine implementation) and another fitness switch in 2011 (the year of PCV13 implementation). We used this model in the main text.

**Estimation of individual serotype fitness.** Using the same framework, we also estimated the growth rate per serotype to capture whether the individual dynamics were concordant with, or deviated from, what was expected according to their respective group (NVT, PCV7 or PCV13) (Fig. 4a–c). The reference strain in this analysis was set to serotype 13, which is a NVT and for which samples span all years.

**Exploration of predicted GPSC dynamics based on their serotype composition.** We explored whether using the proportion of isolates that were VT versus NVT within each GPSC at the start of the study period and our fitness estimates could explain the subsequent dynamics of individual GPSCs.

To predict the dynamics of individual GPSCs based on their serotype composition, we used the same framework as described above, applied to each GPSC serotype group in our dataset. As the number of such groups is large ($n$ = 340 GPSC serotype groups), we restricted the analysis to the GPSCs present at a minimum prevalence of 1%. This led to 26 GPSCs being considered for this analysis (representing 74.7% of the dataset), split in a total of 101 GPSC serotype groups. We then modelled the dynamics of each GPSC serotype group, estimating one

starting frequency per group and using the previously estimated fitness parameters, either by VT or serotype. We also considered a model with no fitness parameters (relative fitness = 1). To assess the model performances, we computed the Akaike information criterion (AIC)[74] using the average likelihood of each model and the number of parameters used in each model. We also computed the predicted GPSC dynamics over time for each model by summing all the predicted GPSC serotype group dynamics for each GPSC. As a measure of goodness of fit we used the coefficient of determination of the observed versus predicted GPSC proportions each year:

$$R^2 = 1 - \frac{\sum_i (\text{observed proportion} - \text{predicted proportion})^2}{\sum_i (\text{observed proportion} - \text{mean observed proportion})^2} \qquad (18)$$

$R^2$ is also the proportion of variation explained by the variables considered in each model.

Our model estimated a constant fitness for each group considered (VTs or individual serotypes), assuming that if a group has the highest fitness, it will eventually replace the other groups. This assumption is meaningful for VTs and serotypes, as some are directly targeted by the vaccines, a strong selective force in the population. However, the fitness of each GPSC cannot be modelled with this simple assumption as it has been shown that their fitness is inherently multifactorial, as described in the NFDS model[24].

**AMR.** We then used the model to capture the decreasing proportion of penicillin resistance in the population. To do this, we incorporated the dynamics of VTs (PCV7 and PCV13) and NVTs in the population and the respective proportion of penicillin resistance within them. To keep the model tractable, we did not differentiate here between serotypes included in PCV7 or PCV13, instead we group them into VTs. We use the PCV7 implementation (2009) as the year of fitness shift. This parametrization marginally differed from the best model (next best) (Extended Data Fig. 7) and enabled us to keep the number of categories tractable. We also compared the single switch model to a model with no change in fitness at the time of vaccine implementation. This approach showed that the model with a switch was superior (Supplementary Fig. 17).

We used the same approach described above to model the proportion of VTs, $f_{\mathrm{VT}}$, and NVTs, $f_{\mathrm{NVT}}$. We then modelled the proportion of strains that were and were not penicillin resistant within each group, either VT, $f_{\mathrm{AMR|VT}}$ or NVT, $f_{\mathrm{AMR|NVT}}$. We estimated the fitness of resistance in each.

We then fit the model to all four groups, $f_{\mathrm{VX,AMR}}$ (resistant NVT, susceptible NVT, resistant VT and susceptible VT).

$$f_{\mathrm{VX,AMR}}(t) = f_{\mathrm{VX}}(t) \cdot f_{\mathrm{AMR|VX}}(t) \qquad (19)$$

Where $f_{\mathrm{VX}}$ is the proportion of either VT or NVT in the population and $f_{\mathrm{AMR|VX}}$ is the proportion of penicillin resistance within each of those groups. We then derived the proportion of penicillin resistance overall in the population by summing the proportion of VT resistant, $f_{\mathrm{VT,AMR}}$, and NVT resistant, $f_{\mathrm{NVT,AMR}}$, strains.

**Estimating effect of fitness on migration.** We included the calculated average relative growth rates in the simulated branching epidemic to calculate the mean number of municipalities visited, distance travelled and probability of being in the home municipality over 5 years. We report uncertainty at one standard deviation from the mean. We repeated this incorporating the post-PCV selection coefficients for NVTs and VTs and estimated the relative increase in the number of municipalities visited for NVT serotypes compared with PCV serotypes after 2 years of transmission.

**Fitness model carrying capacity sensitivity.** Our fitness model assumed constant fitness over time. This specifically means that if a

serotype is found to be fitter than the rest of the population, we expect it to replace the whole population after some time. However, the currently best-supported model proposes that NFDS drives pneumococcus population dynamics[24], whereby lower frequency genes become more fit.

To test the effect of our assumption on our fitness estimates, we performed a sensitivity analysis. We introduced minimum and maximum carrying capacities in our logistic model. In equation (20), we let the relative frequency $f_{i,\text{ref}}$ go to a maximum of $K_{\max}$ and in equation (21), a minimum of $K_{\min}$:

$$f_{i,\text{ref}}(t) = \frac{K_{\max}}{1 + \left(\frac{K_{\max} - f_{i,\text{ref},0}}{f_{i,\text{ref},0}}\right)\exp(-r_{i,\text{ref}} \cdot t)}, \text{ if } r_{i,\text{ref}} > 0 \quad (20)$$

$$f_{i,\text{ref}}(t) = 1 - \frac{1 - K_{\min}}{1 + \left(\frac{(1 - K_{\min}) - (1 - f_{i,\text{ref},0})}{(1 - f_{i,\text{ref},0})}\right)\exp(r_{i,\text{ref}} \cdot t)}, \text{ if } r_{i,\text{ref}} < 0 \quad (21)$$

with $K_{\min} \leq K_{\max}$, and $K_{\min} \in [0, 1]$, $K_{\max} \in [0, 1]$.

We considered a range of values for $K_{\min}$ (0.1, 0.05 and 0) and $K_{\max}$ (0.9, 0.95 and 1) (Supplementary Fig. 20). We tested the effect of this carrying capacity has on both the vaccine status model (Supplementary Fig. 18) and the AMR and vaccine status model (Supplementary Fig. 19). In each case, we compared the fitness estimates obtained. We found that the fitness estimates were robust to variable carrying capacities across those tested, which implied that our assumption does not affect this framework. However, it is important to note that this model is robust to set variable carrying capacities while being unable to estimate them, which is in contrast to NFDS. Our model assumes that there is a population replacement up to the carrying capacity and does not allow for fine-scale estimates of an equilibrium. This fitness model can be used as a quantitative descriptor of the growth of distinct groups (that is, penicillin resistance or VT serotypes in the context of a perturbation) but cannot describe the complex underlying fitness effects of changing frequencies of genes under NFDS[24].

All statistical analysis for the fitness model was performed in R (v.4.0.5)[68].

## Ethics

The study was coordinated by the GPS (https://www.pneumogen.net/gps/), whose activities are approved by respective ethics committee in-country. Isolates sequenced in this study come from the National Institute for Communicable Diseases and the University of Witwatersrand carriage studies as previously published. These isolates were all collected as part of existing public health surveillance approved protocols in each country. No personally identifiable information was used as part of this study.

## Reporting summary

Further information on research design is available in the Nature Portfolio Reporting Summary linked to this article.

## Data availability

All data and code for figures and analysis are accessible at GitHub (https://github.com/sophbel/geomig_evo_pneumo). All whole-genome sequences were deposited into the European Nucleotide Database and accession numbers are available in the GitHub repository and on FigShare (https://doi.org/10.6084/m9.figshare.24219214)[75]. Associated metadata are available from the Microreact webserver (https://microreact.org/project/7wqgd2gbBBEeBLLPKonbaT-belman2024southafricapneumococcus), as well as from the GitHub repository and Global Pneumococcal Sequencing Project Monocle database (https://data.monocle.sanger.ac.uk/).

## Code availability

All code and scripts for analysis and figures are available from GitHub (https://github.com/sophbel/geomig_evo_pneumo).

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

**Acknowledgements** This work was supported by the Wellcome Trust (grant number WT QQ2016-2021, reference 206194, to S.B., S.L. and S.D.B.); the Bill & Melinda Gates Foundation (under Investment ID INV-003570 to S.A.M., A.v.G., M.d.P., S.D. and S.N.); the NIH (grant number R01AI160780 to H.S.); and the European Research Council (grant number 804744 to

H.S. and N.L.). For the purpose of Open Access, the author has applied a CC BY public copyright licence to any Author Accepted Manuscript version arising from this submission. We would like to thank all Global Pneumococcal Sequencing project partners; and M. O'Driscoll, S. Farr, and V. Carr for code review.

**Author contributions** S.B., H.S. and S.D.B. conceived the study. S.N., S.D., A.v.G., S.A.M., L.M., M.d.P. and S.D.B. collected and contributed the genomes from South Africa, and the GPS Consortium contributed other global genomes. S.B. collated the genomes and ran bioinformatics pipelines. N.L. developed the fitness framework. S.B., N.L., H.S. and S.D.B. performed the analyses. S.B., A.v.G., M.d.P., S.D., S.A.M., S.W.L., S.D.B. and H.S. discussed results and helped to contextualize them within a South African context. S.B., H.S. and S.D.B. wrote the first draft of the manuscript. All authors reviewed the first draft of the manuscript with special attention by S.B., N.L., S.N., S.D., M.d.P., S.W.L., L.M., S.A.M., A.v.G., S.D.B., H.S., A.P., C.M.A., K.K., J.L., N.C., D.L., R.K., R.D., S.M., B.V. S.B., N.L., H.S. and S.D.B., discussed the results and contributed to manuscript revisions. Funding was acquired by S.D.B., H.S. and S.B.

**Competing interests** The authors declare no competing interests.

**Additional information**
**Correspondence and requests for materials** should be addressed to Sophie Belman.

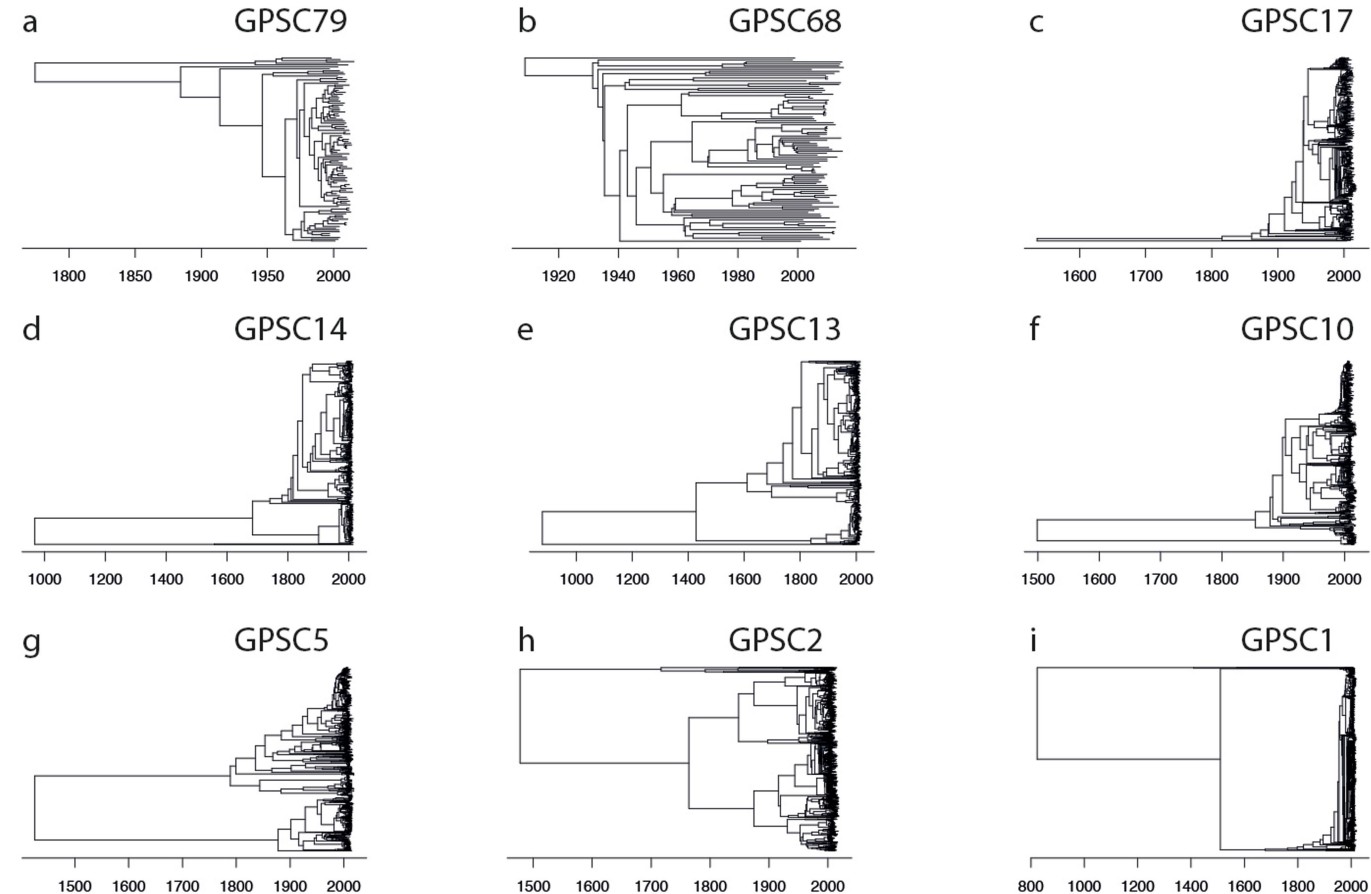

**Extended Data Fig. 1 | Time Resolved Trees for Dominant GPSCs.** Trees are recombination masked, aligned to a reference for each GPSC. Time resolution was performed using BactDating. The dates are along the x-axis. (a) GPSC79, N = 102 (b) GPSC68, N = 97 (c) GPSC17, N = 531 (d) GPSC14, N = 521 (e) GPSC13, N = 611 (f) GPSC10, N = 718 (g) GPSC5, N = 841 (h) GPSC2, N = 1430 (i) GPSC1, N = 1943.

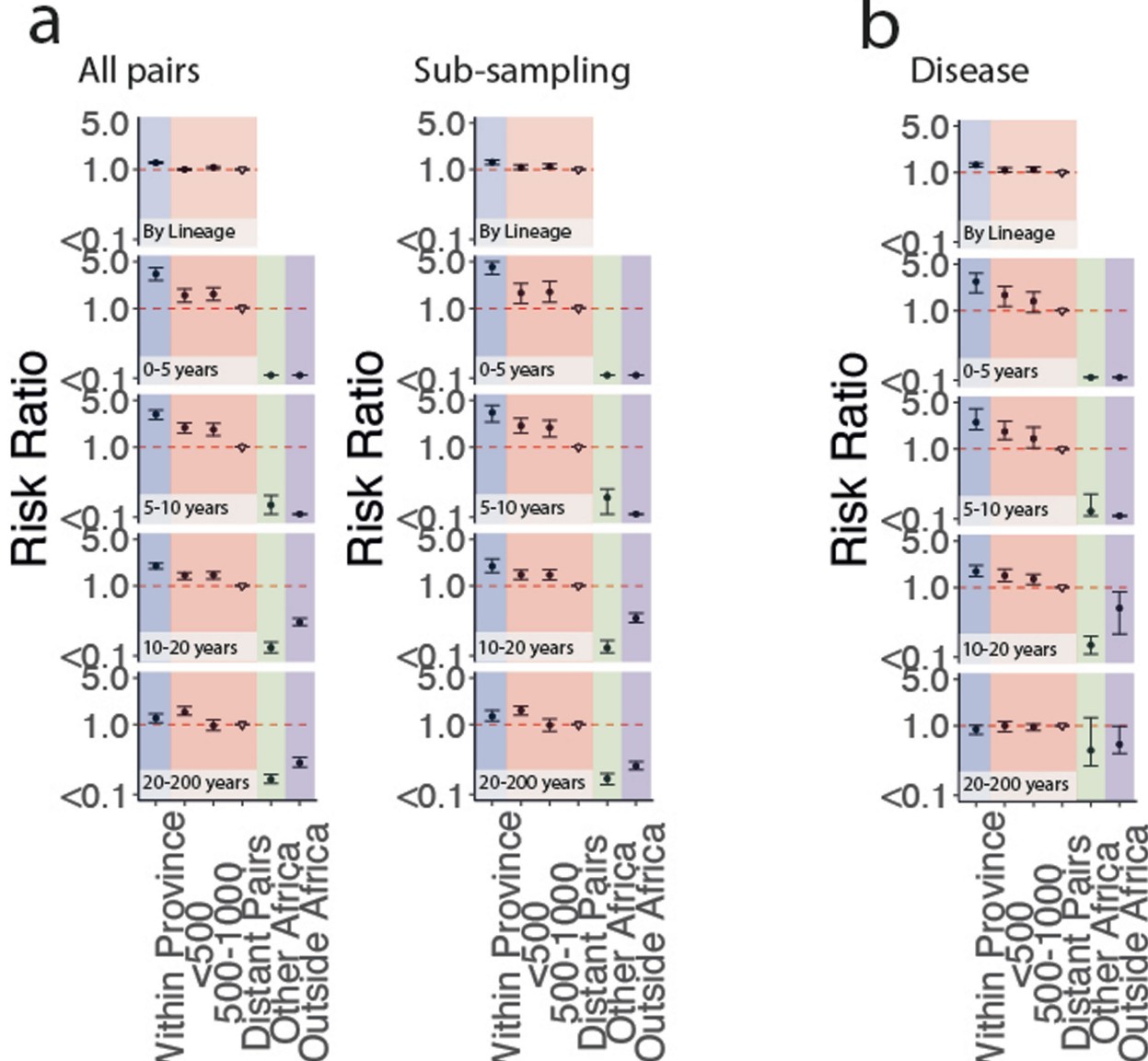

**Extended Data Fig. 2 | Risk ratio framework sensitivity analyses to determine geographic structure when sub-sampling and only including disease isolates.** Risk ratio of being the same GPSC within a province (blue), between different provinces over increasing distance (red), compared to geographically distant pairs (>1000 km) (reference) (top) and the risk ratio of having a tMRCA 0–5, 5–10, 10–20, or 20–200 years ago within South African provinces (blue), across larger distances within South Africa (red), from South Africa to other countries in Africa (green), and from South Africa to countries outside of Africa (purple) (South Africa; N = 6910). (a) Including all isolates from each province (left) compared with sub-sampling to 300 with replacement to compensate for biased sampling in each province. (b) Including only isolates sampled from patients with pneumococcal disease. All plots use a reference of pairs which are from distant provinces in South Africa (open triangle). Error bars represent 2.5 to 97.5 CIs.

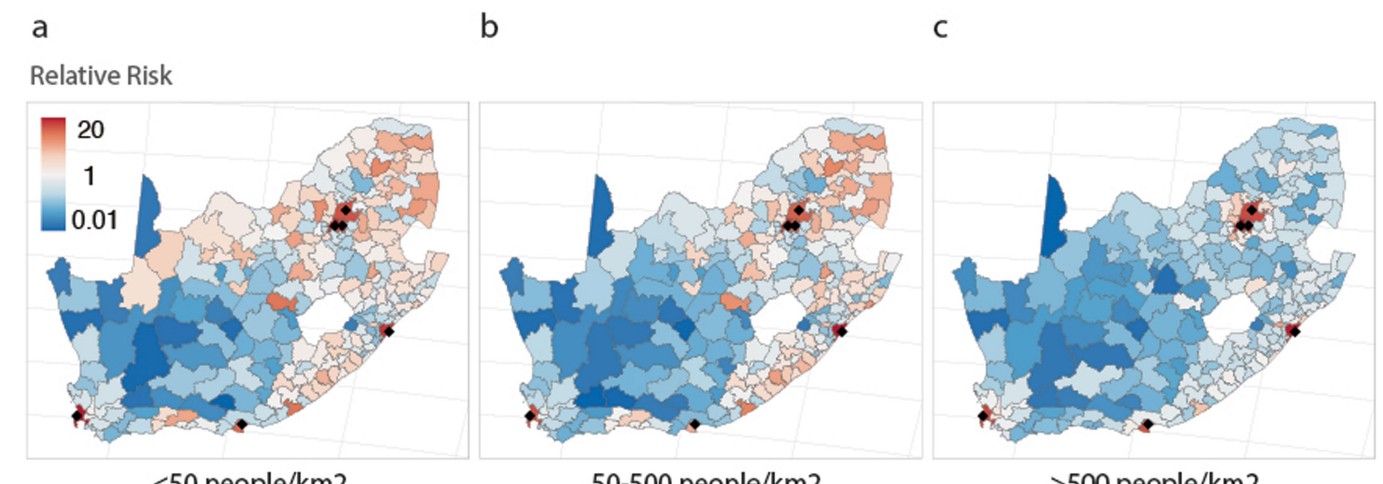

**Extended Data Fig. 3 | Relative risk of a pneumococcal strain being in each of municipality after 1 year of transmission.** Sequential transmission chains starting in municipalities with (a) <50 people/km2, (b) 50–500 people/km2, or (c) >500 people/km2 across 100,000 samples.

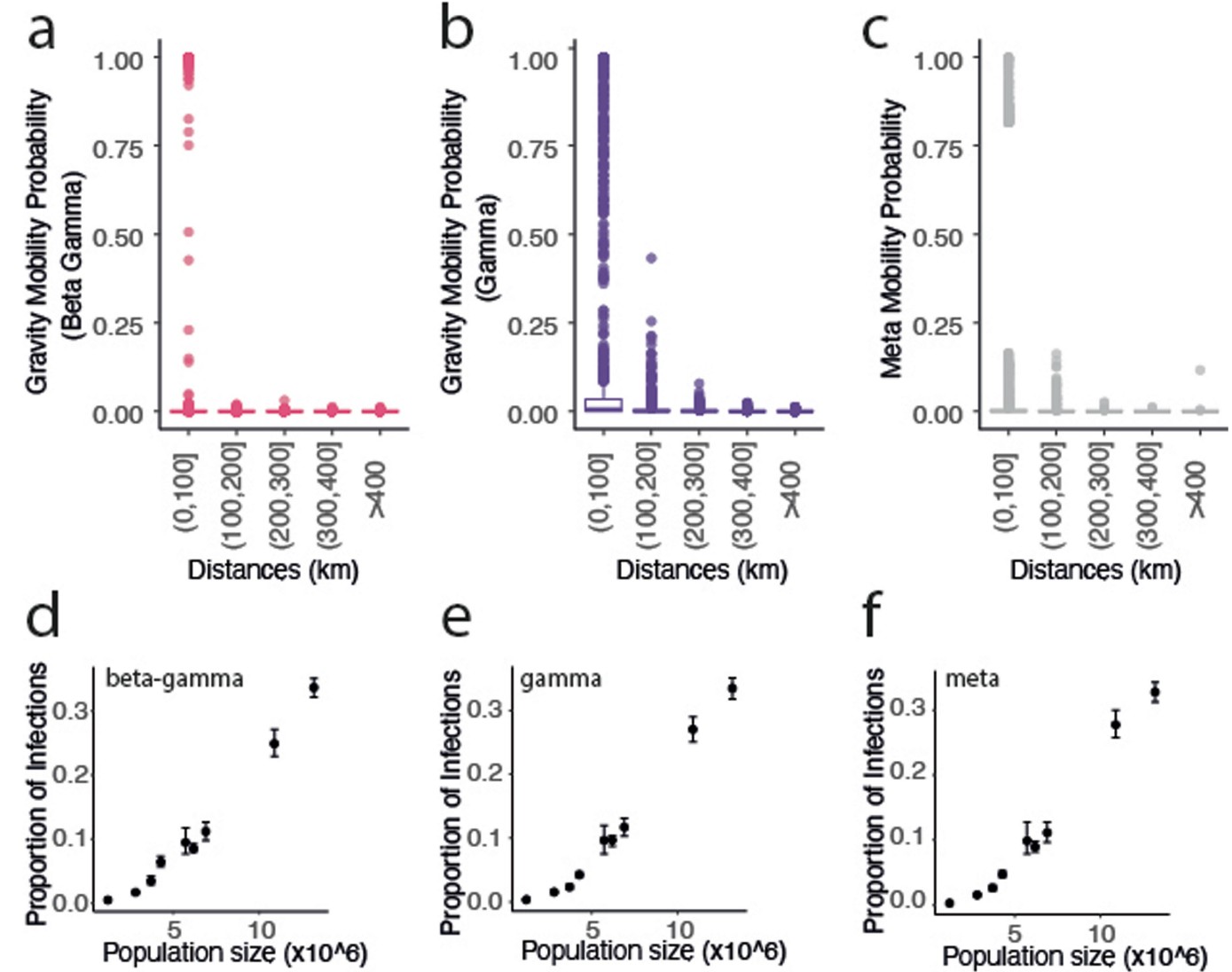

**Extended Data Fig. 4 | Estimated mobility and proportion of infections.** (a) a gravity model with two parameters adjusting the destination population size (beta) and the distance between locations (gamma) (pink), (b) a distance model whereby the probability of mobility is a function of the distance between locations adjusted with parameter gamma, (c) is the Meta mobility data between locations with a parameter adjusting the probability of staying in the home location (main model) across distances. (d-f) are the estimated proportion of infections by each of these models compared to the population size.

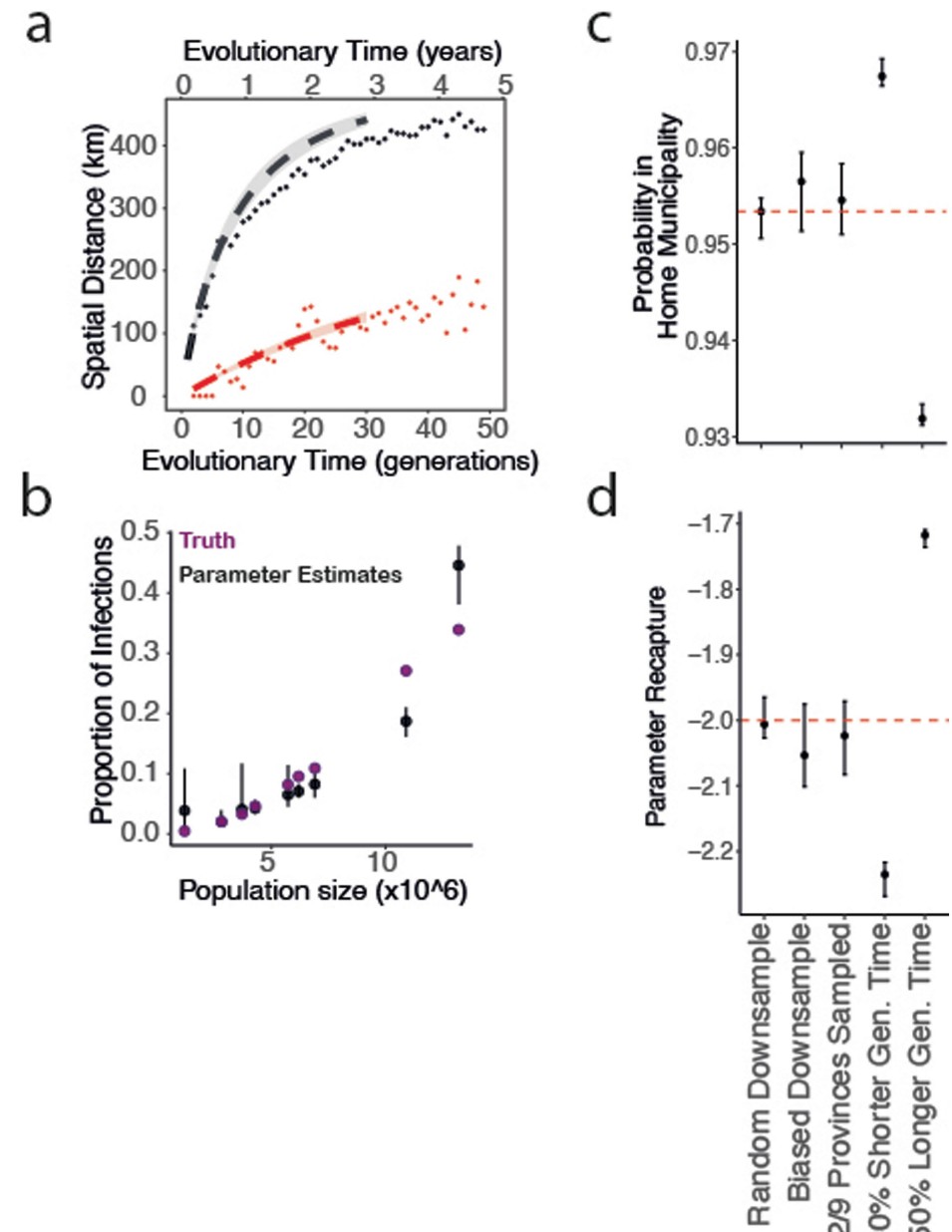

**Extended Data Fig. 5 | Replicated simulations for model performance testing.** Testing model performance using replicated simulations. (a) Simulated the total epidemic (black dots), biased down-sampled data as per true proportion per province (red dots) ("Biased Down-sample" in c and d), model fit to down-sampled data (red line), removing the sampling probability the model recapturing the true epidemic (black dot) (b) Population size (x-axis) compared to the proportion of infections from the down-sampled data (black) compared to the truth from the overall simulated epidemic (purple). (c) The probability of being in the home municipality and (d) the recaptured parameter after inputting a parameter of −2 to adjust the diagonal of the mobility matrix after one transmission generation. Both c and d include values from left to right for sampling as per the true data proportions in each province (6.5% of total infections), down-sampling to fit on only 2 of the 9 provinces, and if our generation time estimate is 50% smaller than the truth, exactly right, or 50% larger. Error bars represent 2.5 to 97.5 percentiles.

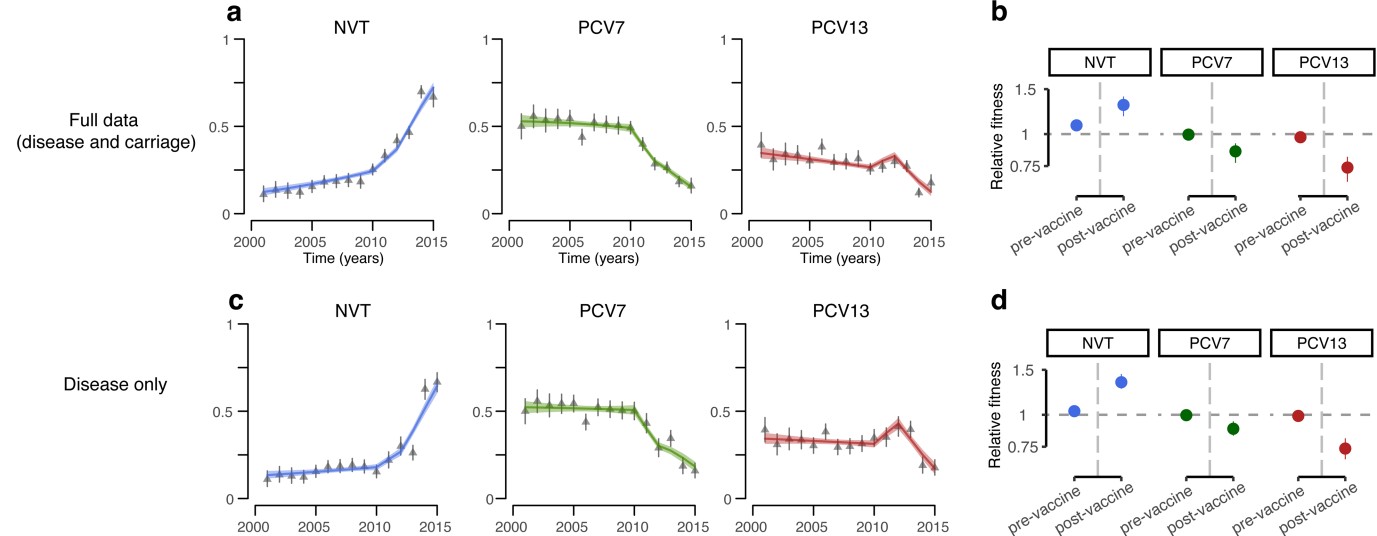

**Extended Data Fig. 6 | Comparison of fitness model results with full data or disease only data.** (a-b) Results with the full data. (c-d) Results with the disease-only data. a and c present the model fits for the proportion of serotypes from non-vaccine type (NVT), PCV7 types, and additional PCV13 types not included in PCV7 from the years 2000 to 2014 in this study. Points represent data and line represent the model fit. b and d present the relative fitness estimates for all three groups of serotypes in each era. Pre-vaccine era is prior to 2009 for NVTs, prior to 2009 for PCV7 and prior to 2011 for PCV13. Post-vaccine era is post-2009 for NVTs, post-2009 for PCV7 and post-2011 for PCV13. Error bars represent 2.5 and 97.5 percentiles.

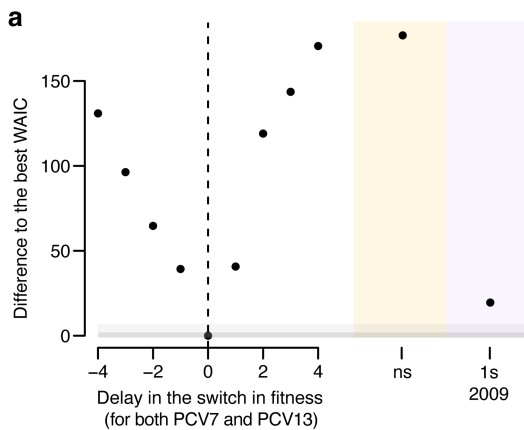

**a**

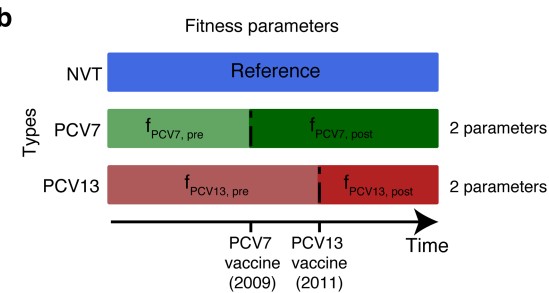

**b**

**Extended Data Fig. 7 | Fitness growth model testing year of switch and schematic.** (a) Testing year of fitness switch for the logistic growth -fitness model. Adjusting the year of the fitness switch in the model fitting to vaccine status. The difference to the best WAIC (2009 [PCV7 implementation] & 2011 [PCV13 implementation]) is on the y-axis where the year of fitness switch relative to 2009 & 2011 is on the x-axis. Further we test no fitness switch (ns; yellow) and the impact of including one fitness switch in 2009 (purple). The dark gray box highlights equivalent models (ΔWAIC ≤2) and light gray box highlights similar models (ΔWAIC ≤ 7). (b) Schematic denoting the fitness growth model parameterisation which accounts for the specific timing of the PCV impacting each group of serotypes (NVT in blue; PCV7 in green; PCV13 in red).

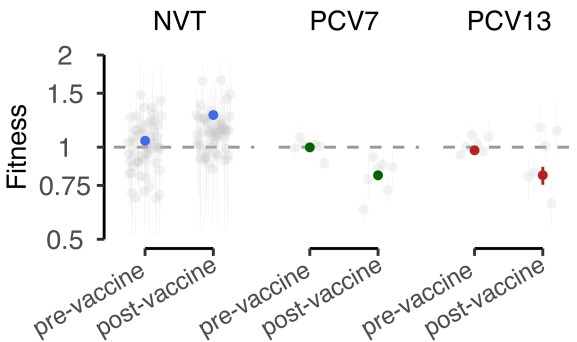

**Extended Data Fig. 8 | Serotype fitness estimates.** Fitness estimates pre- and post-PCV (y-axis) for each serotype (grey), superimposed by group including NVT (blue), PCV7 (green), and PCV13 (red). Pre-vaccine and post-vaccine refer to pre- and post- 2009 and 2011 for PCV7 and PCV13 respectively. Individual serotype fitness estimates can be found in Fig. S9.

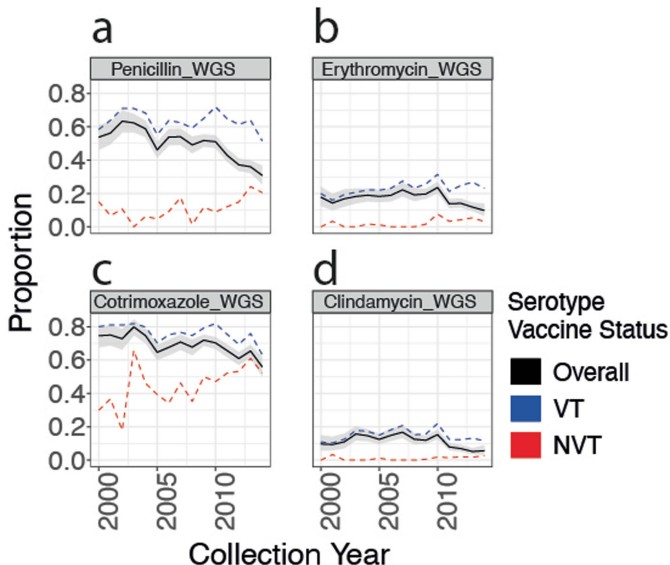

**Extended Data Fig. 9 | Antimicrobial resistance summary.** The proportional trends in antimicrobial resistance overall (black) and within Vaccine type [VT] (in blue) and Non-Vaccine Type [NVT] (in red) serotypes for in-silico predicted (a) penicillin (b) erythromycin (c) co-trimoxazole and (d) clindamycin.

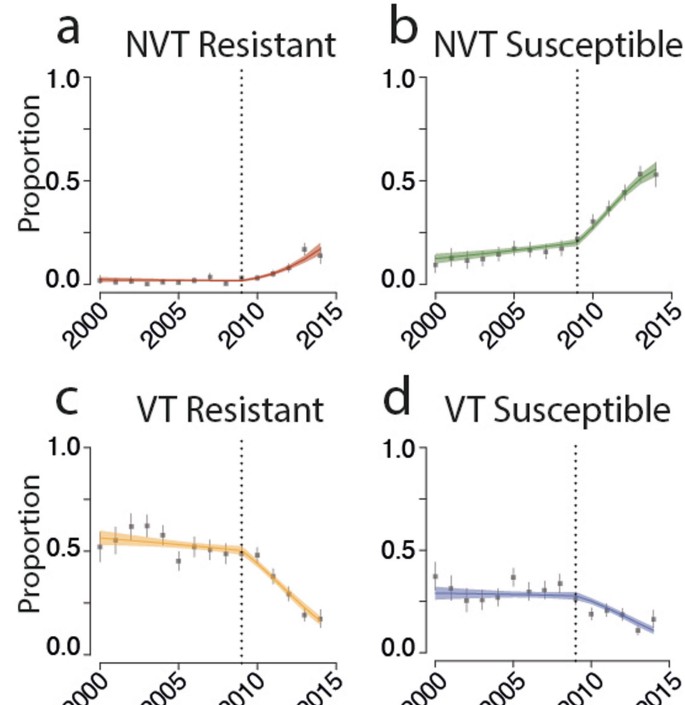

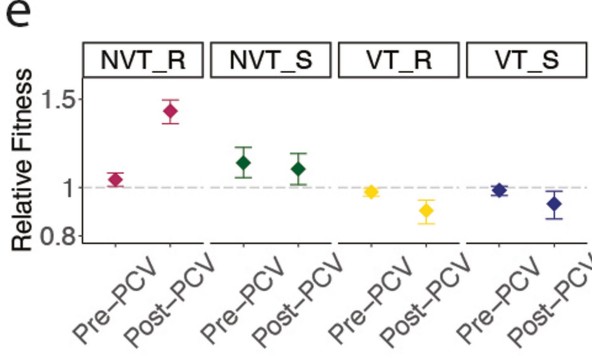

**Extended Data Fig. 10 | Data fits for model accounting for proportions and fitness over time in four groups.** (a) NVT-penicillin resistant (red), (b) NVT-penicillin susceptible (green), (c) VT-penicillin resistant (yellow) and (d) VT-penicillin susceptible (blue). The dashed lines indicate the year of PCV7 implementation and fitness switch model implemented. (e) Fitness estimates pre-PCV and post-PCV for each group and colored accordingly. e is on a log scale. This model uses a shift in fitness in 2009. Error bars represent 2.5 to 97.5 percentiles.

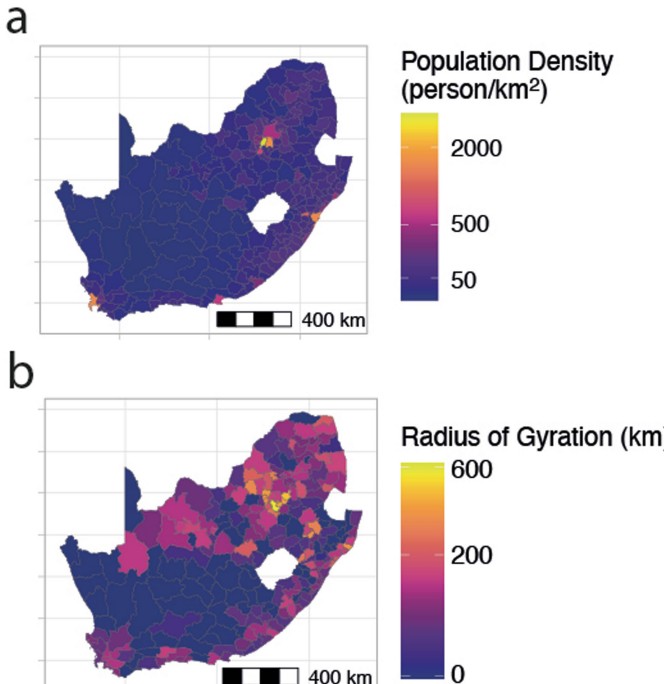

**Extended Data Fig. 11 | Data descriptions across the 234 municipalities of South Africa.** (a) Population density as estimated given the area of each municipality and the populations estimated by LandScan and (b) the radius of gyration for each municipality given the distance between each municipality at the centroid weighted by the human mobility data from Meta Data for Good.

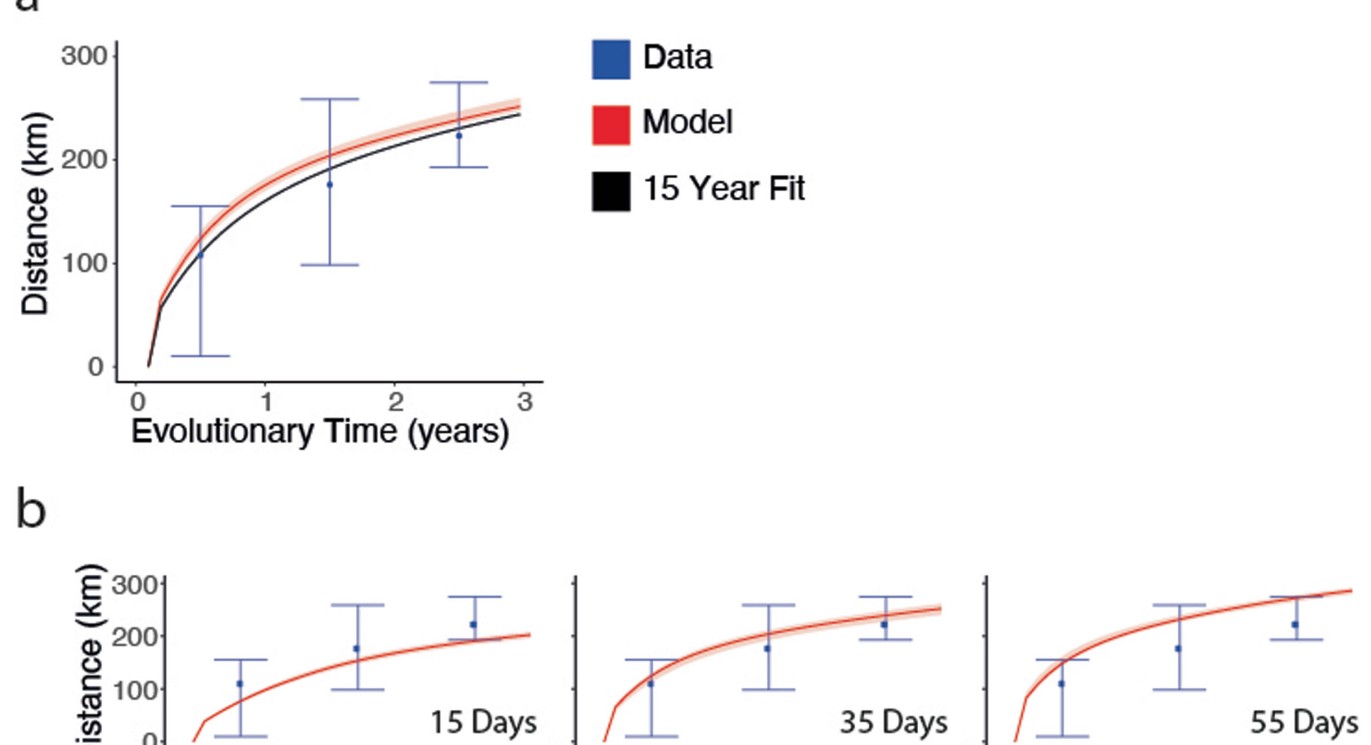

**Extended Data Fig. 12 | Parameter adjustment sensitivity analysis.**
(a) Fitting the mobility model to 15 years of evolutionary distance. Fitting the probabilistic mobility model to pairs of genomes which are 15 years divergent from their MRCA (black), compared against the 10 years used in the main model (red), and the data (blue). (b) Assessing the mean geographic distance per evolutionary time in years for the data (blue), including the sampling probability (red) for, (left) generation time of 15 days, (middle) generation time of 35 days, and (right) generation time of 55 days.

# Reporting Summary

Please do not complete any field with "not applicable" or n/a. Refer to the help text for what text to use if an item is not relevant to your study.
For final submission: please carefully check your responses for accuracy; you will not be able to make changes later.

## Statistics

For all statistical analyses, confirm that the following items are present in the figure legend, table legend, main text, or Methods section.

| n/a | Confirmed | |
|---|---|---|
| ☐ | ☒ | The exact sample size (*n*) for each experimental group/condition, given as a discrete number and unit of measurement |
| ☐ | ☒ | A statement on whether measurements were taken from distinct samples or whether the same sample was measured repeatedly |
| ☐ | ☒ | The statistical test(s) used AND whether they are one- or two-sided *Only common tests should be described solely by name; describe more complex techniques in the Methods section.* |
| ☒ | ☐ | A description of all covariates tested |
| ☐ | ☒ | A description of any assumptions or corrections, such as tests of normality and adjustment for multiple comparisons |
| ☐ | ☒ | A full description of the statistical parameters including central tendency (e.g. means) or other basic estimates (e.g. regression coefficient) AND variation (e.g. standard deviation) or associated estimates of uncertainty (e.g. confidence intervals) |
| ☐ | ☒ | For null hypothesis testing, the test statistic (e.g. *F*, *t*, *r*) with confidence intervals, effect sizes, degrees of freedom and *P* value noted *Give P values as exact values whenever suitable.* |
| ☐ | ☒ | For Bayesian analysis, information on the choice of priors and Markov chain Monte Carlo settings |
| ☒ | ☐ | For hierarchical and complex designs, identification of the appropriate level for tests and full reporting of outcomes |
| ☐ | ☒ | Estimates of effect sizes (e.g. Cohen's *d*, Pearson's *r*), indicating how they were calculated |

*Our web collection on statistics for biologists contains articles on many of the points above.*

## Software and code

Policy information about availability of computer code

| Data collection | Population size estimates were collected from LandScan Global 2017 provided by the Oak Ridge National Laboratory (https://landscan.ornl.gov/). Human mobility data was provided by Meta Data for Good, and is available on the Data for Good portals by access request (https://dataforgood.facebook.com/dfg/about). |
|---|---|
| Data analysis | Computational analysis was conducted in R version 3.6.2 and version 4.0.5 as specified in the GitHub README Bioinformatic analysis was conducted using tools described in the methods section including stan version 2.26.1, VelvetOptimiser v2.2.5, Velvet v1.2.10, SSPACE v2.0, GapFiller v1.11, ABACAS v1.3.1, bwa-MEM v0.7.17, samtools mpileup v1.6, Gubbins v2.4.1, BactDating v1.0, and BEAST v1.10.4. The MCMC was conducted using fMCMC v0.5-1. All bespoke code is available on GitHub at https://github.com/sophbel/geomig_evo_pneumo |

For manuscripts utilizing custom algorithms or software that are central to the research but not yet described in published literature, software must be made available to editors and reviewers. We strongly encourage code deposition in a community repository (e.g. GitHub). See the Nature Portfolio guidelines for submitting code & software for further information.

## Data

Policy information about availability of data

 All manuscripts must include a data availability statement. This statement should provide the following information, where applicable:
  - Accession codes, unique identifiers, or web links for publicly available datasets
  - A description of any restrictions on data availability
  - For clinical datasets or third party data, please ensure that the statement adheres to our policy

> All data are available in the main text or the supplementary materials. All data and code for figures and analysis are accessible at GitHub (https://github.com/ sophbel/geomig_evo_pneumo). All whole genome sequences were deposited in the European Nucleotide Database and accession numbers are available in the GitHub repository and on FigShare (doi: 10.6084/m9.figshare.24219214). Associated metadata is available in the Microreact webserver at the following URL: https://microreact.org/project/7wqgd2gbBBEeBLLPKonbaT-belman2024southafricapneumococcus . All scripts for analysis are available on GitHub at: https:// github.com/sophbel/geomig_evo_pneumo

## Research involving human participants, their data, or biological material

Policy information about studies with human participants or human data. See also policy information about sex, gender (identity/presentation), and sexual orientation and race, ethnicity and racism.

| | |
|---|---|
| Reporting on sex and gender | N/A |
| Reporting on race, ethnicity, or other socially relevant groupings | N/A |
| Population characteristics | N/A |
| Recruitment | N/A |
| Ethics oversight | N/A |

Note that full information on the approval of the study protocol must also be provided in the manuscript.

# Field-specific reporting

Please select the one below that is the best fit for your research. If you are not sure, read the appropriate sections before making your selection.

☒ Life sciences   ☐ Behavioural & social sciences   ☐ Ecological, evolutionary & environmental sciences

For a reference copy of the document with all sections, see nature.com/documents/nr-reporting-summary-flat.pdf

# Life sciences study design

All studies must disclose on these points even when the disclosure is negative.

| | |
|---|---|
| Sample size | The sample sizes encompass the number of genomes sequenced as part of the Global Pneumococcal Sequencing Project. |
| Data exclusions | N/A - No data were excluded |
| Replication | Co-authors and reviewers reviewed the code and verified the reproducibility of the results. |
| Randomization | N/A - the analysis does not involve experimental groups. |
| Blinding | N/A - the analysis does not involve experimental groups. |

# Reporting for specific materials, systems and methods

We require information from authors about some types of materials, experimental systems and methods used in many studies. Here, indicate whether each material, system or method listed is relevant to your study. If you are not sure if a list item applies to your research, read the appropriate section before selecting a response.

## Materials & experimental systems

| n/a | Involved in the study |
|-----|----------------------|
| ☒ | ☐ Antibodies |
| ☒ | ☐ Eukaryotic cell lines |
| ☒ | ☐ Palaeontology and archaeology |
| ☒ | ☐ Animals and other organisms |
| ☒ | ☐ Clinical data |
| ☒ | ☐ Dual use research of concern |
| ☒ | ☐ Plants |

## Methods

| n/a | Involved in the study |
|-----|----------------------|
| ☒ | ☐ ChIP-seq |
| ☒ | ☐ Flow cytometry |
| ☒ | ☐ MRI-based neuroimaging |

