## [Peer Review File · Nature]

Manuscript Title: Geographic migration and fitness dynamics of *Streptococcus pneumoniae*

Reviewer Comments & Author Rebuttals

Reviewer Reports on the Initial Version:

Referees' comments:

Referee #2 (Remarks to the Author):

In this manuscript, Belman and colleagues use *Streptococcus pneumoniae* phylogenomics combined with human mobility data to reconstruct the transmission of *S. pneumoniae* strains across South Africa, and they estimate fitness of serotypes and penicillin resistant and susceptible strains pre- and post-vaccination. They demonstrate that despite extensive diversity even within a province, there is geographic structure among closely related isolates, and spread of pneumococcus across South Africa is slow. To my knowledge, this is the first study incorporating both human mobility data from Meta and endemic bacterial pathogen WGS data to reconstruct pathogen transmission. In addition, they demonstrate that a gravity model of mobility also performed well, suggesting that similar approaches could be used to describe pathogen transmission in settings where mobility data from Meta is unavailable. I have a few comments, which I believe if addressed would strengthen the manuscript.

1. The supplementary material describing the model is extensive and well written. However, references are missing in the supplementary material, which would have been helpful for evaluating the methods. It appears that the list of references was just left out of the document.
2. Code for the mobility modeling work is available on GitHub, and detailed and interactive metadata for isolates is available through microreact. However, the methods used for assembly and phylogenetic analysis are only briefly described and specific versions for software and parameters used in these analyses are missing.
3. While TMRCA's are extensively used in the results, it is not clear if phylogenetic uncertainty was included in the model. Are the results sensitive to placement of ancestral nodes?
4. The results described here are not put into context with previous literature on the impact of negative frequency dependent selection on post-vaccination strain fitness in pneumococcus in the main text (perhaps discussed in the supplement, but the references are missing).

Additional specific comments:

5. Based on Supplementary Text p. 7, error bars represent 2.5 and 97.5 percentiles for statistics. Is this correct? If so, it would be useful to explicitly state this in figure legends as well.
6. It would be easier to read if the supplementary figures were ordered based on the order they are discussed in the text.
7. Line 89: Is this the correct reference here?
8. Line 91-92: What about work estimating strain specific fitness under NFDS post-vaccination (e.g.

Corander et al 2017, Azarian, Martinez et al. 2020)?

9. Lines 143-148/Figure 2A: Is the <500 km category inclusive or exclusive of isolate pairs from the same province?
10. Line 193: Please define DIC acronym.
11. Line 290-310: Is data on antimicrobial use available and if so, has use changed significantly over the time period analyzed here? Would this be expected to impact fitness of resistant and susceptible strains?
12. Line 298: Could a supplementary figure with trends for other antimicrobials be added?
13. Line 333: The manuscript referenced here (describing PopPunk) doesn't address fitness dynamics.
14. Figure 1A. How was this tree rooted? What is the scale for branch lengths?
15. Table S1. Table is cut off on the right side in the supplementary PDF
16. Figure 1F/Table S3. This is predicted resistance from genotype rather than phenotypic resistance, right? Might be helpful to make this clearer in the text and legends.
17. Table S5. Is relatedness SNP distance here? With or without recombinant SNPs?
18. Figure 2F: Adding the word "same" to this panel would make the meaning more clear
19. Figure 3B: What does "infector" indicate in this figure? Also, this figure is referenced in the text on line 180 regarding individuals having a different probability of staying home than the Meta users, but it's not clear where this is displayed in the figure. A more descriptive y axis label would be helpful.
20. Supplementary Text, p. 5: A brief description of WGS processing would be useful here. Additionally, the pipeline for creating GPSC specific alignment should be more thoroughly described (or the pipeline should be made publicly available) including any specific parameters and version numbers of software.
21. Supplementary Text p. 7: Based on the GitHub repository, it seems that two different version of R were used for analyses.

Referee #3 (Remarks to the Author):

Thank you for the opportunity to review “Geographic migration and vaccine-induced fitness changes of *Streptococcus pneumoniae*” by Belman and colleagues. The authors use 6910 pneumococcal genomes collected from carriage and disease in South Africa between 2000-2014 to investigate the rate of dispersal and the impact of conjugate vaccines on vaccine-type and non-vaccine-type fitness. The authors seek to understand the phylogeography of *S. pneumoniae*. Of particular interest is that this study puts the evolutionary time of pneumococcal lineages/serotypes into context, incorporating human mobility to explain the rate of spread. I find the analysis and results of extreme interest. Their methods appear robust, for example, including sensitivity analysis when appropriate or comparing two different methods (e.g., BEAST and Bactdating) to preempt potential questions about their approach. Their conclusions, and the methods as a whole, are of interest to pneumococcal epidemiologists and the broader microbial genomic epidemiology community. I have some general comments below for discussion and some minor points that should be addressed.

1) Line 100 – The authors state that current phylogeographic models don’t allow the investigation of pathogen dispersal processes in the context of human level (e.g., mobility) data. I feel that this statement warrants, at the least, discussion/consideration of structured coalescent models that can allow for joint inference of spatial spread and host level factors, while accounting for uneven temporal sampling. Even early work from Phillippe Lemey and others assess the migration of rabies and human mobility patterns. More recently, improved methods such as those described in <https://doi.org/10.1093/bioinformatics/btu201> details the implementation of structured coalescent models and [doi:10.1093/ve/vez030](https://doi.org/10.1093/ve/vez030) details the use of GLM models within the structured coalescent framework. While these methods are more often applied to viral datasets (see work by Gytis Dudas), it should be described here why they may or may not be appropriate for this application. As a note, it doesn’t appear that this was discussed in the original method publication by Salje et al.

2) In their analysis, the authors include both GPSC level and serotype level analysis. The shift occurs at line 245 when the authors present the results from the Fitness Impact of Vaccine and Antimicrobial Resistance analysis. Prior to that, the authors were focusing on 9 dominant GPSCs. I think this transition could be more explicitly detailed. Also, in instances where strain and lineages are used, it should be clarified if these are referring to GPSCs.

My major question relates to the interplay between the GPSC-level and serotype-level analysis. For example, for the 9 GPSCs assessed, it isn’t clear whether these include multiple or single serotypes. This should be detailed and perhaps included as a heatmap on Figure S1A-I. If there are GPSCs that include multiple serotypes, does the GPSC as a whole behave the same as the serotypes it contains (i.e., do they have the same fitness values and dispersal patterns)? This addresses the question of whether GPSCs can be treated as the same evolutionary unit. Further, Figure S14 shows the estimates for each serotype. I feel that it would be of considerable interest to include the same figure for the GPSCs that were identified in the sample and to detail whether there was any discordance between fitness values of different serotypes found in the same GPSC or the same serotype found in different SCs. This again would speak to whether a GPSC can be treated (in most cases) as a single unit. It may be that all GPSCs studied here only included one serotype (in which you can ignore this comment). Any additional results could be described at Line 280.

3) When discussing AMR, and in particular penicillin resistance, the authors should detail whether this is due to the expansion of resistant lineages that existed prior to PCV or the emergence of resistance in new, previously sensitive, lineages. For the results at line 306, it appears that some of both may be occurring. In this case, the relative contribution of each dynamic to the overall change in AMR should be described.

Minor:

Methods – generation time distribution – Chaguza 2021 appears to show variation in carriage duration by serotype. Were serotype specific clearance rates used or was the range part of the exponential distribution?

Line 60 – I bring this up because I recently went through the exercise of finding the citation for >100 serotypes. So the most appropriate citation is likely: "Ganaie, Feroze, et al. "A new pneumococcal capsule type, 10D, is the 100th serotype and has a large cps fragment from an oral streptococcus." MBio 11.3 (2020): 10-1128."

Line 161 – difference were observed in mixing among isolates collected from different ages (>5 yrs). Were any other differences in age group assessed?

Maps of population density and mobility data would be of interest in the supplemental material.

Author Rebuttals to Initial Comments:

Referee #2 (Remarks to the Author):

In this manuscript, Belman and colleagues use *Streptococcus pneumoniae* phylogenomics combined with human mobility data to reconstruct the transmission of *S. pneumoniae* strains across South Africa, and they estimate fitness of serotypes and penicillin resistant and susceptible strains pre- and post-vaccination. They demonstrate that despite extensive diversity even within a province, there is geographic structure among closely related isolates, and spread of pneumococcus across South Africa is slow. To my knowledge, this is the first study incorporating both human mobility data from Meta and endemic bacterial pathogen WGS data to reconstruct pathogen transmission. In addition, they demonstrate that a gravity model of mobility also performed well, suggesting that similar approaches could be used to describe pathogen transmission in settings where mobility data from Meta is unavailable. I have a few comments, which I believe if addressed would strengthen the manuscript.

Response: We thank the reviewer for their positive feedback on the manuscript. We have addressed their comments below and agree that they have strengthened the manuscript most notably the important incorporation of phylogenetic uncertainty into our estimates.

Comment 1.1. The supplementary material describing the model is extensive and well written. However, references are missing in the supplementary material, which would have been helpful for evaluating the methods. It appears that the list of references was just left out of the document.

Response: All references are numbered in the supplementary materials continuing from the main text as recommended in the submission guidelines. However, it seems that they became unlinked when split into the supplementary file - they are now referenced appropriately.

Specific changes to document: *This has been corrected in the updated manuscript.*

Comment 1.2. Code for the mobility modeling work is available on GitHub, and detailed and interactive metadata for isolates is available through microreact. However, the methods used for assembly and phylogenetic analysis are only briefly described and specific versions for software and parameters used in these analyses are missing.

Response: Thank you for noticing this absence. We have now included the appropriate citations linking to the GitHub repositories containing the assembly and mapping pipelines which was absent in the previous version. Additionally we include further detail on the methods, software, specific versions, and quality parameters included in these pipelines.

46. Sanger Pathogen Informatics, vr-codebase (2022), (available at <https://github.com/sanger-pathogens/vr-codebase>).

51. BactGenScripts (2023), (available at <https://github.com/sanger-pathogens/bact-gen-scripts>).

Specific changes to document: *The methods section now includes:*

“Assembly was performed using Wellcome Sanger Institute pathogen informatics automated pipelines and is freely available for download from GitHub under an open source license, GNU GPL 3 (46). For each sample, sequence reads were used to create multiple assemblies using VelvetOptimiser v2.2.5 and Velvet v1.2.10 (47). An assembly improvement step was applied to the assembly with the best N50 and contigs scaffolded using SSPACE v2.0 (48) and sequence gaps filled using GapFiller v1.11 (49). Assembly quality control parameters include

a minimum average sequencing depth of 20X and an assembly length of 1.9-2.3Mb. Sequences with more than 15% heterozygous SNP sites were excluded.

We created reference genomes for each GPSC using ABACAS v1.3.1 to order the contigs from a representative of each GPSC mapped to *Streptococcus pneumoniae* (strain ATCC 700669/Spain 23F-1) [EMBL accession: FM211187](50). Any contigs which did not align were concatenated to the end. We multiply mapped all genomes from each Dominant GPSC against these references respectively using a custom mapping, variant calling, and local realignment around indels pipeline (`multiple_mappings_to_bam.py`)(51) using `bwa-MEM` v0.7.17 (52) and `samtools mpileup` v1.6 (53). The minimum base quality for a base to be considered was 50. The minimum mapping quality for a SNP to be called was 20 with a minimum of 8 reads matching the SNP. We built trees masking recombination regions using `Gubbins` v2.4.1 (54) with the hybrid model which uses `FastTree` for the first iteration and `RAxML` subsequently (55) and a GTR model. We converted branch length to time using `BactDating` v1.0 with a mixed gamma, relaxed clock model(56). We compared concordance between `BEAST` v1.10.4 (57) with both strict and relaxed clocks, and a Bayesian skyline prior. As the results were concordant, we used `BactDating` due to its shorter runtime (Figure S23). ”

Comment 1.3. While TMRCA's are extensively used in the results, it is not clear if phylogenetic uncertainty was included in the model. Are the results sensitive to placement of ancestral nodes?

Response: We agree that phylogenetic uncertainty is important to incorporate. We previously obtained uncertainty estimates of the date of each node from our maximum clade credibility trees, however, phylogenetic uncertainty was not included in the model estimation. We have now incorporated phylogenetic uncertainty both into the Relative Risk Framework and the Mechanistic Mobility model more directly.

In the Relative Risk Framework we have re-estimated the uncertainty using 100 randomly selected trees from the `BactDating` posterior. This produced a small change in the risk of pairs from the same province having an ancestor within 5-years to 3.87 [95%CI 3.01-5.1] from 2.43 [95%CI 2.02-3.17]. We have included this uncertainty estimation throughout the relative risk framework.

In the Mechanistic mobility model, we now include uncertainty in the phylogenetic reconstructions by repeatedly refitting the model using 50 randomly selected trees from the `BactDating` posterior. We note that our phylogeographic models are informed by tips separated by small evolutionary distances (<10 years). As the uncertainty for pairs of tips separated by under 10 years of evolutionary time is typically small (unlike the uncertainty associated with deeper nodes), there is only a small change in the resulting confidence intervals for the mean duration in individuals' home community: CIs of 93.8-95.0% (previously 93.8-94.7%).

Specific changes to document:

For the relative risk framework we have updated Figures 1D, Figure 2, Figures S4-S5 as well as Tables S5-S6 to reflect the revised estimates incorporating the `BactDating` posterior. We have also updated the main results to reflect the new values.

For the mechanistic mobility model we now include revised confidence intervals for the mean duration that individuals spend in their home community that includes phylogenetic uncertainty and updated Figure 3B to reflect this.

We have also updated the methods to include this uncertainty estimation process incorporating phylogenetic uncertainty across node dates.

Comment 1.4 The results described here are not put into context with previous literature on the impact of negative frequency dependent selection on post-vaccination strain fitness in pneumococcus in the main text (perhaps discussed in the supplement, but the references are missing).

Response: We agree that negative frequency dependent selection has an important role in dictating the prevalence of a specific lineage. In the revised manuscript, we highlight existing knowledge on the role of NFDS.

In our base model we use a simplified assumption that all groups have a single fitness parameter based on NVT/VT type or AMR. This ultimately assumes that fitter groups will completely replace other groups. The presence of NFDS would instead result in an intermediate equilibrium level, not complete absence/presence. To assess the impact on having an equilibrium level away from complete absence/presence on our estimates of fitness, we conducted sensitivity analyses that enforced different equilibria. We found that the fitness parameter estimates were largely unchanged (see Figure S29 below).

In the revised manuscript, we further discuss this important point.

Figure S29. **Fitness estimates with varying carrying capacity across VT and NVT serotypes** A) pre-PCV, B) post-PCV introduction, C) and testing the difference to the best WAIC for each. The black triangle represents the carrying capacity utilized in the main model.

Specific changes to document: In the main text we have now included these citations directly

“While there has been success in predicting the fitness of individual isolates based on the overall gene distribution in a population, quantitative measures of fitness linked to the serotype of individual isolates are lacking (19, 20). This includes quantifying the time it takes after implementation for vaccines to impact the serotype composition in the country as well as

quantifying the serotype growth pre- and post- vaccine at different time points. This is a critical knowledge gap, as serotype distributions ultimately drive vaccine development and deployment strategies.”

We have also detailed the importance of NFDS in the “Fitness model carrying capacity sensitivity” section in the methods with the following line.

“However, it is important to note that this model is robust to set variable carrying capacities while being unable to estimate them, in contrast to NFDS. Our model assumes that there is a population replacement up to the carrying capacity and doesn’t allow for fine scale estimates of an equilibrium. This fitness model can be used as a quantitative descriptor of the growth of distinct groups (i.e. penicillin-resistance or vaccine type serotypes in the context of a perturbation) but cannot describe the complex underlying fitness impacts of changing frequencies of genes under NFDS (19).”

19. J. Corander, C. Fraser, M. U. Gutmann, B. Arnold, W. P. Hanage, S. D. Bentley, M. Lipsitch, N. J. Croucher, *Frequency-dependent selection in vaccine-associated pneumococcal population dynamics. Nature Ecology & Evolution. 1, 1950–1960 (2017).*

20. T. Azarian, P. P. Martinez, B. J. Arnold, X. Qiu, L. R. Grant, J. Corander, C. Fraser, N. J. Croucher, L. L. Hammitt, R. Reid, M. Santosham, R. C. Weatherholtz, S. D. Bentley, K. L. O’Brien, M. Lipsitch, W. P. Hanage, *Frequency-dependent selection can forecast evolution in Streptococcus pneumoniae. PLOS Biology. 18, e3000878 (2020).*

Additional specific comments:

Comment 1.5 Based on Supplementary Text p. 7, error bars represent 2.5 and 97.5 percentiles for statistics. Is this correct? If so, it would be useful to explicitly state this in figure legends as well.

Response: Thank you, this is correct and we have now included this in legends throughout.

Comment 1.6 It would be easier to read if the supplementary figures were ordered based on the order they are discussed in the text.

Response: We have reordered the supplementary figures according to their appearance in the text beginning with those in the main text and then the methods.

Comment 1.7 Line 89: Is this the correct reference here?

Response: This is the reference for the GPS collection which includes papers using the pre- and post- PCV WGS data for many countries describing the population dynamics. However, we have now additionally specifically referenced 3 of these papers including 2 describing dynamics in South Africa specifically, and one describing post-PCV dynamics in a carriage population in Cambodia.

Comment 1.8 Line 91-92: What about work estimating strain specific fitness under NFDS post-vaccination (e.g. Corander et al 2017, Azarian, Martinez et al. 2020)?

Response: This is an important point. We agree that NFDS based models are well placed to describe the dynamics of the overall accessory genes in the population and can be leveraged to forecast the post-PCV population (as is described very clearly in Azarian, Martinez et al. 2020). What is lacking currently is a method to quantify the growth and describe the timing of that growth given population level serotype information rather than the entire genomic composition. We have changed the sentence to clarify that while NFDS based models are powerful, they do not provide a simple quantification and do not provide quantification of serotype specific growth (see Response 1.4)

Specific Changes to Document: We now include the following text: *“While there has been success in predicting the fitness of individual isolates based on the overall gene distribution in a population, quantitative measures of fitness linked to the serotype of individual isolates are lacking (19, 20). This includes quantifying the time it takes after implementation for vaccines to impact the serotype composition in the country as well as quantifying the serotype growth pre- and post- vaccine at different time points. This is a critical knowledge gap, as serotype distributions ultimately drive vaccine development and deployment strategies.”*

Comment 1.9 Lines 143-148/Figure 2A: Is the <500 km category inclusive or exclusive of isolate pairs from the same province?

Response: This is exclusive of pairs of isolates from the same province as is detailed in the legend and colour scheme for Figure 2A.

As stated in the figure legend:

*“(A) Risk ratio of being the same GPSC within a province (blue), **between different provinces** over increasing distance (red), compared to geographically distant pairs (>1000km) (reference).”*

Comment 1.10 Line 193: Please define DIC acronym.

Response: We have now included “deviance information criterion (DIC)” in the main text.

Comment 1.11 Line 290-310: Is data on antimicrobial use available and if so, has use changed significantly over the time period analyzed here? Would this be expected to impact fitness of resistant and susceptible strains?

Response: Unfortunately the data for antimicrobial use is not available at a useful resolution to incorporate into the models across South Africa. Co-trimoxazole was recommended in 2000 by the WHO as prophylaxis against pneumocystic pneumonia in HIV infected individuals and is still used extensively in the country as of 2021, and Klein et al. 2018 (<https://www.pnas.org/doi/10.1073/pnas.1717295115#supplementary-materials>) highlighted that South Africa, an upper middle income country, had increased by 5-10 defined daily doses (DDD) of antimicrobials across 2000-2015 in line with the increase seen in the rest of the world. This may partially explain the increasing proportion of penicillin resistance among NVTs however, the consistently decreasing proportion of penicillin resistance in VTs suggests that the vaccine associated pressure is greater than any applied by increased exposure to beta-lactams. Should a precise dataset become available, detailed estimation of the impact of antimicrobial usage on fitness dynamics would be possible and very interesting. Hopefully this will be included in future work.

Specific Changes to Document: *As this is a very important point we have now included the sentence “Changing patterns of antimicrobial use in this population may also result in shifts in resistance patterns.”*

Comment 1.12. Line 298: Could a supplementary figure with trends for other antimicrobials be added?

Response: The trends for penicillin, erythromycin, co-trimoxazole, and clindamycin are summarised in Figure 1F. In the revised manuscript, we have now additionally included the detailed trends for erythromycin, co-trimoxazole, and clindamycin in Figure S19. We fit the model to penicillin-resistance due its relevance as a first line drug for pneumococcal disease and high proportion of resistance in the population.

Specific Changes to Document:

We now include in the main body the following text: “The trends, while still present, were less clear in the other investigated antimicrobials (Figure S19). Due to the relevance of penicillin as a first-line antimicrobial for pneumococcal disease and high proportion of resistance in this population we used our same modelling framework and were able to recover the observed proportions of strains that were resistant to penicillin over time.”

“Figure S19. The proportional trends overall (black) and within VT (blue) and NVT (red) serotypes for in-silico predicted A) penicillin B) erythromycin C) co-trimoxazole and D) clindamycin.”

Comment 1.13 Line 333: The manuscript referenced here (describing PopPunk) doesn't address fitness dynamics.

Response: The reference was numbered incorrectly. It now references Gladstone et al. 2019, EBioMedicine describing the pneumococcal population dynamics following vaccination as well as the Global Pneumococcal Sequencing collection in Microbial Genomics.

Comment 1.14 Figure 1A. How was this tree rooted? What is the scale for branch lengths?

Response: *This tree is midpoint rooted and the scale refers to SNPs per site. We have now stated this in the figure legend.*

Specific Changes to Document:

“The branch length legends refer to SNPs per site and trees are midpoint rooted.”

Comment 1.15 Table S1. Table is cut off on the right side in the supplementary PDF

Response: Thank you for noticing this; it is now amended.

Comment 1.16 Figure 1F/Table S3. This is predicted resistance from genotype rather than phenotypic resistance, right? Might be helpful to make this clearer in the text and legends.

Response: We have now included this in both places.

Specific Changes to Document:

*“F) The proportion of **in-silico predicted** antimicrobial resistant isolates for 4 drugs across the study period. The vertical lines denote the introduction of PCV7 in 2009 and PCV13 in 2011. An interactive phylogeny and metadata are available on Microreact.”*

“Table S3.

*Antimicrobial resistance across South Africa for three classes of antimicrobials. The antimicrobials investigated include penicillin (beta-lactam), erythromycin, clindamycin (macrolides), and co-trimoxazole (sulfonamide). Some isolates may be resistant to more than one antibiotic. **Antimicrobial-resistance is as predicted by the in-silico antimicrobial resistance pipeline.**”*

Comment 1.17 Table S5. Is relatedness SNP distance here? With or without recombinant SNPs?

Response: Relatedness refers to the divergence time between pairs as determined by BactDating excluding recombination as masked by Gubbins. We have made this clearer in the table by changing “Relatedness” to “Divergence time (years)”

Comment 1.18 Figure 2F: Adding the word “same” to this panel would make the meaning more clear.

Response: We have now included the word ‘same’ in Figure 2F for clarity.

Comment 1.19 Figure 3B: What does “infector” indicate in this figure? Also, this figure is referenced in the text on line 180 regarding individuals having a different probability of staying home than the Meta users, but it’s not clear where this is displayed in the figure. A more descriptive y axis label would be helpful.

Response: *As our phylogenetic approaches incorporate the behaviour of both the infector and the surrounding susceptible population, we can disentangle the role of each in pathogen*

spread (something traditional phylogenetic approaches cannot do). This reflects that the mobility of the pathogen at one transmission generation reflects the mobility of the infector and the mobility of the susceptible individual. However, we agree that the figure caption was poorly worded. We have now corrected this. As some of the dots the y-axis represents proportion of human mobility and others it represents overall pathogen mobility we have now included clarification as to what each dot represents in the caption rather than changing the y-axis directly.

Specific changes to document: The figure caption for this panel is now: “(B) The proportion of individual or pathogen mobility as a function of distance from the origin location. We compare the mean distance travelled when we consider the infector only (crossed square), when we consider the mobility of both the infector and the infectee (black filled circle) after a single transmission generation, as well as the overall mobility of the pathogen after 10 generations (triangle). As a comparison, we present the expected pathogen spread after a single generation if transmission was completely spatially random (maroon). We also present the difference between Meta users (blue circle) and the movement of those involved in transmission.”

Comment 1.20 Supplementary Text, p. 5: A brief description of WGS processing would be useful here. Additionally, the pipeline for creating GPSC specific alignment should be more thoroughly described (or the pipeline should be made publicly available) including any specific parameters and version numbers of software.

Response: Agreed, thank you for noting this absence. As mentioned in Comment 1.2 we have included details of the pipelines used including specific versions and appropriate citations. Further, we have included citations to the appropriate GitHub repositories containing the reproducible pipelines.

Comment 1.21 Supplementary Text p. 7: Based on the GitHub repository, it seems that two different version of R were used for analyses.

Response: We have adjusted the text appropriately detailing both R versions and what they were used for respectively. We cite R Core Team as requested <https://ropensci.org/blog/2021/11/16/how-to-cite-r-and-r-packages/>

Specific Changes to Document:

“All statistical analysis for the Relative Risk Framework and the Mobility Model was performed in R v3.6.2(62).”

“All statistical analysis for the Fitness Model was performed in R v4.0.5(62).”

62. R Core Team, R: A Language and Environment for Statistical Computing (2022), (available at <https://www.r-project.org/>).

Referee #3 (Remarks to the Author):

Thank you for the opportunity to review “Geographic migration and vaccine-induced fitness changes of *Streptococcus pneumoniae*” by Belman and colleagues. The authors use 6910

pneumococcal genomes collected from carriage and disease in South Africa between 2000-2014 to investigate the rate of dispersal and the impact of conjugate vaccines on vaccine-type and non-vaccine-type fitness. The authors seek to understand the phylogeography of *S. pneumoniae*. Of particular interest is that this study puts the evolutionary time of pneumococcal lineages/serotypes into context, incorporating human mobility to explain the rate of spread. I find the analysis and results of extreme interest. Their methods appear robust, for example, including sensitivity analysis when appropriate or comparing two different methods (e.g., BEAST and BactDating) to preempt potential questions about their approach. Their conclusions, and the methods as a whole, are of interest to pneumococcal epidemiologists and the broader microbial genomic epidemiology community. I have some general comments below for discussion and some minor points that should be addressed.

Response: *We thank the reviewer for their positive comments particularly their interest in the fitness model including GPSCs as this has led to additional interesting insight into the amount of variance explained by GPSC-serotype and GPSC-vaccine type composition as well as how much variance is left unexplained (GPSC-specific).*

Comment 2.1 Line 100 – The authors state that current phylogeographic models don't allow the investigation of pathogen dispersal processes in the context of human level (e.g., mobility) data. I feel that this statement warrants, at the least, discussion/consideration of structured coalescent models that can allow for joint inference of spatial spread and host level factors, while accounting for uneven temporal sampling. Even early work from Phillippe Lemey and others assess the migration of rabies and human mobility patterns. More recently, improved methods such as those described in <https://doi.org/10.1093/bioinformatics/btu201> [doi.org] details the implementation of structured coalescent models and doi:10.1093/ve/vez030 details the use of GLM models within the structured coalescent framework. While these methods are more often applied to viral datasets (see work by Gytis Dudas), it should be described here why they may or may not be appropriate for this application. As a note, it doesn't appear that this was discussed in the original method publication by Salje et al.

Response: *This is an important point. We completely agree that structured coalescent approaches, such as those referenced by the reviewer, are a key part of the toolbox to understand pathogen spread. However, they rely on characterising drivers of overall pathogen flow between any two locations - i.e., the overall effect of many different transmission events between locations. There is no simple way for these approaches to consider what happens at each transmission event - i.e., the behaviour of the infected individual, the behaviour of the surrounding susceptible population. These differing ecological scales of analysis are a critical issue, as we would prefer to make mechanistic insights into what happens at each transmission event. The approaches we implement do allow us to consider individual transmission events, as we translate each edge into transmission events then integrate over all possible ways that two places can be linked. We clarify these important differences in the revised manuscript.*

Specific Changes to Document: *We now include the following text:*

“Mathematical models applied to geolocated pathogen genome sequence data provide a powerful tool to disentangle the changing prevalence of different lineages. However, most phylogeographic models focus on the rate of pathogen flow between locations, which represents the overall effect of multiple transmission chains, and therefore considers a different ecological scale to the specific behaviours of infected people and the surrounding population at each transmission generation (22). The relationship between behaviours of individuals at each transmission step and the overall patterns of pathogen flow between locations are complex and non-linear. Most phylogeographic approaches also struggle to

account for changing levels of surveillance in both space and time. We have developed mechanistic models that use the generation time distribution to translate time-resolved phylogeny branch lengths into individual transmission events. Together with measures of human mobility probabilities and human population distribution we can elucidate mechanisms of pneumococcal migration at each transmission generation.”

Comment 2.2 In their analysis, the authors include both GPSC level and serotype level analysis. The shift occurs at line 245 when the authors present the results from the Fitness Impact of Vaccine and Antimicrobial Resistance analysis. Prior to that, the authors were focusing on 9 dominant GPSCs. I think this transition could be more explicitly detailed. Also, in instances where strain and lineages are used, it should be clarified if these are referring to GPSCs.

Response: *Thank you for highlighting this lack of clarity in the paper. The 9 dominant GPSCs are included in the mobility model while the fitness model includes all 184 GPSCs that we detected in the entire dataset. We have altered the text to make this clearer. We focus on the 9 dominant GPSCs for the mobility model as these approaches rely on phylogenetic reconstructions with many tips. There was limited value in adding GPSCs with fewer sequences into this analysis. We also noted the remarkable stability of our estimates across the nine GPSCs suggesting our results were robust across the wider distribution of GPSCs.*

Of the 9 dominant GPSCs only GPSC2 and GPSC79 comprise a single serotype (serotype 1 and serotype 4 respectively) where the other 7 GPSCs contain a mix of different serotypes in varying frequencies. We have now included this breakdown in Table S4.

Specific Changes to Document: *In the opening paragraph we have included a sentence to clarify the differences between strains and lineages throughout **“We refer to lineage synonymously with GPSC while a strain references any particular circulating phenotype (including specific serotypes and AMR).”***

In the beginning of the Inferring Mechanisms of Migration using Human Mobility section we have included further clarification.

*“To understand whether human mobility can explain the slow spread of the pneumococcus, we built a mechanistic model of geographic spread **fit to the nine dominant GPSCs** and the observed province in which our genome sequences were isolated.”*

In the Fitness Impact of Vaccine and Antimicrobial Fitness section we have also included further clarification.

*“To quantify changes in fitness linked to the vaccines, we fitted models to the annual distribution of serotypes **across 184 GPSCs from the full data set**, allowing for differential fitness in serotypes included in PCV7 (serotypes: 4,6B,9V,14,19F,18C,23F), PCV13 (additional serotypes: 1,3,5,6A,7F,19A), and those not included in the vaccine (NVT).”*

Comment 2.3. My major question relates to the interplay between the GPSC-level and serotype-level analysis. If there are GPSCs that include multiple serotypes, does the GPSC as a whole behave the same as the serotypes it contains (i.e., do they have the same fitness values and dispersal patterns)? This addresses the question of whether GPSCs can be treated as the same evolutionary unit. This again would speak to whether a GPSC can be treated (in most cases) as a single unit. It may be that all GPSCs studied here only included

one serotype (in which you can ignore this comment). Any additional results could be described at Line 280.

Response: *We thank the reviewer for this point. Due to many GPSCs comprising multiple serotypes and the strong effects of PCV targeting individual serotypes, GPSCs cannot be treated as a single evolutionary unit. This can be visualised and explored in the Microreact at the following link as is included in Figure legend 1: <https://microreact.org/project/4g3tQZkt4ncWWV9ybPaG3W-southafrica6910>.*

In our base model, we consider that the increase or decrease of a lineage as being solely driven by the VT or NVT label attached to the lineage and a single estimate of the VT or NVT fitness. Therefore there is an inherent assumption that at a GPSC level, whether or not the GPSC becomes more or less prevalent will depend only on the VT/NVT composition at the start of the study period. This critical point from the reviewer has led us to consider the predictive power of simply knowing the VT/NVT status versus the serotype distribution of isolates within a GPSC to be able to predict the dynamics of GPSCs over the study period.

In the revised manuscript, we consider whether using (a) fitness estimates by NVT/VT status and (b) the estimated proportion of isolates that are VT vs NVT within a GPSC at the start of the study period, can explain the changing prevalence of GPSC over the study period. In a competing model, we explore whether allowing for differential fitness by serotype improves model performance. We find remarkably that just using our NVT/VT fitness estimates can explain 56% of the variance of the heterogeneity in GPSC proportion. Including serotype-specific information results in a small improvement to 58% of the variance explained. This provides strong support for our model. It also suggests that the unexplained variance may be explained by GPSC-specific differences in fitness.

We believe this additional analysis is an exciting new addition to the paper and a valuable insight as a whole.

Specific Changes to Document:

We now include a description of the new analysis in the Main Text:

“We explored whether using the proportion of isolates which were VT versus NVT within each GPSC at the start of the study period and our fitness estimates could explain the subsequent dynamics of individual GPSCs. We found that simply using VT/NVT fitness estimates could explain 56% of the variance in individual GPSC prevalence at any time. Allowing for serotype-specific differences produced a small improvement, explaining 58% (Figure S16). The unexplained variance reflects GPSC-specific fitness and is likely driven by NFDS. This highlights the predictive nature of a GPSCs serotype composition in determining PCV driven GPSC dynamics.”

We have additionally now included this extra section in the supplementary methods:

“Exploration of predicted GPSC dynamics based on their serotype composition

We explored whether using the proportion of isolates which were VT versus NVT within each GPSC at the start of the study period and our fitness estimates could explain the subsequent dynamics of individual GPSCs.

To predict the dynamics of individual GPSCs based on their serotype composition, we used the same framework as described above, applied to each GPSC-serotype group in our

dataset. As the number of such groups is large ($N = 340$ GPSC-serotype groups), we restricted the analysis to the GPSCs present at a minimum prevalence of 1%. This led to 26 GPSCs being considered for this analysis (representing 74.7% of the dataset), split in a total of 101 GPSC-serotype groups. We then modelled the dynamics of each GPSC-serotype group, estimating one starting frequency per group and using the previously estimated fitness parameters, either by vaccine-type (VT) or serotype. We also considered a model with no fitness parameters (relative fitness = 1). To assess the model performances, we computed the Akaike information criterion (AIC)(68) using the average likelihood of each model and the number of parameters used in each model. We also computed the predicted GPSC dynamics over time for each model by summing all the predicted GPSC-serotype group dynamics for each GPSC. As a measure of goodness of fit we used the coefficient of determination of the observed vs predicted GPSC proportions each year:

$$R^2 = 1 - \frac{\sum (observed\ proportion - predicted\ proportion)^2}{\sum (observed\ proportion - mean\ observed\ proportion)^2}$$

R^2 is also the proportion of variation explained by the variables considered in each model.

Our model estimates a constant fitness for each group considered (vaccine-types or individual serotypes), assuming that if a group has the highest fitness, it will eventually replace the other groups. This assumption is meaningful for vaccine-types and serotypes, as some are directly targeted by the vaccines, a strong selective force in the population. However; the fitness of each GPSC cannot be modelled with this simple assumption as it has been shown that their fitness is inherently multifactorial, as described in the NFDS model (19).”

68. H. Akaike, "Information Theory and an Extension of the Maximum Likelihood Principle" in Selected Papers of Hirotugu Akaike, E. Parzen, K. Tanabe, G. Kitagawa, Eds. (Springer, New York, NY, 1998; https://doi.org/10.1007/978-1-4612-1694-0_15), Springer Series in Statistics, pp. 199–213.

19. J. Corander, C. Fraser, M. U. Gutmann, B. Arnold, W. P. Hanage, S. D. Bentley, M. Lipsitch, N. J. Croucher, Frequency-dependent selection in vaccine-associated pneumococcal population dynamics. *Nature Ecology & Evolution*. 1, 1950–1960 (2017).

Figure S16. The observed versus expected R^2 model fits for each GPSC overall, when including A) no fitness parameters ($R^2=0.49$)(green) B) VT/NVT fitness parameters ($R^2 =0.56$) (red) and C) the serotype specific fitness parameters ($R^2 =0.58$)(orange)

Table S12. Model outputs and information including three models 1) no fitness parameters (top), 2) the fitness parameters from the VT/NVT model, and 3) the serotype specific fitness

parameters.(bottom) Model specific information including the number of fitness parameters per model and the AIC are on the left. The middle includes the coefficient of determination for the observed versus expected GPSC-serotype proportions overall and including only the NVT serotypes, the PCV7 serotypes, and the PCV13 serotypes. The final column includes the coefficient of determination for GPSC proportion overall in the population for each model.

Model Specific Information			GPSC-serotype				GPSCs
Model	Number of fitness parameters	AIC	R2 - overall	R2 NVT	R2 PCV7	R2 PCV13	R2
No fitness	100	4200.935	0.7101255	0.3021831	0.7168886	0.7971715	0.4897332
VT	106	3754.629	0.7599263	0.5968793	0.7566153	0.7872197	0.5607038
Serotype	184	3648.862	0.776494	0.6938755	0.7756981	0.779783	0.5836145

Comment 2.4. Further, Figure S14 shows the estimates for each serotype. I feel that it would be of considerable interest to include the same figure for the GPSCs that were identified in the sample and to detail whether there was any discordance between fitness values of different serotypes found in the same GPSC or the same serotype found in different SCs.

Response: We agree that understanding the interplay between serotype fitness and GPSC fitness is important. However, our current model does not allow us to answer this question as GPSCs cannot be treated independently of their serotypes. This would require further analysis, that is beyond the scope of this paper. Indeed, our model estimates a constant fitness for each group considered (vaccine-types or individual serotypes), assuming that if a group has the highest fitness, it will eventually replace the other groups. This assumption is meaningful for vaccine-types and serotypes, as some are directly targeted by the vaccines, and would be strongly selected against in the population. However; the fitness of each GPSC cannot be modelled with this simple assumption as it has been shown that their fitness is inherently multifactorial, as described in the NFDS model (19).

Nonetheless, to visually explore the different serotype dynamics across GPSCs, we have included additional Figures S17 and S18 containing the model fits for the GPSC-serotype model clustered both by serotype and by GPSC.

Specific Changes to Document:

We have also included additional figures and tables including the coefficient of determination plot, the values associated with this plot, and the GPSC-serotype fits clustered by both serotype and GPSC.

Fig. S17. GPSC-Serotype fits using the serotype specific fitness parameters. The fits are for each GPSC-serotype pair for all pairs where the same serotype appears across multiple GPSCs. These are grouped by PCV7 serotypes, PCV13 serotypes, and NVTs. NVT serotypes which appear in only a single GPSC are excluded.

Fig. S18. GPSC-Serotype fits using the serotype specific fitness parameters. The fits are for each GPSC-serotype pair for all pairs where the same serotype appears across multiple GPSCs. These are grouped GPSCs.

Comment 2.5 When discussing AMR, and in particular penicillin resistance, the authors should detail whether this is due to the expansion of resistant lineages that existed prior to PCV or the emergence of resistance in new, previously sensitive, lineages. For the results at line 306, it appears that some of both may be occurring. In this case, the relative contribution of each dynamic to the overall change in AMR should be described.

Response: *This is an important point. Penicillin resistance was widely present at the start of our dataset, including prior to vaccination. This would suggest that the increase in AMR in NVT strains was due to expansion of these pre-existing lineages which harboured resistance. However, we note we cannot explicitly disentangle expansion from the emergence of resistance within individual isolates using this phylogenetic analysis. We have described this in the main text.*

We have also now included AMR in the interactive Microreact phylogeny and metadata contained in the Figure 1 caption, for exploration by readers:

<https://microreact.org/project/9piQGo6YNAsYodfrCEVvi6-southafrica6910amr>

Specific Changes to Document:

“The widespread presence of penicillin resistance at the beginning of our data collection period would implicate expansion of resistance among existing lineages; however, we cannot exclude the emergence of some novel resistance forms.”

Minor Comments

Comment 2.6 Methods – generation time distribution – Chaguza 2021 appears to show variation in carriage duration by serotype. Were serotype specific clearance rates used or was the range part of the exponential distribution?

Response: *We included the overall carriage duration median estimate of 38 days which transforms to an exponential clearance rate of 1/38 or 0.0263 as detailed in the fourth paragraph of the results section in Chaguza et al. 2021. Our model interrogates the overall mobility of the pneumococcus rather than serotype specifically.*

Specific Changes to Document:

*“We sample 1000 carriage durations from an exponential distribution with means which are inverse to the clearance rates estimated **across serotypes** in Chaguza, C., et al. 2021(24) (Clearance Rate: 0.026 [95% CI: 0.025–0.028 episodes/day), and **in** Abdullahi, O., et al., 2012 (Clearance Rate: 0.032 episodes/day [95% CI: 0.030–0.034])(25).”*

Comment 2.7 Line 60 – I bring this up because I recently went through the exercise of finding the citation for >100 serotypes. So the most appropriate citation is likely: “Ganaie, Feroze, et al. “A new pneumococcal capsule type, 10D, is the 100th serotype and has a large cps fragment from an oral streptococcus.” MBio 11.3 (2020): 10-1128.”

Response: Thank you very much for noting this. We have now included this citation here as well as the more recent publication Ganaie et al. 2023

<https://pubmed.ncbi.nlm.nih.gov/36971549/> describing the antigenically distinct serotypes 36A and 36B within serotype 36.

Specific Changes to Document:

“The pneumococcus comprises >100 known antigenically distinct serotypes and over 900 classified lineages (also known as global pneumococcal sequence clusters; GPSCs)(3–5)”

3. J. A. Lees, S. R. Harris, G. Tonkin-Hill, R. A. Gladstone, S. W. Lo, J. N. Weiser, J. Corander, S. D. Bentley, N. J. Croucher, *Fast and flexible bacterial genomic epidemiology with PopPUNK. Genome Res.* **29**, 304–316 (2019).

4. F. Ganaie, J. S. Saad, L. McGee, A. J. van Tonder, S. D. Bentley, S. W. Lo, R. A. Gladstone, P. Turner, J. D. Keenan, R. F. Breiman, M. H. Nahm, *A New Pneumococcal Capsule Type, 10D, is the 100th Serotype and Has a Large cps Fragment from an Oral Streptococcus. mBio.* **11**, e00937-20 (2020).

5. F. A. Ganaie, J. S. Saad, S. W. Lo, L. McGee, S. D. Bentley, A. J. van Tonder, P. Hawkins, J. D. Keenan, J. J. Calix, M. H. Nahm, *Discovery and Characterization of Pneumococcal Serogroup 36 Capsule Subtypes, Serotypes 36A and 36B. J Clin Microbiol.* **61**, e0002423 (2023).

Comment 2.8 Line 161 – difference were observed in mixing among isolates collected from different ages (>5 yrs). Were any other differences in age group assessed?

Response: *Differences in carriage and disease risk by age is a critical issue hence our attempt to disentangle it. However, 23.7% of our dataset was missing age data and the vast majority of isolates come from younger children (78.8% <5). The concentration of our dataset within one age group unfortunately limits the granularity of the analyses we could perform. As the majority of pairs with <5 years between them are under 5 years-old, the greater time to homogenise when only including pairs with >5 years between them, may be due to transmission being mostly between children rather than via adults.*

Specific Changes to Document: We have now included Table S7 for clarity:

Table S7. Age breakdown of isolates. Across All South African isolates (N=6910) and across Dominant GPSCs (N=2575). The percent of isolates in each age breakdown are of those with age data (N=5166 and N=1964 respectively).

Age Group	All South Africa Isolates (N)	All South Africa Isolates (%)	Dominant GPSCs (N)	Dominant GPSCs (%)
≤5	3873	75	1547	78.8
5 - 20	391	7.6	112	5.7
20 - 60	835	16.2	284	14.5
60 - 80	60	1.2	20	1
≥80	7	0.1	1	0.1
Unknown	1744	-	611	-

Comment 2.9 Maps of population density and mobility data would be of interest in the supplemental material.

Response: Thank you for this comment, we have included the radius of gyration from the Meta Data for Good human mobility data and the population density in Figure S22A.

Specific Changes to Document:

“We define the radius of gyration (R_i) for each municipality as:

$$R_i = \sqrt{\sum_j m_{ij} d_{ij}^2}$$

Where d_i is distance to region i and m_i is the probability of mobility to region i . The sum is across all municipalities ($N=234$)(Figure S22B).”

Fig. S22. Data descriptions across the 234 municipalities of South Africa. A) Population density as estimated given the area of each municipality and the populations estimated by LandScan and B) the radius of gyration for each municipality given the distance between each municipality at the centroid weighted by the human mobility data from Meta Data for Good.

Reviewer Reports on the First Revision:

Referee #2 (Remarks to the Author):

In this revision, the authors have satisfactorily addressed the points raised in the first review. They have incorporated phylogenetic uncertainty into their analysis, which did not impact the conclusions of the manuscript. They have also added additional discussion putting their work in context with previous work on negative frequency dependent selection in pneumococcus and added relevant details on genomic methods, which were previously missing.

Referee #3 (Remarks to the Author):

Thank you for the opportunity to review the revised manuscript. The authors have made considerable revisions based on the reviewer comments and the current version is significantly improved. In particular, I appreciate the addition of the new analysis that considers serotype and GPSC fitness, which indicates a significant role of the genomic background of the strain. Overall, the response to the reviewers was one of the most detailed and comprehensive that I have seen. I have no further comments at this time.

Referee #5 (Remarks to the Author):

The authors undertake an important evaluation of pneumococcal molecular epidemiology and transmission in the context of conjugate pneumococcal vaccine pressure.

I will limit my review to that of bacteriology and vaccinology, rather than critiquing the model itself.

From the outset, the language around availability of PCV's (line 83-ish) will need to be updated in light of recent approval of PCV15 (Merck) and PCV20 (Pfizer). One could also ask whether PPSV should be mentioned (PPSV23), though its impact on colonization would be less critical.

The authors' data regarding PCV13-relevant serotypes is consistent with other investigators, adding external validation to this aspect of their dataset, which is highly encouraging. The model also provides a sense of the increased fitness potential of non-vaccine serotypes, providing an objective measure of how NVT may fill the colonization niche previously occupied by VT strains.

The authors use fitness a bit differently than I would have anticipated it, but I'm supportive of it. We sometimes limit bacterial fitness to growth parameters in a plate or inherent replication characteristics based on genome size, etc. In fact, there is a dogma that would suggest that as bacteria amass additional resistance elements, fitness (overall) decreases. The authors should be commended for ensuring that we understand fitness in the context in which the organism finds itself. AMR-resistant pneumococci appear to have substantially (and significantly) higher fitness in a post-PCV13 world in which penicillin is used somewhat nondiscriminately. These data may prove helpful as we determine how to keep ahead of pneumococci after introduction of new vaccines that cover additional serotypes.

I think the manuscript could be improved with 2-3 additions to the text.

1. Given the approval and widespread use of PCV15 and PCV20 in the US, there needs to be additional thought in the manuscript to the serotypes now included in the vaccine and whether all NVT strains are 'created equally.' they of course are not, as shown by your pre-PCV fitness estimates, but there are constraints to how well a NVT can occupy a niche after PCV introduction.
2. I may have missed it in some of the time-to-event modeling, but can the authors be more clear as to whether there is an estimated honeymoon period between introduction of an expanded PCV vaccine (or any pneumococcal vaccine that affects colonization dynamics) and substantial emergence of NVT strains, particularly AMR strains?
3. How do the authors account for the impact of frequent introductions to the population of VT/NVT serotypes from across the globe? I think I missed this in the model.

Referee #6 (Remarks to the Author):

In this paper, Belman et al use geolocated pneumococcus isolates, bacterial genomics, and develop a new probabilistic mobility model that incorporates human mobility data to reconstruct *S. pneumoniae* transmission across south Africa before and after the introduction of the pneumococcal conjugate vaccines. The modelling appears innovative and robust, with comprehensive supplementary material provided to aid interpretation of much of the methodology. The areas of novelty that would be of interest to epidemiology, microbiology and vaccine development are the inclusion of human mobility data from Meta, and the quantitative measures of fitness changes after vaccine implementation.

I have included some areas where I feel further clarification or acknowledgement would improve the manuscript.

Major Comments:

1. Mobility data from during a pandemic:

The authors used Meta values for an 18 month period during the COVID-19 pandemic (Jan 2020-July 2021) to infer human mobility data during the study period (up to 10 years prior). However, South Africa imposed restrictions on human mobility through lockdowns and 'Alert levels', which included the specific restriction of movement between provinces (see link below) – the precise variables the authors are using the data to model. These measures could be expected to impact mobility measures between municipalities, and the authors should acknowledge this confounder in their description of the meta data and discuss whether this would affect their classifications of “the focal nature of human mobility” and “slow cross-border transmission”

<https://www.gov.za/covid-19/about/about-alert-system#:~:text=The country was on adjusted alert level 4 from 28,4 from 16 June 2021>

2. Single Transmission events:

The authors stress throughout their manuscript the novelty of their method to describe “each transmission generation”. However it is not clear from either their main figures or their extensive supplementary data how closely related the ~6900 sequences were that their model was built upon. Specifically, while figure S6 includes a histogram of ‘transmission generations’ for data from the Gambia and Kenya, I cannot find a similar description for the South African samples. This would be helpful to allow to reader to understand how much extrapolation occurred, or if there were indeed very closely related samples that could be inferred as a true ‘transmission events’. In several places throughout the authors include strong claims about being able to “translate” or “elucidate” biology at single transmission level, however this would appear to be mostly inferred from their modelling rather than definitive data on transmission between two individuals in time and space. These claims could be toned down or further clarified.

3. Combination of disease and carriage data.

The disease data is spread across the whole study period. In contrast the carriage data is all 2010 onwards (post PCV), from 2 provinces only, includes unvaccinated adults. While these samples likely

capture a representation of what strains are circulating, it is unclear what influence these carriage samples from adults would have on the estimates of vaccine-induced fitness changes, particularly when compared to the pre-2009 data which is only from disease isolates. The authors state that they tested all their models for robustness with carriage and disease data separately, however these comparisons have not been included in most places. This would be beneficial to include, particularly as 1) Figure S12 the authors show that there is differences in prevalence of VT and NVT strain types between the carriage and disease datasets, and 2) in FigS11 one of the provinces with carriage data, Guateng, has one of the most striking increases in NVT and reductions in PCV serotypes from the whole dataset. The authors could include data modelled as per Figures 4D-G but with just 'disease' data.

4. Modelling of Serotype fitness and vaccine impact:

The authors state that when building the model, they whether a single fitness shift in 2009 or multiple shifts reflecting the two vaccine roll outs fit the data better, with a single 2009 shift returning the best WAIC value (Fig S28). However, it is difficult to see how a 2009 shift is appropriate for modelling the PCV13-only strains as this precedes vaccine deployment by 2 years. This doesn't also seem to reflect the data for the PCV-13 only strains (E.g. Figure 4 and Fig S13) where strongest effect seem to occur from 2013 onwards for serotypes 19A and 6A, with the only other PCV13 with a strong downward trend - serotype 1 - trending downwards before PCV13 rollout.

Could the authors clarify whether in their modelling of a '2 fitness shifts' did that incorporate that the PCV7 strains would be affected from 2009, but the PCV13 strains from 2011 onwards? If this was the case, the supplementary methods should include this additional detail. If this wasn't the case, and all vaccine strains were included together, then splitting by vaccine content could be explored.

5. Penicillin resistance among NVT strains:

Figure S20 nicely shows that the NVT are increasing for both susceptible and resistance strains after implementation of the vaccine, and the authors describe a 1.3 increase in fitness. The authors state that prior to the vaccine, 8.8% of NVT strains were penicillin resistant. In Fig 4G, the distribution for the prevalence of resistance among NVT serotypes does not seem to have strongly deviated from this value, and the models error overlaps with the pre-vaccination time-points.

While I agree that their data in S20 seems to support the claim that the "the overall reduction in penicillin resistance seen following vaccine implementation may not persist", it seems that the data included in Fig4G and S21 poorly supports this point.

Minor comments or requests for clarification:

1. Vaccination coverage: From VIEW-hub.org it appears that 81% coverage in young children by 2012. Is there any information about vaccination coverage at an individual province level? Does your model account for vaccination coverage in fitness calculations?
2. Fig S19 – The legend is missing the word "resistance"?
3. The acronym IPD is not defined.
4. Fig S26 legend generation time should be in days rather than "years".

Author Rebuttals to First Revision:

Referees' comments:

Referee #2 (Remarks to the Author):

Comment 2.1., *In this revision, the authors have satisfactorily addressed the points raised in the first review. They have incorporated phylogenetic uncertainty into their analysis, which did not impact the conclusions of the manuscript. They have also added additional discussion putting their work in context with previous work on negative frequency dependent selection in pneumococcus and added relevant details on genomic methods, which were previously missing.*

Response: We thank the reviewer for their positive feedback on the paper.

Referee #3 (Remarks to the Author):

Comment 3.1., *Thank you for the opportunity to review the revised manuscript. The authors have made considerable revisions based on the reviewer comments and the current version is significantly improved. In particular, I appreciate the addition of the new analysis that considers serotype and GPSC fitness, which indicates a significant role of the genomic background of the strain. Overall, the response to the reviewers was one of the most detailed and comprehensive that I have seen. I have no further comments at this time.*

Response: We thank the reviewer for their positive feedback on the paper.

Referee #5 (Remarks to the Author):

The authors undertake an important evaluation of pneumococcal molecular epidemiology and transmission in the context of conjugate pneumococcal vaccine pressure. I will limit my review to that of bacteriology and vaccinology, rather than critiquing the model itself.

Comment 5.1. *From the outset, the language around availability of PCV's (line 83-ish) will need to be updated in light of recent approval of PCV15 (Merck) and PCV20 (Pfizer). One could also ask whether PPSV should be mentioned (PPSV23), though its impact on colonization would be less critical.*

Response: Thank you for this comment. We have now included details about PPSV23 and PCV15 and PCV20 with citations about the history of vaccination and recent licensure in the USA and Europe.

“A pneumococcal polysaccharide vaccine (PPSV) including 23 serotypes was licensed in the USA in 1983 but the absence of mucosal immunity has seen it replaced by pneumococcal conjugate vaccines (PCVs) except for in older adults and the immunocompromised(12). PCVs target a small subset of the polysaccharide capsular serotypes, with the most common formulations having included PCV7 and PCV13 (Pfizer)(13) and PCV10 (GSK)(14). These target 7, 13, and 10 serotypes respectively (with all the serotypes included within PCV7 and PCV10 also included within PCV13). Additional PCV formulations PCV15 (Merck) and PCV20 (Pfizer) were licensed for use in the United States and Europe in 2021(15). Pneumococcal vaccination is dynamic with new vaccine compositions often being tested.”

12. D. M. Musher, R. Anderson, C. Feldman, *The remarkable history of pneumococcal vaccination: an ongoing challenge. Pneumonia* 14, 5 (2022).

14. M. Kobayashi, J. L. Farrar, R. Gierke, A. Britton, L. Childs, A. J. Leidner, *Use of 15-valent pneumococcal conjugate vaccine and 20-valent pneumococcal conjugate vaccine among U.S. adults: Updated Recommendations of the Advisory Committee on Immunization Practices - United States, 2022. MMWR Morb Mortal Wkly Rep* 71, 109–17 (2022).

Comment 5.2. *The authors' data regarding PCV13-relevant serotypes is consistent with other investigators, adding external validation to this aspect of their dataset, which is highly encouraging. The model also provides a sense of the increased fitness potential of non-vaccine serotypes, providing an objective measure of how NVT may fill the colonization niche previously occupied by VT strains.*

Response: We thank the reviewer for this positive comment - we also concluded the results were nicely in line with previous studies, and are very excited by the results of the fitness model.

Comment 5.3. *The authors use fitness a bit differently than I would have anticipated it, but I'm supportive of it. We sometimes limit bacterial fitness to growth parameters in a plate or inherent replication characteristics based on genome size, etc. In fact, there is a dogma that would suggest that as bacteria amass additional resistance elements, fitness (overall) decreases. The authors should be commended for ensuring that we understand fitness in the context in which the organism finds itself. AMR-resistant pneumococci appear to have substantially (and significantly) higher fitness in a post-PCV13 world in which penicillin is used somewhat nondiscriminately. These data may prove helpful as we determine how to keep ahead of pneumococci after introduction of new vaccines that cover additional serotypes.*

Response: We thank the reviewer for this positive comment. We are happy with how using fitness across a microbial population provides a tractable measure of microbial dynamics as a whole.

Comment 5.4. I think the manuscript could be improved with 2-3 additions to the text.
1. Given the approval and widespread use of PCV15 and PCV20 in the US, there needs to be additional thought in the manuscript to the serotypes now included in the vaccine and whether all NVT strains are 'created equally.' they of course are not, as shown by your pre-PCV fitness estimates, but there are constraints to how well a NVT can occupy a niche after PCV introduction.

Response:

This is an interesting perspective. We agree with the reviewer that all strains are not 'created equally'. From our approach it is possible to quantify the underlying fitness of different serotypes prior to decision making on which serotypes to include into a new vaccine.

We have now included the following text in the revised manuscript: “Our findings highlight that all strains are fundamentally different in underlying fitness (Figure S15) meaning that NVTs will differ in their ability to alter their ecological niche following changes in vaccine formulation. This needs to be taken into consideration in the development of new vaccine formulations.”

Figure S15. Fitness estimates pre- and post-PCV (y-axis) for each serotype (grey), superimposed by group including NVT (blue), PCV7 (green), and PCV13 (red). Pre-vaccine and post-vaccine refer to pre- and post- 2009 and 2011 for PCV7 and PCV13 respectively. Individual serotype fitness estimates can be found in Figure S17.

Furthermore we have explicitly highlighted those NVTs which are included in PCV15 and PCV20 in S19 (old figure number Figure S17).

Comment 5.5. 2. I may have missed it in some of the time-to-event modeling, but can the authors be more clear as to whether there is an estimated honeymoon period between introduction of an expanded PCV vaccine (or any pneumococcal vaccine that affects colonization dynamics) and substantial emergence of NVT strains, particularly AMR strains?

Response: We thank the reviewer for their comment - this is an interesting question. We have explored the time-lag between vaccine implementation and change in fitness of circulating strains. We identify that the change is rapid (the best fitting model has a change in the same year as the implementation) - see figure below.

Fig. S14. Testing year of fitness switch for the logistic growth -fitness model. Adjusting the year of the fitness switch in the model fitting to vaccine status. The difference to the best WAIC (2009 [PCV7 implementation] & 2011 [PCV13 implementation]) is on the y-axis where the year of fitness switch relative to 2009 & 2011 is on the x-axis. Further we test no fitness switch (ns; yellow) and testing the impact of including one fitness switch in 2009 (purple). The dark gray box highlights equivalent models ($\Delta\text{WAIC} \leq 2$) and light gray box highlights similar models ($\Delta\text{WAIC} \leq 7$).

We have highlighted this result in the revised manuscript: “*We assessed whether there was a delay between vaccine implementation and resulting changes in strain fitness Figure S14. We find the best fitting model assumes the change in fitness occurred in the same year as the vaccine implementation.*”

We note that the delay to identifying substantial changes in case numbers from particular types will depend on pre-existing strain frequencies prior to vaccine implementation. Therefore there isn't a simple relationship between vaccination timing and large changes (emergence or reduction) in the number of cases.

Comment 5.6. 3. How do the authors account for the impact of frequent introductions to the population of VT/NVT serotypes from across the globe? I think I missed this in the model.

Response: In our spatial analyses, we compared the genetic similarity of strains within South Africa versus pairs where one is in South Africa and one is outside South Africa (elsewhere in Africa and outside Africa). We find that the relative risk of closely related strains (MRCA <2 years) being across the South African border but still within Africa is 0.01 times the probability of both being within South Africa. The relative risk when we consider countries outside Africa is even smaller (0.006) This highlights that pathogen spread is dominated by locally circulating strains.

If transmission was driven by external introductions, in particular from countries with different vaccines, we would not be able to fit the observed distribution of strain distributions using our fitness models.

These results strongly suggest that while external introductions certainly do occur, they are insufficient to impact our estimates of the strain fitness or movement. We have included this conclusion in the revised manuscript: *“By comparing the spatial location of genetically closely related pairs, we found that pneumococcus flow was dominated by within-country movement as compared to either movement between South Africa and other African countries, or non-African countries (Figure 2C-E, Figure S5A, Table S5).”*

Referee #6 (Remarks to the Author):

Comment 6.1. *In this paper, Belman et al use geolocated pneumococcus isolates, bacterial genomics, and develop a new probabilistic mobility model that incorporates human mobility data to reconstruct S. pneumoniae transmission across south Africa before and after the introduction of the pneumococcal conjugate vaccines. The modelling appears innovative and robust, with comprehensive supplementary material provided to aid interpretation of much of the methodology. The areas of novelty that would be of interest to epidemiology, microbiology and vaccine development are the inclusion of human mobility data from Meta, and the quantitative measures of fitness changes after vaccine implementation. I have included some areas where I feel further clarification or acknowledgement would improve the manuscript.*

Response: We thank the reviewer for their thoughtful review, and overall positive comments on the paper. We have provided specific responses below.

Major Comments:

Comment 6.2. 1. *Mobility data from during a pandemic. The authors used Meta values for an 18 month period during the COVID-19 pandemic (Jan 2020-July 2021) to infer human mobility data during the study period (up to 10 years prior). However, South Africa imposed restrictions on human mobility through lockdowns and ‘Alert levels’,*

which included the specific restriction of movement between provinces (see link below) – the precise variables the authors are using the data to model. These measures could be expected to impact mobility measures between municipalities, and the authors should acknowledge this confounder in their description of the meta data and discuss whether this would affect their classifications of “the focal nature of human mobility” and “slow cross-border transmission”

[https://www.gov.za/covid-19/about/about-alert-system \[gov.za\]#:~:text=The country was on adjusted alert level 4 from 28,4 from 16 June 2021.](https://www.gov.za/covid-19/about/about-alert-system [gov.za]#:~:text=The country was on adjusted alert level 4 from 28,4 from 16 June 2021.)

Response:

The human mobility data available for use for South Africa is the Meta mobility data. We use baseline data, averaged over a 17 month period. This covers the period January 2020 to June 2021. We agree with the reviewer that there were some severe mobility restrictions during this period, including an Alert Level 5 (i.e. lockdowns) from 23 March 2020 to 30 April 2020.

Critically, our analytical approach specifically accounts for systematic differences between Meta users in the mobility dataset and those involved in pneumococcus transmission. These differences may be due to fundamental differences between Meta users and those involved in transmission, or due to changes in mobility as was experienced during lockdowns. We do this by including an extra parameter that changes the probability of staying within an individual's home location. We note that the model with the Meta data performs well at being able to recover the observed spatial spread in the genetic data, and outperforms gravity mobility models.

We have now included text addressing this issue more explicitly in the text under the limitations section.

“The human mobility data is Meta baseline data which was released by Meta due to the SARS-CoV-2 pandemic disaster in 2020. We used aggregated data across 17 months (January 2020-June 2021) into a single mobility pattern matrix. As mobility was altered during this period, and further, to address the possibility that Meta mobility data is different to the movement of individuals involved in pneumococcal transmission, we include an additional parameter that adjusts the human mobility data, to account for more or less time being spent in home municipalities. As we were able to obtain good fits to the observed spread of pneumococcus, and these models outperformed standard gravity models, our findings highlight how imperfect mobility data can nevertheless be useful. ”

With respect to cross-border transmission, as set out in response to comment 5.6, in our spatial analyses, we specifically compared the genetic similarity of strains within South Africa versus pairs where one is in South Africa and one is outside South Africa (elsewhere in Africa and outside Africa). We find that the relative risk of closely related strains (MRCA <2 years) being across the South African border but still within Africa is 0.01 times the probability of both being within South Africa. The relative risk when we consider countries outside Africa is even smaller (0.006). This highlights that pathogen spread is dominated by locally circulating strains. This finding is completely independent of the Meta mobility data and identifiable from just looking at the sequence data.

This result strongly suggests that while external introductions certainly do occur, they are insufficient to impact our estimates of strain fitness or movement. We have included this conclusion in the revised manuscript: “By comparing the spatial location of genetically closely related pairs, we found that pneumococcus flow was dominated by within-country movement as compared to either movement between South Africa and other African countries, or non-African countries (Figure 2C-E, Figure S5A, Table S5).”

Comment 6.3. 2. Single Transmission events:

The authors stress throughout their manuscript the novelty of their method to describe “each transmission generation”. However it is not clear from either their main figures or their extensive supplementary data how closely related the ~6900 sequences were that their model was built upon. Specifically, while figure S6 includes a histogram of ‘transmission generations’ for data from the Gambia and Kenya, I cannot find a similar description for the South African samples. This would be helpful to allow to reader to understand how much extrapolation occurred, or if there were indeed very closely related samples that could be inferred as a true ‘transmission events’. In several places throughout the authors include strong claims about being able to “translate” or “elucidate” biology at single transmission level, however this would appear to be mostly inferred from their modelling rather than definitive data on transmission between two individuals in time and space. These claims could be toned down or further clarified.

Response: As with virtually all sequence datasets that come from a national population, it is very unlikely that any pair of sequences will come from pairs of individuals that are directly transmission linked (i.e., an infector-infectee pair). We therefore need models to recover what must be happening (i.e, pathogen movement) at each transmission event for it to be consistent with the observed locations of sequences and their genetic relatedness. To do this, we use the generation time distribution to translate branch lengths from a tree into the number of transmission generations between a pair of sequences’ Most Recent Common Ancestor (MRCA) and each sequence. So for example, if the MRCA between sequences A and B is 300 days prior to sequence A, and there is a mean generation time of 30 days, then there have been an average of 10 transmission generations between the MRCA and sequence A. Using this information, we can calculate the transmission matrix for each generation that is most consistent with the ultimate locations of A and B. This is done through matrix multiplication that integrates over all possible transmission pathways linking a pair of sequences. In this way we don’t need to know exactly the pathway each strain took - but can incorporate the uncertainty of the different possible pathways. Note that we don’t just use the mean generation time (as in the toy example above) but use the whole distribution of possible generation times per generation to get a probabilistic number of generations between the MRCA and a sequence.

The data from Gambia and Kenya was used to quantify the transmission generation time distribution (i.e., the time between successive infections in a transmission chain) using the results from intensively followed cohorts where it was possible to recover the duration of carriage in an individual.

Our revised manuscript includes the following clarification:

“The relationship between behaviours of individuals at each transmission step and the overall patterns of pathogen flow between locations are complex and non-linear. Most existing phylogeographic approaches also struggle to account for changing levels of surveillance in both space and time. We have developed mechanistic models that use the generation time distribution to estimate the number of transmission events that separate MRCA from each pair of tips in a time-resolved phylogeny”

Comment 6.4. 3. Combination of disease and carriage data.

The disease data is spread across the whole study period. In contrast the carriage data is all 2010 onwards (post PCV), from 2 provinces only, includes unvaccinated adults. While these samples likely capture a representation of what strains are circulating, it is unclear what influence these carriage samples from adults would have on the estimates of vaccine-induced fitness changes, particularly when compared to the pre-2009 data which is only from disease isolates. The authors state that they tested all their models for robustness with carriage and disease data separately, however these comparisons have not been included in most places. This would be beneficial to include, particularly as 1) Figure S12 the authors show that there is differences in prevalence of VT and NVT strain types between the carriage and disease datasets, and 2) in Fig S11 one of the provinces with carriage data, Guateng, has one of the most striking increases in NVT and reductions in PCV serotypes from the whole dataset. The authors could include data modelled as per Figures 4D-G but with just ‘disease’ data.

Response: Figure S12 was included to show that trends observed in ‘carriage-only’ versus ‘disease-only’ were consistent. We have now also included complete sensitivity analyses comparing the full dataset against only disease data and find consistent results. As carriage data only comes from two provinces, we did not consider a full model using data on carriage only. Please find below the model fits using all the data and with disease data (now included in Figure S13).

Figure S13. Comparison of fitness model results with full data or disease only data.

(A-B) Results with the full data. (C-D) Results with the disease-only data.

A and C present the model fits for the proportion of serotypes from non-vaccine type (NVT), PCV7 types, and additional PCV13 types not included in PCV7 from the years 2000 to 2014 in this study. Points represent data and lines represent the model fit. B and D present the relative fitness estimates for all three groups of serotypes in the post-vaccination (post-2009) to the pre-vaccination (pre-2009) eras. Error bars represent 2.5 and 97.5 percentiles.

We also include the fits for Gauteng here to demonstrate the continuing consistency across both datasets (see below).

Comment 6.5. 4. Modelling of Serotype fitness and vaccine impact:

The authors state that when building the model, they whether a single fitness shift in 2009 or multiple shifts reflecting the two vaccine roll outs fit the data better, with a single 2009 shift returning the best WAIC value (Fig S28). However, it is difficult to see how a 2009 shift is appropriate for modelling the PCV13-only strains as this precedes vaccine deployment by 2 years. This doesn't also seem to reflect the data for the PCV-13 only strains (E.g. Figure 4 and Fig S13) where strongest effect seem to occur from 2013 onwards for serotypes 19A and 6A, with the only other PCV13 with a strong downward trend - serotype 1 - trending downwards before PCV13 rollout.

Could the authors clarify whether in their modelling of a '2 fitness shifts' did that incorporate that the PCV7 strains would be affected from 2009, but the PCV13 strains from 2011 onwards? If this was the case, the supplementary methods should include this additional detail. If this wasn't the case, and all vaccine strains were included together, then splitting by vaccine content could be explored.

Response: We thank the reviewer for this comment. We completely agree that a two shift model would be the most biologically relevant model, rather than our original simplified approach. We have therefore revisited the model to incorporate this change. In the updated manuscript, as the reviewer suggests, we consider a new model that has the following parameterisation which accounts for the specific timing of the PCV impacting each group of serotypes (see below).

We now allow for two shifts, each of them affecting only the serotypes included in the vaccine. This model has the best WAIC (see below). Importantly, this model performs slightly better than the model with one common shift for both PCV7 and PCV13. We thank the reviewer for encouraging our additional consideration here and have updated the manuscript accordingly.

Fig. S14. Testing year of fitness switch for the logistic growth -fitness model. Adjusting the year of the fitness switch in the model fitting to vaccine status. The difference to the best WAIC (2009 [PCV7 implementation] & 2011 [PCV13 implementation]) is on the y-axis where the year of fitness switch relative to 2009 & 2011 is on the x-axis. Further we test no fitness switch (ns; yellow) and the impact of including one fitness switch in 2009 (purple). The dark gray box highlights equivalent models ($\Delta WAIC \lesssim 2$) and light gray box highlights similar models ($\Delta WAIC \lesssim 7$).

We have updated the manuscript throughout with this new model parameterization. The results are essentially unchanged.

Comment 6.6. Penicillin resistance among NVT strains:

Figure S20 nicely shows that the NVT are increasing for both susceptible and resistance strains after implementation of the vaccine, and the authors describe a 1.3 increase in fitness. The authors state that prior to the vaccine, 8.8% of NVT strains were penicillin resistant. In Fig 4G, the distribution for the prevalence of resistance among NVT serotypes does not seem to have strongly deviated from this value, and the models error overlaps with the pre-vaccination time-points.

While I agree that their data in S20 seems to support the claim that the “the overall reduction in penicillin resistance seen following vaccine implementation may not persist”, it seems that the data included in Fig4G and S21 poorly supports this point.

Response: While we agree there is some uncertainty around the estimates for penicillin resistant NVT strains, our fitness model explicitly explores the changing fitness in the different strain types, taking account of uncertainty from the differential level of sampling over time. This fitness model is able to recover the observed trends in the proportion of strains that are resistant over time. We find that the fitness of penicillin resistant NVT strains following the implementation of the vaccine was highly significant (RR: 1.30 [95% CI 1.19-1.43]). Our results therefore provide strong evidence that there is a significant increase in fitness for NVT resistant strains, which will result in ongoing increases in the proportion of NVT strains which will be penicillin resistant. This is also supported by the strong trend in penicillin resistant NVT strains between 2010 and 2015 (Figure 4G). Further, our estimates of the increased fitness of penicillin resistant NVT strains remain highly significant in sensitivity analyses where we consider models with different carrying capacities (see below, zoomed in from Figure S31B).

Fig. S31. Fitness estimates with varying carrying capacity for penicillin resistant isolates across A) VT serotypes and B) NVT serotypes where the top row for each is the relative fitness pre-PCV across carrying capacities. and the bottom row is post-PCV introduction relative fitness estimates.

Minor comments or requests for clarification:

Comment 6.7.1. Vaccination coverage: From VIEW-hub.org it appears that 81% coverage in young children by 2012. Is there any information about vaccination coverage at an individual province level? Does your model account for vaccination coverage in fitness calculations?

Response:

We have not been able to access vaccination coverage at individual province level. Our models do not account for vaccination coverage. We do stratify by each province and see consistent trends across the 9 provinces (Fig S11). Further, it's likely that vaccination coverage in South Africa varies more within provinces than between provinces due to nuances in vaccine access across demographics. As of 2022 89% coverage is reported across South Africa (<https://view-hub.org/>). We include this point in the limitations:

“Vaccination levels for PCV7 were reported to be 89% by 2022. We did not have the data to consider changes in coverage in time or across provinces. Other settings with different levels of coverage may observe different fitness effects from the implementation of the vaccine”

Comment 6.8.2. *Fig S19 – The legend is missing the word “resistance”?*

Response: We have now amended the legend to include that we are referring to antimicrobial resistance in what is now Figure S21.

Comment 6.9.3. *The acronym IPD is not defined.*

Response: We have defined it on line 96.

Comment 6.10.4. *Fig S26 legend generation time should be in days rather than “years”.*

Response: We have now fixed the figure legend in what is now Figure S28.

Reviewer Reports on the Second Revision:

Referee #5 (Remarks to the Author):

The authors have strengthened the manuscript by responding so intentionally to comments suggested by the reviewers. I have no further comments.

Referee #6 (Remarks to the Author):

In this revision, the authors have provided a comprehensive and easy to follow response to reviewers document and made several changes to improve the manuscript. Overall, the addition of further descriptions in the text of the manuscript to address my and other reviewer's comments has improved the manuscript. To my knowledge, the modelling and statistical approaches are robust, well described in the Reporting Summary, and include appropriate confidence intervals. I have included a point by point discussion of the authors' response to my original comments below. With the exception of comment 6.6, the authors have sufficiently addressed my concerns. I have included a few new comments for other minor errors I have identified.

Comment 6.2. The authors have included text to sufficiently address my concerns.

Comment 6.3: The authors have included a useful statement at line 125, particularly "use the generation time distribution to estimate the number of transmission events separating MRCAs from each pair of tips in a time-resolved phylogeny"

I still find the use of "elucidate" in line 129 to be a bit of an overinterpretation of the model, and feel that alternative words like "infer" or "estimate" would be more appropriate.

Comment 6.4: The addition of Figure S13 by the authors shows that the models are robust and consistent with and without carriage data.

Comment 6.5: I was very pleased to see that the authors amended their model to include two time switches in their fitness calculations, and Figure S14 nicely shows that this new model is superior. I

found the schematic figure representing the time shifts the authors included in their rebuttal document to be very helpful, and suggest the authors include this within Fig S14.

Comment 6.6: To this reviewer, the finding of increasing penicillin resistance among NVT types is the most novel finding of this work, however it is also the one I find the most difficult to reconcile with the raw data on strain prevalence provided. In the rebuttal document authors describe their finding of a highly significant increase in RR of 1.3 for NTV resistant strains following vaccine implementation “is able to recover the observed trends in the proportion of strains that are resistant over time.” While my expertise does not enable me to fully understand all the nuances of their fitness modelling, their results of their model seem at odds with the data in the following ways:

- Figure 4G shows that the proportion resistant among NVT’s is highly variable between years. While it appears there might be a trend towards an increase post 2011, these values still have not dramatically deviated from pre-vaccination levels, particularly when compared to the early 2000’s.
- Figure S23D is very convincing of a striking overall increase in NVT post vaccine implementation. However, similar to 4G, the data in S23 E and F indicate the high degree of variability over time and overall a very minor effect.
- Fig S22 shows that the upwards trend for NVT susceptible strains is much more pronounced than for the NVT resistant strains, which is counterintuitive for the increased fitness for NVT resistant strains
- For the analysis of AMR the authors have chosen not to differentiate between PCV7 and PCV13 for fitness shift estimates to “keep the model tractable” (line 1101). The authors could comment on how the fitness estimates for penicillin resistance vary between the PCV7 and PCV13. For example, does the model still return robust fitness advantage when just the PCV7 strains are included? This seems particularly relevant as the upwards trend for resistance seems to occur post 2011 (E.g. Fig22B).

Additionally, the authors could consider making a change to Figure 4 to include a graph of relative fitness across the four groups (VT_R, VT_S, NST_R, NVT_S) similar to 4F, as currently I find the Figs 4G, S23 and S22 alone do not convincingly support the authors claims.

Minor comments:

1) Figure legends for Fig S16 and 17: Both figs refer to serotype fitness estimates, however the scale of graphs is different. Is this correct?

2) Line 325 : consider using future tense to describe AMR trends

3) Fig S11 The vertical line is mentioned in the legend but missing on the plot. Authors should include as they have done for Fig S16.

4) Line 336 : "... NVT resistant strains were 1.30 [95% CI 1.19-1.43] times as fit as penicillin-susceptible NVT strains (Figure S22A-D)." Figure S22A-D shows the proportions of the various groups, but doesn't show the fitness estimates. Referring to Fig S31 seems more appropriate.

Referee #6 (Remarks on code availability):

Not my area of expertise

Referee #7 (Remarks to the Author):

This article was a very interesting look into the spread and evolution of *Streptococcus pneumoniae* across South Africa. Interestingly, the authors found a correlation with trends in Meta mobility data--a striking result even though the Meta data were from mostly during the pandemic.

I felt this was overall very strong methodologically and the figures conveyed the main points well. While I had a few questions and comments throughout (mostly regarding the usage of Meta mobility data to predict time spent at home/travel to other locations), I also believe these comments were adequately addressed in the responses to other reviewers.

Referee #7 (Remarks on code availability):

I downloaded the GitHub repository and the output files. The code is well-commented--the authors have done a great job here--but I do think a more comprehensive README would help in terms of figuring out how specific data link to processing files/models/figures. I was able to figure it out, so everything's there, but it would be easier for someone doing similar research to use this code if there was more detail in the README describing how data linked ultimately to specific outputs/figures.

I had some success running the code and recreating the figures, but only got so far, partly because I think a file may be missing from the GitHub repository (`GPS_SA.AMR.RData`). Looking through the code it looks like maybe it is mostly identical to `GPS_SA_GPSC.RData`, but it is missing a variable or two. Many of the downstream processing and model files depend on it though, so I couldn't run anything that depended on that initially.

The authors provided some of the output data on a separate website, which could be used to create some of the figures. This was very helpful, but I had the same issue here--some of the figures depended on outputs generated originally from `GPS_SA.AMR.RData`, so I couldn't recreate any figures that depended on that data.

These are very minor issues that I don't think need to be fixed before this paper is published, but I do think it would be helpful if the authors provided `GPS_SA.AMR.RData` or clarified why it's unavailable.

Author Rebuttals to Second Revision:

Reviewer comments

Referee #5 (Remarks to the Author):

Comment 5.1. *The authors have strengthened the manuscript by responding so intentionally to comments suggested by the reviewers. I have no further comments.*

Response: We thank the reviewer for their kind feedback.

Referee #6 (Remarks to the Author):

Comment 6.1. *In this revision, the authors have provided a comprehensive and easy to follow response to reviewers document and made several changes to improve the manuscript. Overall, the addition of further descriptions in the text of the manuscript to address my and other reviewer's comments has improved the manuscript. To my knowledge, the modelling and statistical approaches are robust, well described in the Reporting Summary, and include appropriate confidence intervals. I have included a point by point discussion of the authors' response to my original comments below. With the exception of comment 6.6, the authors have sufficiently addressed my concerns. I have included a few new comments for other minor errors I have identified.*

Response: We thank the reviewer for taking the time to go through each comment and overall positive feedback on the paper. We have addressed the remaining concerns below.

Comment 6.2. *The authors have included text to sufficiently address my concerns.*

Response: None required.

Comment 6.3: *The authors have included a useful statement at line 125, particularly "use the generation time distribution to estimate the number of transmission events separating MRCAs from each pair of tips in a time-resolved phylogeny". I still find the use of "elucidate" in line 129 to be a bit of an overinterpretation of the model, and feel that alternative words like "infer" or "estimate" would be more appropriate.*

Response: We have now changed "elucidate" to "infer" as suggested.

Comment 6.4: *The addition of Figure S13 by the authors shows that the models are robust and consistent with and without carriage data.*

Response: None required.

Comment 6.5: *I was very pleased to see that the authors amended their model to include two time switches in their fitness calculations, and Figure S14 nicely shows that this new model is superior. I*

found the schematic figure representing the time shifts the authors included in their rebuttal document to be very helpful, and suggest the authors include this within Fig S14.

Response: We thank the reviewer for this feedback and agree that this was a very nice additional analysis. As suggested, we have now included this schematic in figure S14.

Fig. S14. Fitness growth model testing year of switch and schematic. (A) Testing year of fitness switch for the logistic growth -fitness model. Adjusting the year of the fitness switch in the model fitting to vaccine status. The difference to the best WAIC (2009 [PCV7 implementation] & 2011 [PCV13 implementation]) is on the y-axis where the year of fitness switch relative to 2009 & 2011 is on the x-axis. Further we test no fitness switch (ns; yellow) and the impact of including one fitness switch in 2009 (purple). The dark gray box highlights equivalent models ($\Delta WAIC < 2$) and light gray box highlights similar models ($\Delta WAIC < 7$). (B) Schematic denoting the fitness growth model parameterisation which accounts for the specific timing of the PCV impacting each group of serotypes (NVT in blue; PCV7 in green; PCV13 in red).

Comment 6.6A-B: To this reviewer, the finding of increasing penicillin resistance among NVT types is the most novel finding of this work, however it is also the one I find the most difficult to reconcile with the raw data on strain prevalence provided. In the rebuttal document authors describe their finding of a highly significant increase in RR of 1.3 for NTV resistant strains following vaccine implementation “is able to recover the observed trends in the proportion of strains that are resistant over time.” While my expertise does not enable me to fully understand all the nuances of their fitness modelling, their results of their model seem at odds with the data in the following ways:

Figure 4G shows that the proportion resistant among NVT's is highly variable between years. While it appears there might be a trend towards an increase post 2011, these values still have not dramatically deviated from pre-vaccination levels, particularly when compared to the early 2000's.

Figure S23D is very convincing of a striking overall increase in NVT post vaccine implementation. However, similar to 4G, the data in S23 E and F indicate the high degree of variability over time and overall a very minor effect.

Response. We thank the reviewer for this comment. We agree that, by eye, there appears year to year variability and the trend is not completely clear. However, it is also important to note that visual qualitative trends are not robust and cannot provide a quantitative assessment of trends, especially when there are substantial differences in the size of the data in each year. To address the reviewer concerns we have conducted two additional analyses.

Firstly, we developed a competing model that assumed no fitness change following the vaccine, and compared the model fit with the model that does include a fitness change. We find that the model that includes a fitness shift in 2009 explains the dynamics significantly better than one without a change as demonstrated by the WAIC in Figure S30.

Fig. S30. Model comparison for the fitness growth model when estimating fitness for penicillin resistance among VTs and NVTs. We include a model with a change in fitness in 2009 upon PCV7 implementation and a model with no switch in fitness.

Secondly, in order to provide a comprehensive assessment of whether there is evidence of change in any of the time series using an independent method, we tested each time series for potential change-point in the trends using the Buishand U test (Buishand, TA, 1984) (1). This is an agnostic and flexible test and provides statistical support for the presence or absence of a change in trend. This independent method identified a significant change in all the time series considered, with the exception of the AMR within vaccine-types (see Table below). Together, these provide comprehensive support for the findings of fitness changes following the implementation of the vaccine and are in complete agreement with the findings of our manuscript.

Trend analysis, using Buishand U Test (Buishand, 1984), to detect a potential single change-point in time series. The null hypothesis of the test each time is that there is no change in trend in the time series.

Time series	NVT Sucep vs. Total (Figure S22A)	NVT AMR vs. Total (Figure S22B)	VT Sucep vs. Total (Figure S22C)	VT AMR vs. Total (Figure S22D)	VT Sucep vs. All VT (Figure S23B)	NVT Sucep vs. All NVT (Figure S23E)
p-value	<0.001	0.0012	0.0022	0.00025	0.43	0.035

Comment 6.6C. Fig S22 shows that the upwards trend for NVT susceptible strains is much more pronounced than for the NVT resistant strains, which is counterintuitive for the increased fitness for NVT resistant strains.

Response. We agree that the increase in NVT susceptible strains appears more pronounced than the increase in NVT resistant strains. However, it is important to note that the slope of the line is driven by both the starting frequency and the relative fitness. As NVT resistant strains are starting from a low base (i.e., they were rare prior to the implementation of the vaccine) - even very large relative fitness differences would result in small visual changes in the increasing proportion over short time frames.

To illustrate this phenomenon, we developed a simulation where we use exactly the same relative fitness parameter ($r=1$) but altered the starting proportions (see below).

Response Figure 6.6C1. Example of dynamics obtained with the same growth rate ($r=1$), but different starting proportions (colours). When looking 2 years after the start (vertical grey dashed line), depending on the starting proportion, the slope can be different, even though all curves share the same fitness parameter.

In Figure S22, the proportion of NVT *susceptible* strains in 2009 was 0.172 (95% CIs 0.153- 0.191) and they had a relative fitness parameter of 1.09 (95% CIs 1.01-1.17), while the proportion for NVT *resistant* strains in 2009 was 0.015 (95% CIs 0.010- 0.020) with a relative fitness parameter of 1.42 (95% CIs 1.34-1.50). These numbers are a reflection of the phenomenon demonstrated in the above figure whereby although the low proportioned NVT resistant strains have a high fitness this is difficult to see initially. Using a similar simulation framework for multiple strains, we see that our estimated fitness parameter combinations recover the observed increases in proportion (see below), and after some time we do see that the strains with the highest fitness take over.

Response Figure 6.6C2. Strain group proportions 10 years after vaccine implementation given the estimated fitness parameters. The group with the highest fitness parameter 1.42 (NVT_R; red) becomes the dominant strain after some time. Other fitness parameter estimates are NVT_S; green (1.09), VT_R; yellow (0.90), VT_S; blue (0.93).

Comment 6.6D. For the analysis of AMR the authors have chosen not to differentiate between PCV7 and PCV13 for fitness shift estimates to “keep the model tractable” (line 1101). The authors could comment on how the fitness estimates for penicillin resistance vary between the PCV7 and PCV13. For example, does the model still return robust fitness advantage when just the PCV7 strains are included? This seems particularly relevant as the upwards trend for resistance seems to occur post 2011 (E.g. Fig22B).

Response D. The AMR model relies on grouping strains by vaccine type, susceptibility to penicillin and province. We chose to not differentiate between PCV7 and PCV13 in this fitness model to minimise the number of groups as much as possible, while remaining biologically sensible. As shown by our model comparison in Figure S14A, the models differentiating PCV7 and PCV13, or using a common fitness parameter with a shift in 2009, are the closest models in terms of WAIC ($\Delta WAIC = 9.40$).

Nonetheless, we agree with the reviewer that testing if the fitness estimates for penicillin resistance would vary between PCV7 and PCV13 is important. To this end, we have conducted a further sensitivity analysis to determine whether there is a difference in the contribution of PCV7 serotypes and PCV13 serotypes to penicillin resistance in the population. We divide the fitness parameter by year including pre-PCV (2000-2009), PCV7 era (2009-2010), and PCV13 era (2011-2015). We find that our estimates of the relative fitness of penicillin resistant strains was unchanged between the PCV7 and PCV13 era.

For the NVT_R group the values for 2009-2010 were 1.49 (95% CIs 1.42-1.57) while those post-2011 were 1.38 (95% CIs 1.30-1.47) and for the VT_R group the values for 2009-2010 were 0.94 (95% CIs 0.91-0.98) while those post-2011 were 0.88 (95% CIs 0.82-0.93). On average, the fitness in the PCV7 era was 0.99 [95%CIs 0.89-1.09] times that of the PCV13 era. Furthermore the model which includes parameters to account for the shift has a WAIC difference to the best model of less than 3 indicating that the models are equivalent.

Comment 6.6E. Additionally, the authors could consider making a change to Figure 4 to include a graph of relative fitness across the four groups (VT_R, VT_S, NST_R, NVT_S) similar to 4F, as currently I find the Figs 4G, S23 and S22 alone do not convincingly support the authors claims.

Response. As suggested by the reviewer, we have now included the fitness of penicillin resistance among different vaccine types in each era (pre-PCV and post-PCV) in Figure 4G. We have also created the suggested figure which nicely sets out the relative fitness across the different groupings in Figure S22E (below). Additionally we included the dashed lines in the rest of the figure highlighting the year of vaccine implementation and fitness switch.

Figure 4. Pneumococcal fitness estimates. Data (points) and model fit (lines) for the proportion of serotypes from (A) non-vaccine types, (B) PCV7 types, and (C) additional PCV13 types not included in PCV7 from the years 2000 to 2014 in this study. The long dashed line indicates the time of PCV7 implementation (2009) and the short dashed line indicates the time of PCV13 implementation (2011). (D) Relative fitness for the three groups of serotypes compared to the NVT fitness estimates in (left) the pre-PCV era and (right) the post-PCV introduction era. (E) Relative fitness estimates for all three groups of serotypes comparing the post-PCV to the pre-PCV eras. (A-E) Pre-PCV refers to prior to 2009 for NVT serotypes, prior to 2009 for PCV7 type serotypes and prior to 2011 for PCV13 type serotypes. (F) Proportion of penicillin resistance overall (black line), within NVT strains (maroon points), and within VT strains (turquoise points) with model fits. The dashed line indicates the time of PCV implementation (2009) (G) Relative fitness of penicillin resistance among NVTs (pink) and VTs (blue) in the pre-PCV era (left) and the post-PCV era (right). D, E, and G are on a log scale. Error bars represent 2.5 and 97.5 percentiles.

Fig. S22 Data fits for model accounting for proportions and fitness over time in four groups, A) NVT-penicillin resistant (red), B) NVT-penicillin susceptible (green), C) VT-penicillin resistant (yellow) and D) VT-penicillin susceptible (blue). The dashed lines indicate the year of PCV7 implementation and fitness switch model implemented. E) Includes the fitness estimates pre-PCV and post-PCV for each group and colored accordingly. E is on a log scale. This model uses a shift in fitness in 2009. Error bars represent 2.5 to 97.5 percentiles.

Minor comments:

Comment 6.7 Figure legends for Fig S16 and 17: Both figs refer to serotype fitness estimates, however the scale of graphs is different. Is this correct?

Response. The y-axis label “proportions” was absent from figure S16. We have now fixed this.

Comment 6.8. Line 325 : consider using future tense to describe AMR trends

Response. We have changed the sentence appropriately.

“The high levels of AMR in the VT strains was also present globally (24). In South Africa, similar to other countries, reduction of AMR has been noted since vaccine implementation; however, it remains unclear whether this trend will persist or if AMR eventually rebounds (24).”

Comment 6.9. Fig S11 The vertical line is mentioned in the legend but missing on the plot. Authors should include as they have done for Fig S16.

Response. We have updated Figure S16 with the vertical line.

Comment 6.10. Line 336 : "... NVT resistant strains were 1.30 [95% CI 1.19-1.43] times as fit as penicillin-susceptible NVT strains (Figure S22A-D)." Figure S22A-D shows the proportions of the various groups, but doesn't show the fitness estimates. Referring to Fig S31 seems more appropriate.

Response. We have changed the reference accordingly (though it is now figure S32).

Referee #7 (Remarks to the Author):

Comment 7.1. *This article was a very interesting look into the spread and evolution of Streptococcus pneumonia across South Africa. Interestingly, the authors found a correlation with trends in Meta mobility data--a striking result even though the Meta data were from mostly during the pandemic.*

I felt this was overall very strong methodologically and the figures conveyed the main points well. While I had a few questions and comments throughout (mostly regarding the usage of Meta mobility data to predict time spent at home/travel to other locations), I also believe these comments were adequately addressed in the responses to other reviewers.

Response. We thank the reviewer for their kind feedback on the paper.

Comment 7.1. *I downloaded the GitHub repository and the output files. The code is well-commented--the authors have done a great job here--but I do think a more comprehensive README would help in terms of figuring out how specific data link to processing files/models/figures. I was able to figure it out, so everything's there, but it would be easier for someone doing similar research to use this code if there was more detail in the README describing how data linked ultimately to specific outputs/figures.*

I had some success running the code and recreating the figures, but only got so far, partly because I think a file may be missing from the GitHub repository (GPS_SA.AMR.RData). Looking through the code it looks like maybe it is mostly identical to GPS_SA_GPSC.RData, but it is missing a variable or two. Many of the downstream processing and model files depend on it though, so I couldn't run anything that depended on that initially.

The authors provided some of the output data on a separate website, which could be used to create some of the figures. This was very helpful, but I had the same issue here--some of the figures depended on outputs generated originally from GPS_SA.AMR.RData, so I couldn't recreate any figures that depended on that data.

These are very minor issues that I don't think need to be fixed before this paper is published, but I do think it would be helpful if the authors provided GPS_SA.AMR.RData or clarified why it's unavailable.

Response. Apologies for the absence of the necessary file. We have now included the relevant RData file in the repository. We have also updated the README to include more detail.

Reviewer Reports on the Third Revision:

Referee #6 (Remarks to the Author):

I thank the authors for a comprehensive explanation of their penicillin resistance modelling, and the additional analyses to support the robustness of their findings. The authors have made changes to text and figures to improve accuracy and readability. I have no further comments.

Referee #6 (Remarks on code availability):

Outside of my area of expertise.

Referee #7 (Remarks to the Author):

I believe the authors have sufficiently addressed reviewer comments.

Referee #7 (Remarks on code availability):

I haven't double checked, but I think the authors have uploaded all the data required to run the code now. I ran the code initially as part of peer review and couldn't get it to work, so this was nice to see.